# Loss of NECTIN1 triggers melanoma dissemination upon local IGF1 depletion

**Julien Ablain** [1,2] ✉**, Amira Al Mahi**[1]**, Harriet Rothschild** [2]**, Meera Prasad**[2]**, Sophie Aires**[1]**, Song Yang**[2]**, Maxim E. Dokukin**[3,4]**, Shuyun Xu**[5]**, Michelle Dang**[2]**, Igor Sokolov**[3]**, Christine G. Lian**[5] **and Leonard I. Zon** [2,6,7]

Cancer genetics has uncovered many tumor-suppressor and oncogenic pathways, but few alterations have revealed mechanisms involved in tumor spreading. Here, we examined the role of the third most significant chromosomal deletion in human melanoma that inactivates the adherens junction gene *NECTIN1* in 55% of cases. We found that *NECTIN1* loss stimulates melanoma cell migration in vitro and spreading in vivo in both zebrafish and human tumors specifically in response to decreased IGF1 signaling. In human melanoma biopsy specimens, adherens junctions were seen exclusively in areas with low IGF1 levels, but not in NECTIN1-deficient tumors. Our study establishes *NECTIN1* as a major determinant of melanoma dissemination and uncovers a genetic control of the response to microenvironmental signals.

A clinical challenge in solid tumors remains the formation of metastases[1]. In melanoma, tumor dissemination is the leading cause of mortality[2]. Although few studies have implicated genetic lesions in tumor dissemination[3], early work in pancreatic cancer detected metastatic clones within parental primary tumors and suggested the absence of a genetic signature of metastasis[4]. More recently, the genomic comparison of metastases and matched primary tumors failed to identify genetic drivers of tumor spreading[5–7], pointing to a role for non-cancer cell-autonomous mechanisms in metastasis[8]. The relative contributions of genetic alterations and signals from the tumor microenvironment to metastatic traits remain controversial[9].

Early events implicated in local tumor invasion or detachment of cells from the primary tumor, such as alterations of cell–cell adhesion[10], have been extensively studied in epithelial tumors. For example E-cadherin loss has been reported to promote metastasis in pancreatic cancer[11], and epithelial-to-mesenchymal transition (EMT) serves as an early driver of metastasis[12,13]. However, the role of cell–cell adhesion remains elusive in non-epithelial cancers like melanoma. Melanocytes themselves emerge from the neural crest during embryonic development and thus possess more mesenchymal than epithelial features[14]. Loss of E-cadherin expression has been shown to favor melanoma progression by disrupting interactions between melanocytes and neighboring keratinocytes in coculture experiments[15], and differences in cadherin patterns, notably a switch from E-cadherin to N-cadherin, have been noted in melanoma metastases compared to primary tumors[16]. Yet, functional evidence explaining the contribution of cell–cell adhesion changes to the metastatic dissemination of non-epithelial tumors is lacking.

Here, we identified deletions of the gene *NECTIN1* as a frequent driver of melanoma dissemination. NECTIN1 is an adhesion molecule that can engage in homotypic or heterotypic interactions between two cells[17,18] to establish early adherens junctions[19]. We found that *NECTIN1* loss changes the response of melanoma cells to microenvironmental insulin-like growth factor 1 (IGF1) signaling by controlling a switch from cell–cell adhesion to cell–matrix adhesion. This mechanism integrates the status of adherens junctions and local growth factor concentration to direct the decision of cancer cells to stay in the niche or to leave.

[1]Centre de Recherche en Cancérologie de Lyon, Centre Léon Bérard, INSERM U1052 CNRS UMR5286, Lyon, France. [2]Stem Cell Program and Division of Hematology/Oncology, Boston Children's Hospital and Dana Farber Cancer Institute, Boston, MA, USA. [3]Department of Mechanical Engineering and Department of Biomedical Engineering, Tufts University, Medford, MA, USA. [4]Sarov Physics and Technology Institute, National Research Nuclear University MEPhI, Sarov, Russian Federation. [5]Program in Dermatopathology, Department of Pathology, Brigham and Women's Hospital, Boston, MA, USA. [6]Harvard Stem Cell Institute, Harvard University, Cambridge, MA, USA. [7]Howard Hughes Medical Institute, Boston, MA, USA. ✉e-mail: julien.ablain@lyon.unicancer.fr

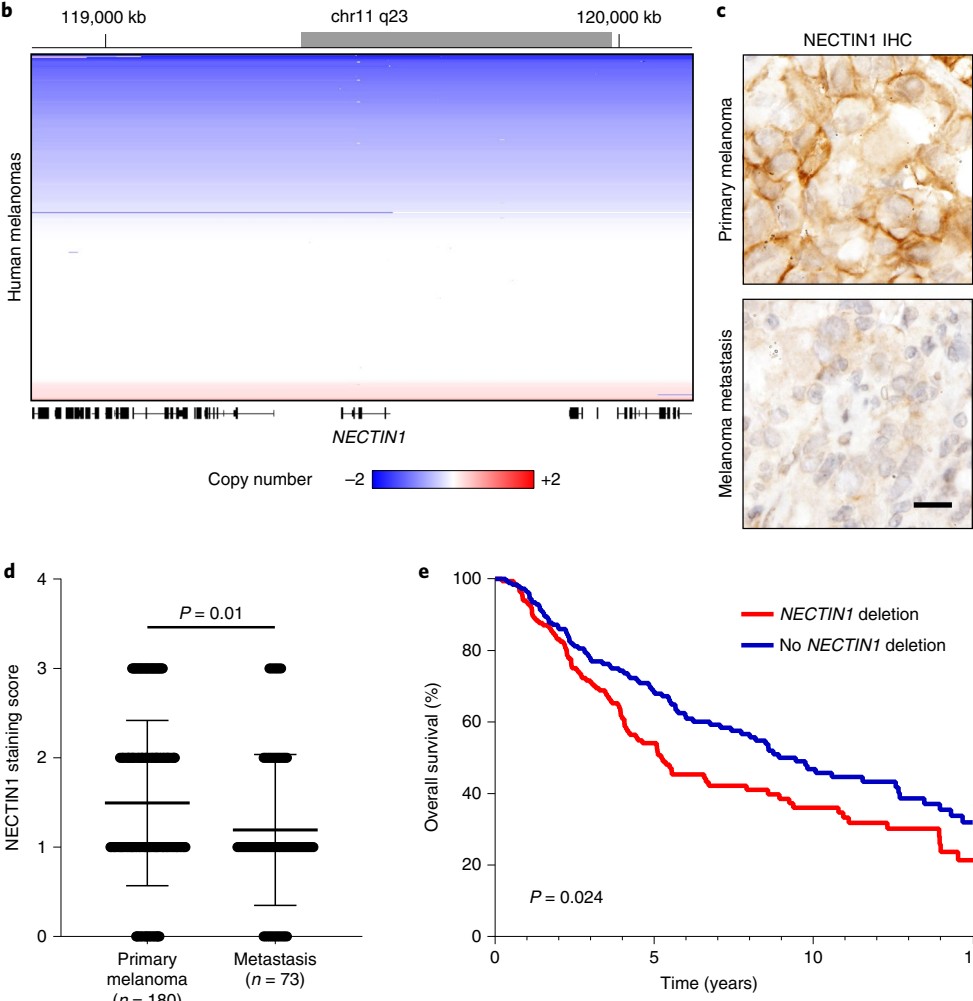

**Fig. 1 | *NECTIN1* is frequently deleted in human melanoma. a**, List of the five most significant chromosomal deletions found in TCGA dataset of 363 human cutaneous melanomas. False discovery rate *q*-values were calculated by GISTIC. **b**, Linear copy number of the *NECTIN1* locus in human melanoma (363 samples represented on the *y* axis). Blue denotes deletions, red, amplifications. The gray bar on the scale represents the focal deletion found by GISTIC analysis. **c**, Images of human melanoma sections stained for NECTIN1 by immunohistochemistry. Scale bar, 20 µm. **d**, Quantification of overall NECTIN1 staining intensity in 180 primary melanomas and 73 melanoma metastases (mean ± standard deviation (s.d.). Two-tailed *t*-test. **e**, Overall survival curves of patients with melanoma stratified according to *NECTIN1* copy number status (>−0.25: no deletion, *n* = 190; <−0.25: deletion, *n* = 168 (log-rank test).

## Results

### NECTIN1 is frequently deleted in human melanoma

By analyzing copy-number alterations[20,21] in The Cancer Genome Atlas (TCGA) cohort of 363 human melanomas, we found that the third most significant focal deletion at chr11q23.3 only contained the gene *NECTIN1* (Fig. 1a and Extended Data Fig. 1a,b). Although biallelic loss of *NECTIN1* was detected in 4.4% of cases (16 cases), half of melanomas exhibited shallow deletions (186 cases) totaling an overall frequency of deletion of 55% (Fig. 1b). Most deletions affected the entire locus, with shallow deletions likely taking out one *NECTIN1* allele and resulting in heterozygous loss (Extended Data Fig. 1c,d). *NECTIN1* loss did not correlate with any alterations in major oncogenes (*BRAF* and *NRAS*) or tumor-suppressor genes (*NF1*, *CDKN2A*, *TP53* and *PTEN*) (Extended Data Fig. 1e). *NECTIN1* mRNA expression correlated with linear copy number (Extended Data Fig. 2a), with human melanoma samples with deep or shallow deletions of *NECTIN1* expressing significantly less *NECTIN1* mRNA than diploid samples (Extended Data Fig. 2b). Immunohisto-chemistry for NECTIN1 in human melanoma revealed that NECTIN1

protein levels were significantly lower in metastases compared to primary tumors (Fig. 1c,d and Extended Data Fig. 2c). Metastases were overrepresented among samples with low NECTIN1 staining whereas samples with high NECTIN1 were enriched for primary melanomas (Extended Data Fig. 2d). In human melanoma, *NECTIN1* status did not correlate with the expression of E-cadherin, another adherens junction component whose loss has been linked to metastatic progression in multiple cancers (Extended Data Fig. 2e). Coimmunofluorescence in 20 human melanoma tissue sections also failed to reveal any correlation between NECTIN1 and E-cadherin protein levels (Extended Data Fig. 2f,g). Importantly, *NECTIN1* deletions were associated with lower overall survival of patients with cutaneous melanoma (Fig. 1e). Together, these data nominate *NECTIN1* as a frequently altered gene in melanoma with a potential role in metastasis.

### NECTIN1 loss promotes melanoma spreading in vivo

We inactivated *nectin1* in primary zebrafish melanomas with a tissue-specific CRISPR technique[22,23] (Fig. 2a and Methods). *NECTIN1* has two orthologs in zebrafish, *nectin1a* and *nectin1b*, of which only *nectin1b* is substantially expressed in primary zebrafish melanomas (Extended Data Fig. 3a). To prevent compensation between the two paralogs, we targeted both concomitantly. Sequencing of 65 primary zebrafish melanomas demonstrated high targeting efficiency at both loci (hereafter together referred to as *nectin1*) (Fig. 2b–d and Supplementary Note).

*Nectin1* inactivation in primary zebrafish melanoma modestly accelerated tumor onset (Fig. 2e). It did not significantly affect melanoma cell proliferation as assessed by immunohistochemistry for phospho-histone3 (Fig. 2f,g). By examining hematoxylin/eosin-stained longitudinal sections of 16-week-old adult zebrafish bearing primary melanomas, we assessed local tumor invasion and dissemination to distant organs including gills, thymus, kidney, spleen, liver, and intestine (Fig. 2h). Although no differences were detected in local tumor invasion into adjacent muscle, we observed significantly higher organ involvement with *nectin1*-knockout tumors than with *control* tumors (Fig. 2i,j). To evaluate the metastatic propensity of genetically-modified zebrafish melanoma cells in vivo, we used a subcutaneous transplantation assay[24], where pigmented primary zebrafish melanoma cells can migrate away from the injection site along the sides of transparent adult recipients, which we call 'spreading' (Fig. 2k). This method enables direct visualization of cancer cell behavior in a physiologically relevant environment[25]. Primary melanoma cells expressing a control CRISPR vector only rarely spread in secondary recipients (Fig. 2k,l). In contrast, *nectin1* inactivation significantly increased tumor cell spreading, as evidenced by the proportion of recipients showing patches of melanoma cells that had disseminated past the midline (Fig. 2k,l, Extended Data Fig. 3b and Supplementary Note). These results demonstrate that *nectin1* loss promotes melanoma dissemination in vivo.

### NECTIN1 loss alters cell adhesion upon serum depletion

The spreading capacity of zebrafish melanoma cells was associated with altered mechanical properties of primary tumors: *nectin1*-deficient primary zebrafish melanomas were significantly softer than their wild-type counterparts, as measured by atomic force microscopy (Extended Data Fig. 3c). To further investigate the features of *nectin1*-deficient tumors, we established cell lines from primary melanomas following a protocol involving the progressive replacement of the initial zebrafish-specific rich media by regular culture media[25]. *Nectin1*-knockout cell lines failed to passage outside of zebrafish media, as evidenced by the sharp drop in the proportion of *nectin1* CRISPR mutant alleles compared to wild-type alleles over time (Fig. 2m and Extended Data Fig. 3d). This was due at least in part to a failure to adhere to the culture dish or to each other, since cells harvested from culture supernatant could reattach and grow in zebrafish media (Extended Data Fig. 3e). Moreover, *nectin1*-deficient melanoma cells displayed altered spheroid formation capacities

in low-attachment conditions in the absence of zebrafish media, compared to *nectin1*-wild-type cells (Extended Data Fig. 3f). These observations suggest a link between cell adhesion and environmental factors in *nectin1*-deficient melanoma cells.

### NECTIN1 inactivation induces human melanoma cell migration

We inactivated *NECTIN1* by short hairpin RNA (shRNA) or CRISPR in the A375 human melanoma cell line (Extended Data Fig. 3g–j and Methods). Consistent with our observations in zebrafish cells, *NECTIN1*-deficient cells appeared more spread out than their wild-type counterparts in low-attachment conditions in the absence, but not in the presence of serum, which translated into the formation of significantly less circular colonies, suggesting either weaker cell–cell interactions or increased cell adhesion to culture surfaces (Fig. 3a,b and Extended Data Fig. 4a–d). This indicated that serum withdrawal elicited a change in melanoma cell adhesion. In the absence of serum, *NECTIN1*-deficient colonies were significantly more dispersed in collagen-rich matrix, implying that *NECTIN1* loss increases the spreading capacity of melanoma cells in a 3D environment (Fig. 3c,d). *NECTIN1*-deficient cells migrated significantly more than their wild-type counterparts in a modified transwell assay where cells were plated without serum for 12 h before the establishment of the chemotactic gradient (Fig. 3e and Extended Data Fig. 4e). Supporting a critical role of serum starvation in this phenotype, the difference in migration between *NECTIN1*-deficient and proficient A375 cells was enhanced by longer starvation (Fig. 3e). Inducible expression of a *NECTIN1* construct mutated in the shRNA target sequence demonstrated partial rescue of this migration phenotype (Extended Data Figs. 3g and 4f). By adding Matrigel to transwells, similar differences were measured between *NECTIN1*-wild-type and deficient cells, indicating that NECTIN1 loss increases both 2D migration and 3D invasion of melanoma cells (Fig. 3f). To evaluate whether the effects of *NECTIN1* inactivation were specific to A375 cells, we downregulated *NECTIN1* in five additional human melanoma cell lines spanning the spectrum of melanocytic differentiation (from AXL^high/MITF^low to AXL^low/MITF^high) and belonging to the major phenotypic subtypes of melanoma: undifferentiated, neural crest-like and melanocytic[26] (Extended Data Fig. 4g). These lines exhibited various levels of NECTIN1 expression and different basal migratory capacities (Extended Data Fig. 4h,i). *NECTIN1* inactivation significantly increased melanoma cell migration following serum withdrawal in all cell lines (Fig. 3g and Extended Data Fig. 4i), indicating that the function of NECTIN1 does not depend on melanoma differentiation state or basal migratory capacity.

### Cell migration is due to failure to form adherens junctions

We next examined cell–cell contacts between melanoma cells. Immunofluorescence in A375 cells with various cell–cell junction markers revealed that serum depletion triggers the robust formation of adherens junctions between *NECTIN1*-wild-type cells, as assessed by α-catenin or N-cadherin staining (Fig. 4a and Extended Data Fig. 5a,b). In contrast, *NECTIN1*-deficient cells failed to form adherens junctions, in accordance with the previously reported requirement of NECTIN1 for their establishment[27] (Fig. 4a,b and Extended Data Fig. 5b). Failure to form adherens junctions was associated with increased stress fiber formation, cytoskeleton reorganization and changes in cell shape, as evidenced by F-actin and α-actinin staining and consistent with the migratory behavior of these cells (Fig. 4a,c and Extended Data Fig. 5b,c). We hypothesized that the inability of *NECTIN1*-deficient cells to establish strong cell–cell contacts under stress conditions might cause their migration. Indeed, limiting cell–cell interactions between *NECTIN1*-wild-type melanoma cells in transwell assays, either by seeding them at low density or by subjecting them to the chemotactic gradient before they could form a cell layer, increased their migratory capacity to a level similar to that of *NECTIN1*-knockout cells (Fig. 4d and Extended Data Fig. 5d), implying that the formation of

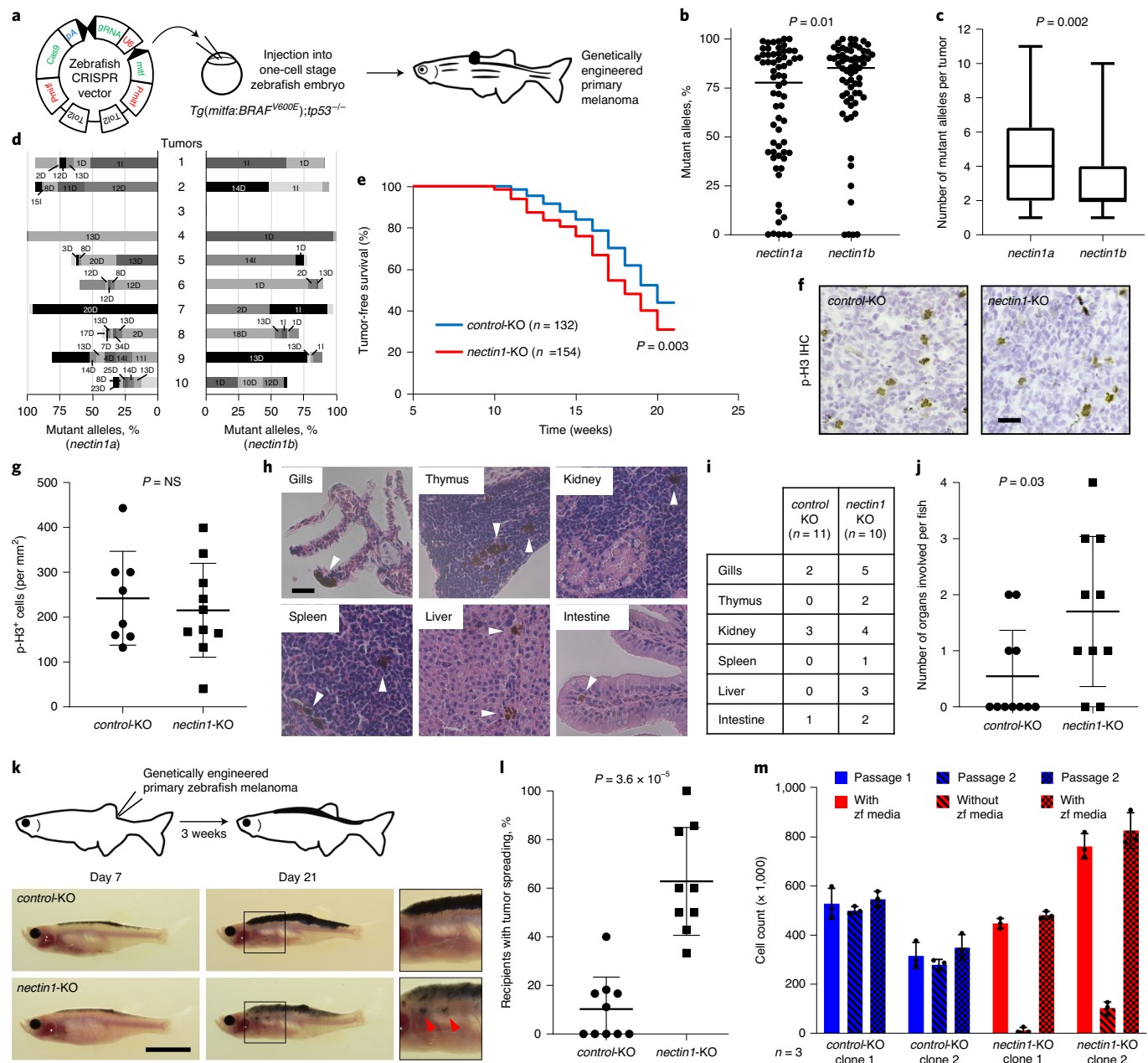

**Fig. 2 | NECTIN1 loss promotes melanoma spreading in vivo and alters cell adhesion upon serum depletion ex vivo. a**, Schematic representation of the generation of genetically engineered primary melanomas in adult zebrafish (Methods). **b**, Proportion of *nectin1a* and *nectin1b* mutant alleles in 65 primary zebrafish melanomas expressing a CRISPR vector targeting *nectin1a* and *nectin1b*. Bar represents median; paired two-tailed *t*-test. **c**, Box-and-whisker plot representing the number of *nectin1a* and *nectin1b* mutant alleles per primary zebrafish melanoma described in panel b (min-max, 1st and 4th quartiles, median, *n* = 65; paired two-tailed *t*-test). **d**, Type and proportion of *nectin1a* and *nectin1b* mutant alleles in 10 representative primary zebrafish melanomas. D, deletion, I, insertion; numbers denote affected base pairs. The last section in each bar represents all other minor alleles. **e**, Tumor-free survival curves of *Tg(mitf:BRAF^V600E);tp53^−/−* zebrafish injected with vectors targeting either *nectin1* or a control gene (pooled data of three independent experiments; log-rank test). **f**, Representative images of sections of *control-* or *nectin1*-knockout (KO) primary zebrafish melanomas, stained for phospho-histone3 (p-H3) by immunohistochemistry (IHC). Scale bar, 20 μm. **g**, Quantification of overall p-H3

staining intensity in 8 *nectin1*-wildtype and 10 *nectin1*-knockout primary zebrafish melanomas (mean ± s.d.; two-tailed *t*-test). NS, not significant. **h**, Pictures of organs presenting with disseminated tumor cells (arrowheads) in hematoxylin/eosin-stained sections of 16-week-old zebrafish bearing *nectin1*-knockout melanoma. Scale bar, 20 μm. **i**, Number of 16-week-old zebrafish with disseminated melanoma cells in the indicated organs. **j**, Quantification of organ involvement in 16-week-old zebrafish bearing *control* (*n* = 11) or *nectin1*-knockout (*n* = 10) melanoma (mean ± s.d.). Two-tailed t-test. **k**, Schematic representation of the melanoma spreading assay in adult zebrafish (top) and representative images of *casper* zebrafish 7 or 21 days after transplantation with *control-* or *nectin1*-knockout melanoma cells (bottom). Insets show ×2 magnification views. Arrows indicate patches of disseminated melanoma cells. Scale bar, 1 cm. **l**, Quantification of the proportion of secondary recipients of *control-* (*n* = 10) or *nectin1*-knockout (*n* = 9) tumors showing tumor spreading (mean ± s.d.; two-tailed *t*-test). **m**, Cell count of *control-* and *nectin1*-knockout primary zebrafish melanoma lines in the presence or absence of zebrafish (zf) culture media. Data are presented as mean ± s.d. of three independent experiments.

adherens junctions inhibits migration. To evaluate the potential contribution of cell proliferation, we measured the growth of melanoma cells upon NECTIN1 manipulation. *NECTIN1*-deficient cells displayed

a lower proliferation rate in 2D culture (Fig. 4e and Extended Data Fig. 5e), as well as reduced colony formation capacity in semi-solid media (Fig. 4f,g and Extended Data Fig. 5f) compared to *NECTIN1*-wild-type

cells, consistent with a previously noted balance between proliferation and migration in melanoma[28]. Together, these results show that *NECTIN1* loss prevents adherens junction formation between melanoma cells and promotes their migration specifically under serum depletion.

**NECTIN1 loss activates integrin-dependent matrix adhesion**

To define the changes in cell state associated with *NECTIN1* inactivation, we compared the transcriptional profiles of *NECTIN1*-knockout and *NECTIN1*-wild-type A375 cells after serum depletion. Gene Ontology analysis of the 156 differentially regulated genes revealed the prevalence of biological process terms related to cell adhesion and migration (Extended Data Fig. 6a). A more detailed gene set enrichment analysis of the same comparison uncovered specific adhesion signatures including integrin αvβ3 and focal adhesion pathways (Extended Data Fig. 6b,c and Supplementary Table 1). That gene expression mirrors the cellular phenotypes described above suggests that the migration of *NECTIN1*-deficient melanoma cells and the formation of adherens junctions between *NECTIN1*-wild-type melanoma cells correspond to specific transcriptional programs.

We examined the cell-surface makeup of *NECTIN1*-deficient and proficient A375 cells upon serum depletion by purifying cell-surface proteins and analyzing them by quantitative mass spectrometry. Integrins (in particular integrins α6β4 and α5β1) dominated the most significantly enriched proteins in *NECTIN1*-deficient cells, whereas N-cadherin (CDH2), the main cadherin expressed in A375 cells, was among the most downregulated proteins at the cell surface, consistent with the absence of adherens junctions (Fig. 5a). Accordingly, adhesion of A375 cells to several extracellular matrix (ECM) proteins, including fibronectin, laminin and, to a lesser extent, vitronectin, but not collagen-I, was significantly increased in the absence of NECTIN1 (Fig. 5b–e). Laminin is the major ligand of integrin α6β4, whereas vitronectin binds to integrins αvβ3 and αvβ5, and fibronectin to integrins α5β1 and αvβ3, among others[29]. We systematically downregulated each beta integrin by short interfering RNA (siRNA) in *NECTIN1*-deficient cells and measured the impact of these genetic manipulations on cell migration in the modified transwell assay (Fig. 5f). Knocking down integrins β3, β4 and β5 substantially reduced the migration of *NECTIN1*-deficient cells, whereas downregulating integrins β1 and β2 had a milder effect (Fig. 5g). Moreover, an antibody blocking integrin α6β4, as well as small-molecule inhibitors of integrins αvβ3 and αvβ5, significantly decreased the migration of A375 cells lacking *NECTIN1* (Fig. 5h,i), directly implicating these integrins in the motility of adherens junction-incompetent melanoma cells. These data indicate that *NECTIN1*-deficient melanoma cells switch to an integrin-dependent cell–matrix adhesion program in response to serum deprivation.

Similar migration differences were observed between *NECTIN1*-deficient and proficient cells in the presence or absence of different ECM proteins (Fig. 3e,f and Extended Data Fig. 6d), suggesting

that, in transwell assays, matrix is dispensable for the migration of melanoma cells lacking NECTIN1. This apparent paradox could be explained by the synthesis of several ECM proteins by melanoma cells themselves. Indeed, RNA-sequencing data indicated that A375 human melanoma cells express high levels of fibronectin (*FN1*) and several laminin genes (Extended Data Fig. 6e). We also observed strong fibronectin and laminin staining of these cells cultured in transwell conditions by immunofluorescence (Extended Data Fig. 6f). Using antibodies previously shown to recognize zebrafish laminin and fibronectin, we further detected extensive laminin and fibronectin matrices around melanoma cells in the transplantation area of recipient fish, suggesting that these matrices could also play a role in the in vivo spreading assay (Extended Data Fig. 6g,h). Our functional data thus support the idea that, to migrate and disseminate, *NECTIN1*-deficient cells at least in part rely on interactions with the ECM that they may have deposited themselves.

**NECTIN1-deficient cells migrate via the FAK/SRC pathway**

We next measured the phosphorylation level of a panel of kinases upon serum depletion using a phospho-kinase array (Extended Data Fig. 7a). Focal adhesion kinase (FAK, also known as PTK2) was the most differentially activated kinase between *NECTIN1*-deficient and proficient A375 melanoma cells (Extended Data Fig. 7b). To identify the pathway mediating migration of *NECTIN1*-deficient cells under serum starvation, we performed a small-scale chemical screen with inhibitors of migration-related pathways using the modified transwell assay. The only molecules that notably suppressed the migration of *NECTIN1*-deficient cells were FAK and SRC inhibitors (Fig. 6a and Extended Data Fig. 7c). These findings were validated using a second set of inhibitors (Extended Data Fig. 6d), as well as genetic downregulation of *FAK* and *SRC* by siRNAs (Extended Data Fig. 7e,f). The repressive effect of FAK and SRC inhibitors was also seen on the invasion of *NECTIN1*-deficient cells in semi-solid media (Extended Data Fig. 7g). FAK blockade suppressed the migration of four out of five additional human melanoma cell lines upon *NECTIN1* downregulation (Fig. 6b), demonstrating that most melanomas lacking NECTIN1 depend on FAK signaling to migrate. Accordingly, we verified the differential activation of FAK and SRC by phosphorylation between *NECTIN1*-deficient and proficient A375 cells by Western Blotting (Fig. 6c). Using pharmacological inhibition of integrins αvβ3 and αvβ5, FAK and SRC, we also confirmed the sequential activation of the three members of the pathway in A375 human melanoma cells (Extended Data Fig. 7h). In line with these observations, *nectin1*-knockout primary zebrafish melanoma cells exhibited much higher levels of phospho-fak than *nectin1*-wild-type cells in the zebrafish tumor spreading assay, as evidenced by immunohistochemistry (Fig. 6d,e and Extended Data Fig. 8a,b). 36% (10/28) of transplanted tumors displayed a staining pattern in the shape of a gradient, with stronger signal towards the invasive edge than directly under the skin (Extended Data Fig. 8c). We targeted

**Fig. 3 | *NECTIN1* inactivation induces cell migration and invasion upon serum depletion in human melanoma cell lines. a,** Images of the structures formed by 10,000 A375 human melanoma cells stably expressing a control shRNA (shCTRL) or an shRNA directed against *NECTIN1* (shNECTIN1) after 8 days in low-attachment conditions in the presence or absence of serum (FBS). Scale bar, 200 μm. Data are representative of nine independent experiments. **b,** Circularity of the colonies formed by A375 human melanoma cells in low-attachment conditions as shown in panel a (Methods). Data are presented as mean ± s.d. of nine independent colonies per condition (paired two-tailed *t*-test). **c,** Images of the colonies formed by 1,000 of the cells described in panel a after 7 days in collagen-rich matrix in the absence of serum (FBS) (top). Scale bar, 50 μm. Data are representative of three independent experiments. Analysis of the images shown on top using ImageJ (bottom). **d,** Spread of the colonies formed by A375 human melanoma cells in collagen-rich matrix as shown in panel c (Methods).

Data are presented as mean ± s.d. (*n* = 41 (shCTRL) and *n* = 45 (shNECTIN1)) of three independent experiments (two-tailed *t*-test). a.u., arbitrary units. **e,** Migration of A375 human melanoma cells stably expressing an shRNA directed against *NECTIN1* (shNECTIN1) relative to cells expressing a control shRNA (shCTRL) in a transwell assay after 12 or 24 h of serum starvation. Data are presented as mean ± s.d. of four independent experiments. Cells were allowed to migrate for 6 h (paired two-tailed *t*-test). **f,** Invasion through Matrigel of A375 human melanoma cells after 12 h of serum starvation as described in panel e. Cells were allowed to migrate for 10 h. Data are presented as mean ± s.d. of four independent experiments (paired two-tailed *t*-test). **g,** Migration of six human melanoma cell lines after 12 h of serum starvation as described in panel e. Cells were allowed to migrate for different times depending on the cell line (Extended Data Fig. 4i and Methods). Data are presented as mean ± s.d. of four independent experiments (paired two-tailed *t*-test).

the two zebrafish orthologs of FAK, *fak1a* and *fak1b* (hereafter collectively referred to as *fak*), in addition to *nectin1* in zebrafish melanocytes to generate *fak/nectin1*-knockout melanomas in vivo (Extended Data Fig. 8d and Supplementary Note). *Fak* inactivation did not affect melanoma onset in zebrafish (Extended Data Fig. 8e) or change proliferation rates in primary tumors, as assessed by phospho-Histone3

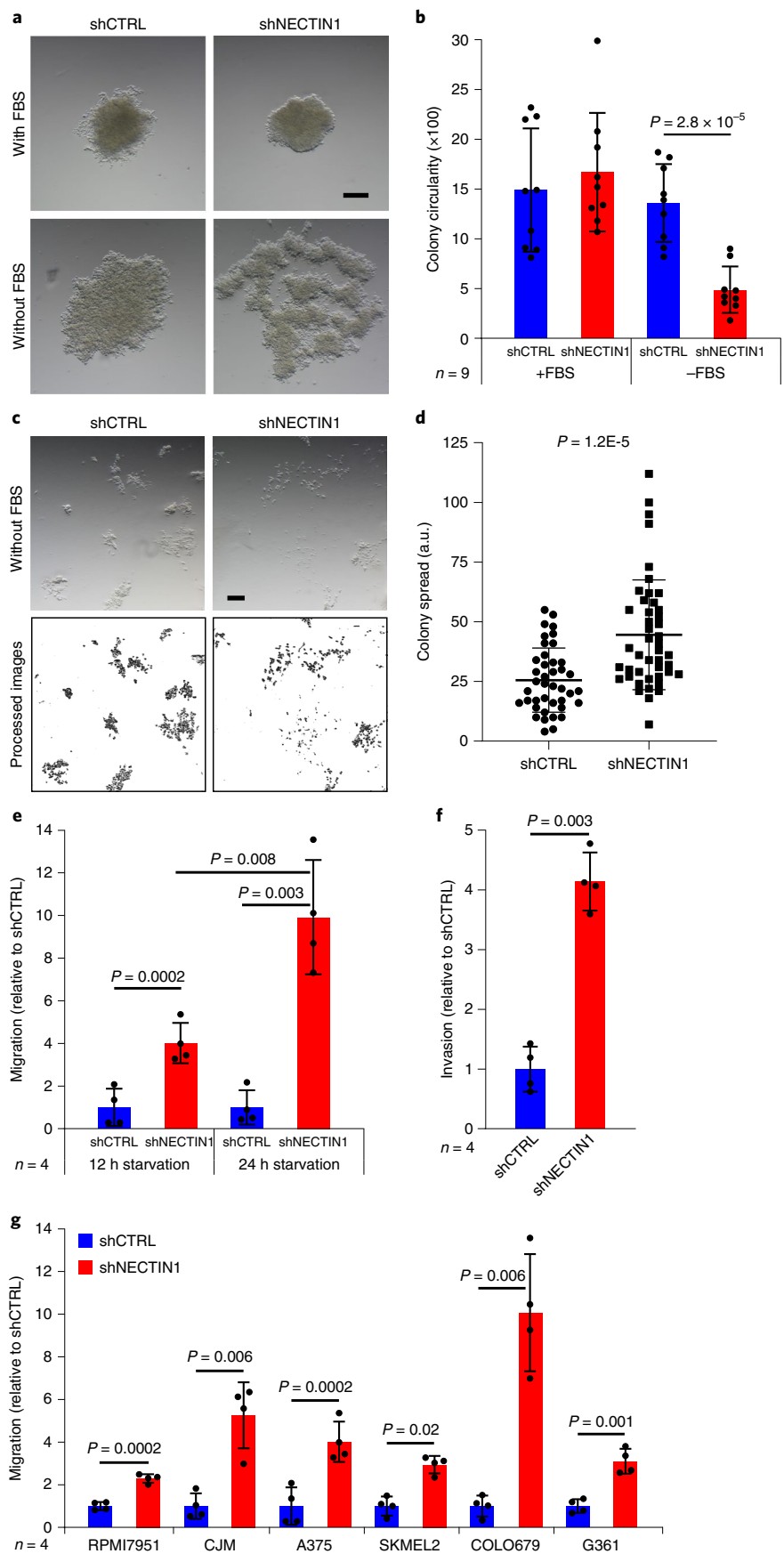

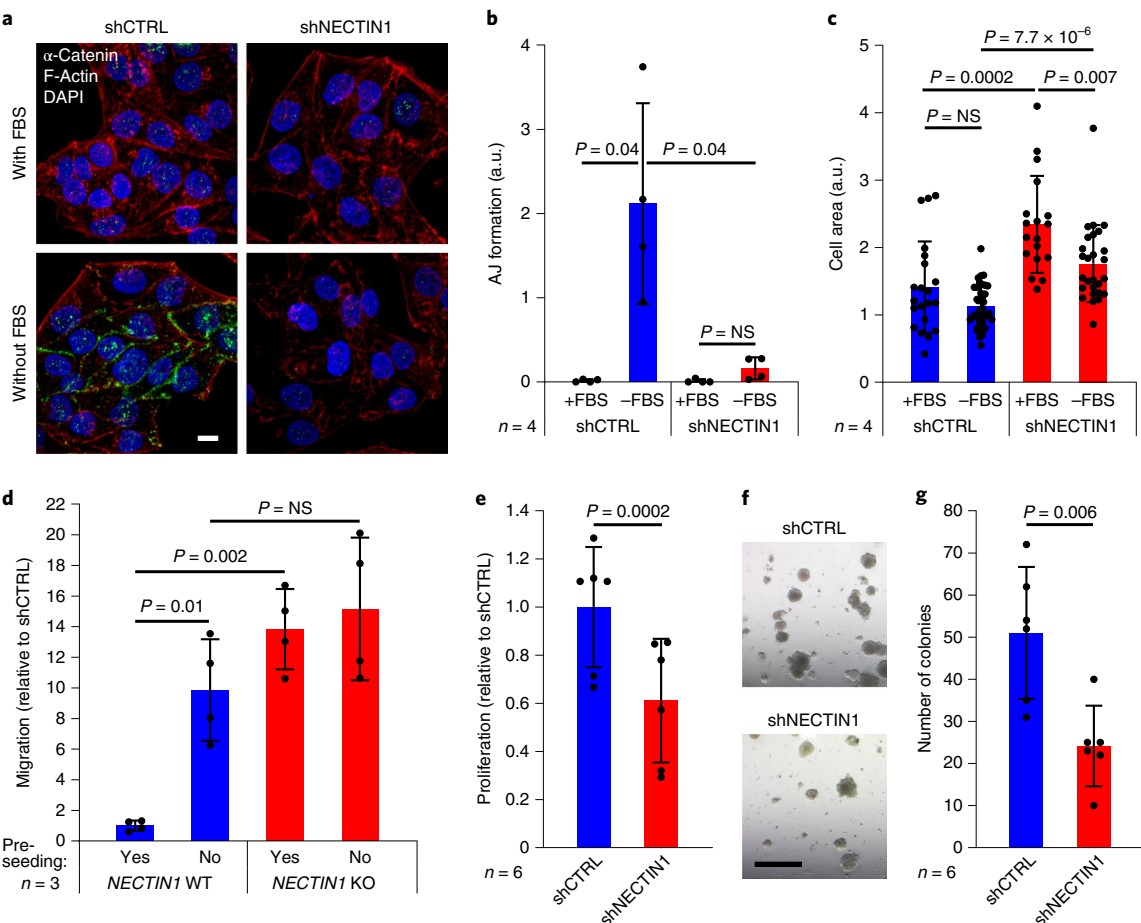

**Fig. 4 | Cell migration induced by *NECTIN1* loss is due to failure to form adherens junctions. a**, Immunofluorescence analysis of α-catenin (green) and F-Actin (red) in A375 human melanoma cells stably expressing a control shRNA (shCTRL) or an shRNA directed against *NECTIN1* (shNECTIN1) cultured in the presence or absence of serum (FBS) for 12 h. Scale bar, 10 μm. Representative images of five independent experiments are shown. DAPI, 4,6-diamidino-2-phenylindole (blue), **b**, Quantification of adherens junction (AJ) formation as measured by α-catenin signal intensity on immunofluorescence images as shown in panel a. Data represent mean ± s.d. (*n* = 4 fields per condition; paired two-tailed *t*-test). **c**, Quantification of cell surface area based on immunofluorescence images as shown in panel **a**. Data represent mean ± s.d. (*n* = 21, 38, 18, 27 cells for shCTRL+FBS, shCTRL-FBS, shNECTIN1+FBS, and shNECTIN1-FBS, respectively;

paired two-tailed *t*-test). **d**, Migration of three *NECTIN1*-knockout (KO) A375 human melanoma cell lines relative to three *NECTIN1*-wild-type (WT) A375 human melanoma cell lines in a transwell assay after 12 h of serum starvation. Cells were either allowed to settle for 12 h (pre-seeding) or not before being put in the presence of the chemotactic gradient. Cells were allowed to migrate for 6 h. Data represent mean ± s.d. of three independent experiments (paired two-tailed *t*-test). **e**, Proliferation of the cells described in panel a. Data represent mean ± s.d. of six independent experiments (paired two-tailed *t*-test). **f**, Representative images of colonies formed in Matrigel by the cells described in panel a after 14 days in culture. Scale bar, 1 mm. **g**, Number of colonies quantified from the experiments described in panel f. Data represent mean ± s.d. of six independent experiments (paired two-tailed *t*-test).

immunohistochemistry (Extended Data Fig. 8f). However, examination of hematoxylin/eosin-stained tissue sections revealed a lower organ involvement upon *fak* knockout in adult zebrafish bearing *nectin1*-knockout primary melanomas (Fig. 6f,g), supporting a critical role for FAK in distant seeding by *NECTIN1*-deficient melanoma cells. Genetic inactivation of *fak* in primary zebrafish melanomas lacking *nectin1* significantly reduced cell dissemination in the zebrafish tumor spreading assay (Fig. 6h, i), confirming the importance of FAK signaling. Surprisingly, western blot analysis of NECTIN1 and phospho-FAK revealed a positive correlation between FAK activity and NECTIN1 levels in our panel of 6 human melanoma cell lines (Extended Data Fig. 8g,h). However, in three out of four cell lines with relatively high NECTIN1 expression, pharmacological inhibition of FAK induced a marked decrease in NECTIN1 levels (Extended Data Fig. 8i), suggesting feedback regulation of NECTIN1 expression by FAK activity. Note that this regulatory loop is likely hindered in cases of *NECTIN1* genetic inactivation. Our results show that upon serum deprivation, *NECTIN1*-deficient cells activate an integrin/FAK/SRC axis fostering cell motility and spreading.

## A drop in IGF1 elicits NECTIN1 loss-of-function phenotypes

We sought to identify the serum components responsible for the difference in cell behavior between NECTIN1-proficient and deficient melanoma cells. Culture media supplemented with dextran-treated charcoal-stripped serum, which contains nutrients but is depleted in hormones and growth factors, triggered adherens junction formation between *NECTIN1*-wild-type, but not *NECTIN1*-deficient, A375 human melanoma cells to the same extent as serum-free media, as assessed by immunofluorescence (Extended Data Fig. 9a), pointing to the lack of growth factors or hormones as the likely cause of this difference. Similar to serum withdrawal, dextran-treated charcoal-stripped serum also activated FAK/SRC signaling in cells lacking NECTIN1 (Extended Data Fig. 9b). We examined the transcriptional response of A375 cells to serum depletion by RNA sequencing. This response was dominated by proliferation arrest and responses to external stimuli, as indicated by Gene Ontology analysis (Extended Data Fig. 9c and Supplementary Table 2). Gene set enrichment analysis revealed a number of growth factor and hormone signaling pathways underlying this response to

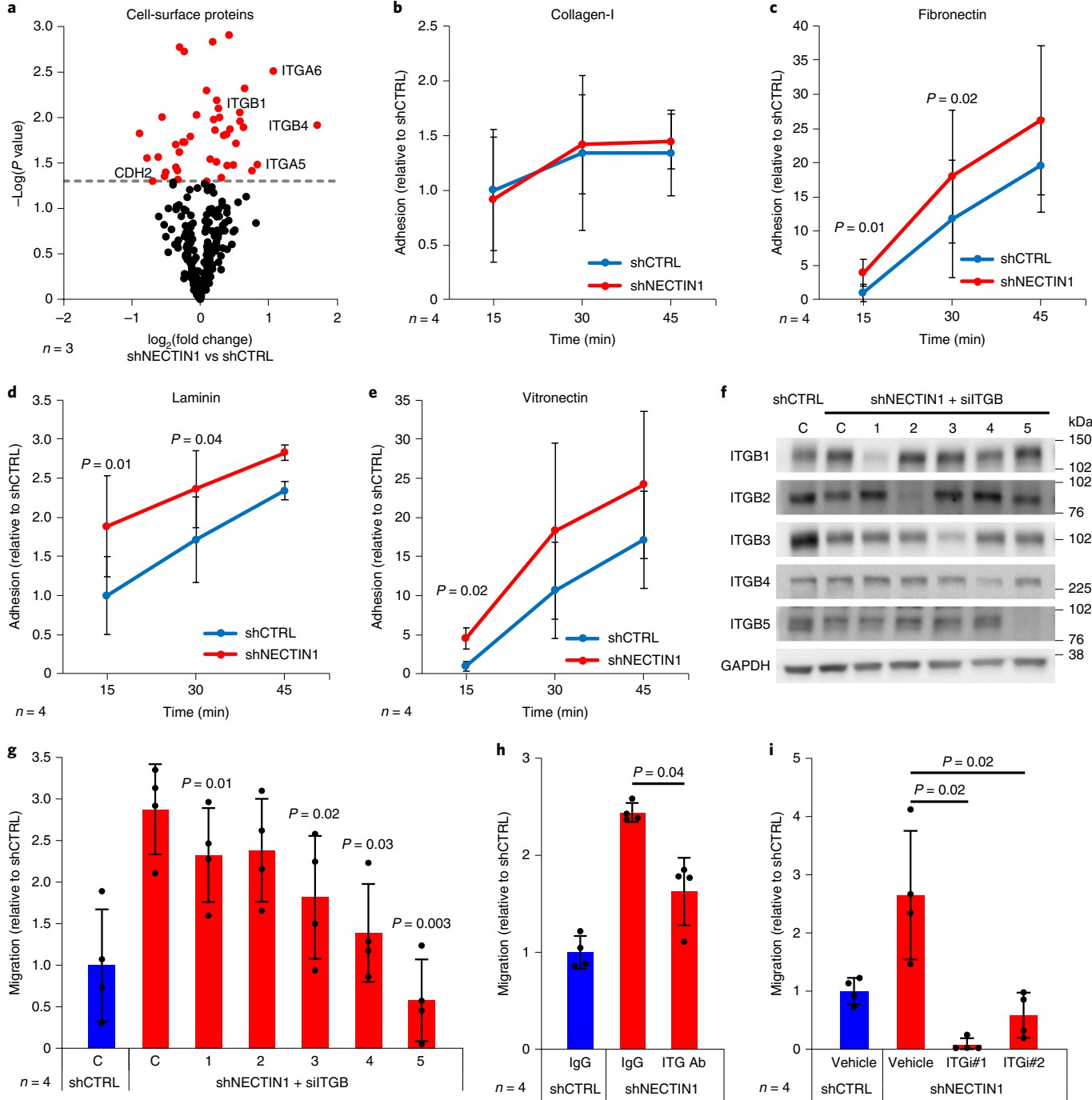

**Fig. 5 | *NECTIN1*-deficient cells activate an integrin-dependent cell-to-matrix adhesion program. a,** Volcano plot representing the significance and relative abundance of cell-surface proteins in A375 human melanoma cells in the presence or absence of NECTIN1. A *P* value (two-tailed *t*-test) threshold of 0.05 was chosen (red: *P* < 0.05, black: *P* > 0.05). Some of the most differentially expressed proteins are indicated. **b–e,** Adhesion of A375 human melanoma cells stably expressing an shRNA directed against *NECTIN1* (shNECTIN1) relative to cells expressing a control shRNA (shCTRL) to collagen-I-, fibronectin-, laminin- or vitronectin-coated surfaces at various timepoints after seeding. Data represent mean ± s.d. of four independent experiments (paired two-tailed *t*-test). **f,** Western blot analysis of ITGB levels in the cells described in panel b transfected with a control siRNA (C) or siRNAs targeting ITGB1 (1), ITGB2 (2), ITGB3 (3), ITGB4 (4) or ITGB5 (5). Data are representative of four independent experiments. **g,** Migration of the cells described in panel f relative to cells stably expressing a control shRNA (shCTRL) and transfected with a control siRNA in a transwell assay after 12 h of

serum starvation. Cells were allowed to migrate for 6 h. Data represent mean ± s.d. of four independent experiments (paired two-tailed *t*-test; *P* values are shown for the comparisons siITGB versus siCTRL (shNECTIN1)). **h,** Migration of A375 human melanoma cells stably expressing an shRNA directed against *NECTIN1* (shNECTIN1) relative to cells expressing a control shRNA (shCTRL) in a transwell assay upon treatment with an integrin α6β4 blocking antibody (ITG Ab, GoH3, 40 µg ml⁻¹). Cells were allowed to migrate for 6 h. Data represent mean ± s.d. of four independent experiments (paired two-tailed *t*-test). IgG, immunoglobulin G. **i,** Migration of A375 human melanoma cells stably expressing an shRNA directed against *NECTIN1* (shNECTIN1) relative to cells expressing a control shRNA (shCTRL) in a transwell assay upon treatment with two integrin αvβ3 and αvβ5 inhibitors (ITGi#1: SB273005; ITGi#2: echistatin). Cells were allowed to migrate for 6 h. Data represent mean ± s.d. of four independent experiments (paired two-tailed *t*-test).

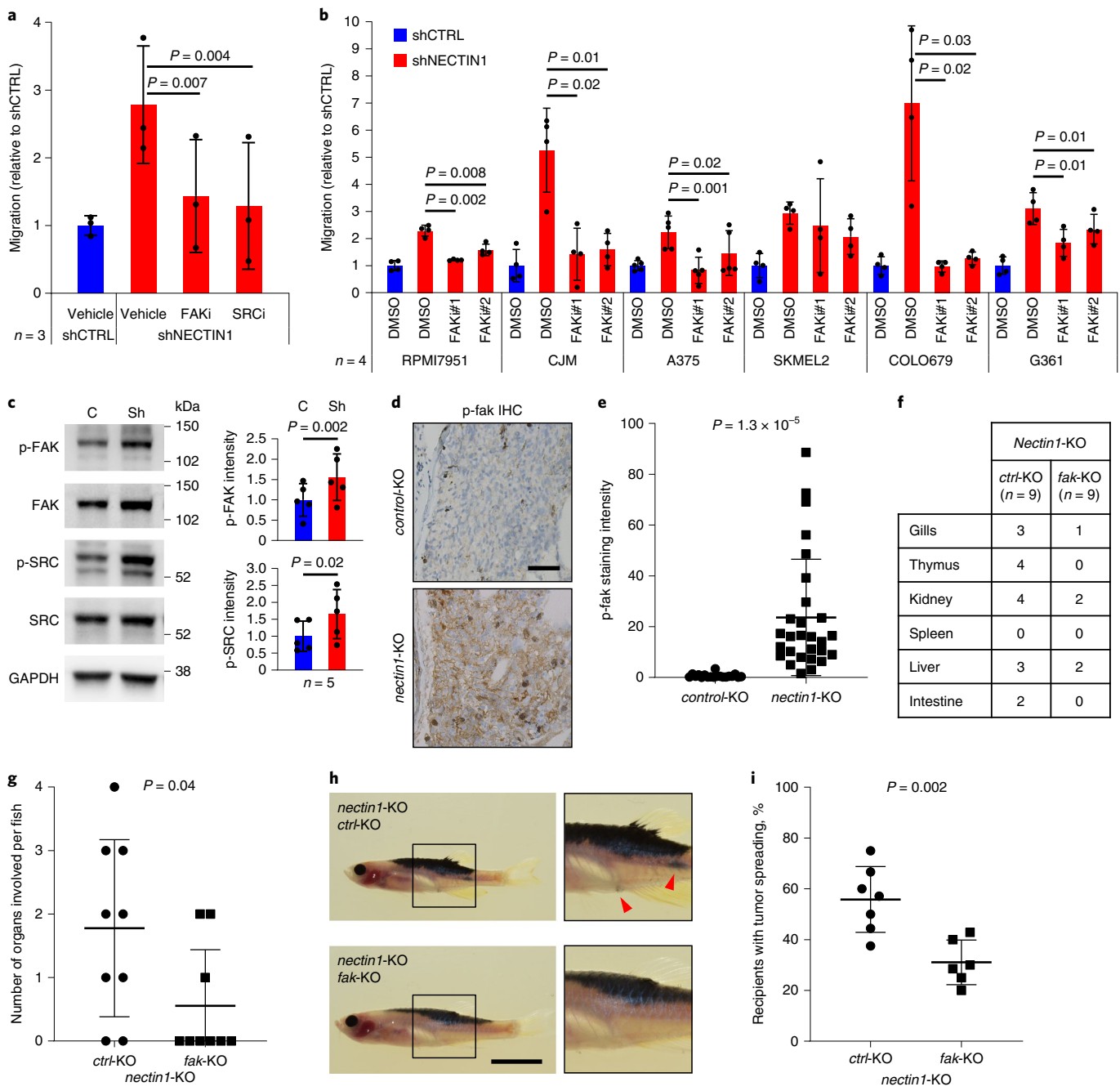

**Fig. 6 | Activation of a FAK/SRC pathway is central to *NECTIN1*-deficient cell migration. a**, Migration of A375 human melanoma cells stably expressing an shRNA directed against *NECTIN1* (shNECTIN1) relative to cells expressing a control shRNA (shCTRL) in a transwell assay upon treatment with a FAK inhibitor (FAKi, PF562271) or a SRC inhibitor (SRCi, dasatinib). Cells were allowed to migrate for 6 h. Data represent mean ± s.d. of three independent experiments. Paired two-tailed *t*-test. **b**, Migration of six human melanoma cell lines stably expressing an shRNA directed against *NECTIN1* (shNECTIN1) relative to cells expressing a control shRNA (shCTRL) in a transwell assay upon treatment with two independent FAK inhibitors (FAKi#1, PF562271; FAKi#2, GSK2256098). Cells were allowed to migrate for different times depending on the cell line (Extended Data Fig. 4i and Methods). Data represent mean ± s.d. of four independent experiments (paired two-tailed *t*-test). **c**, Western blot analysis of FAK and SRC phosphorylation levels in the cells described in panel a in the absence of serum. C, shCTRL; Sh, shNECTIN1. Representative example of five independent experiments. Signal intensity was normalized to GAPDH and presented as a ratio to the vehicle-treated shCTRL condition. Data represent mean ± s.d. (paired two-tailed *t*-test). **d**, Representative images of sections of adult *casper* recipients transplanted with *control*- or *nectin1*-knockout primary zebrafish melanomas, stained for phospho-fak by immunohistochemistry. Scale bar, 50 μm. **e**, Quantification of phospho-fak (p-fak) staining intensity in *casper* recipients transplanted with *nectin1*-wild-type ($n = 19$) or *nectin1*-knockout ($n = 28$) zebrafish melanomas. Data represent mean ± s.d. (two-tailed *t*-test). **f**, Number of 16-week-old zebrafish with disseminated melanoma cells in the indicated organs. **g**, Quantification of organ involvement in 16-week-old zebrafish bearing *nectin1/ctrl*-knockout ($n = 9$) or *nectin1/fak*-knockout ($n = 9$) melanoma. Data represent mean ± s.d. (two-tailed *t*-test). **h**, Representative images of adult *casper* zebrafish transplanted with *nectin1*-knockout primary zebrafish melanoma cells expressing a control gRNA (*ctrl*-knockout) or a gRNA targeting *fak* (*fak*-knockout). Insets show ×2 magnification views. Arrowheads indicate patches of disseminated melanoma cells. Scale bar, 1 cm. **i**, Quantification of the proportion of secondary recipients of *nectin1/ctrl*-knockout (*ctrl*-KO, $n = 7$) or *nectin1/fak*-knockout (*fak*-KO, $n = 6$) tumors showing tumor spreading. Data represent mean ± s.d. (two-tailed *t*-test).

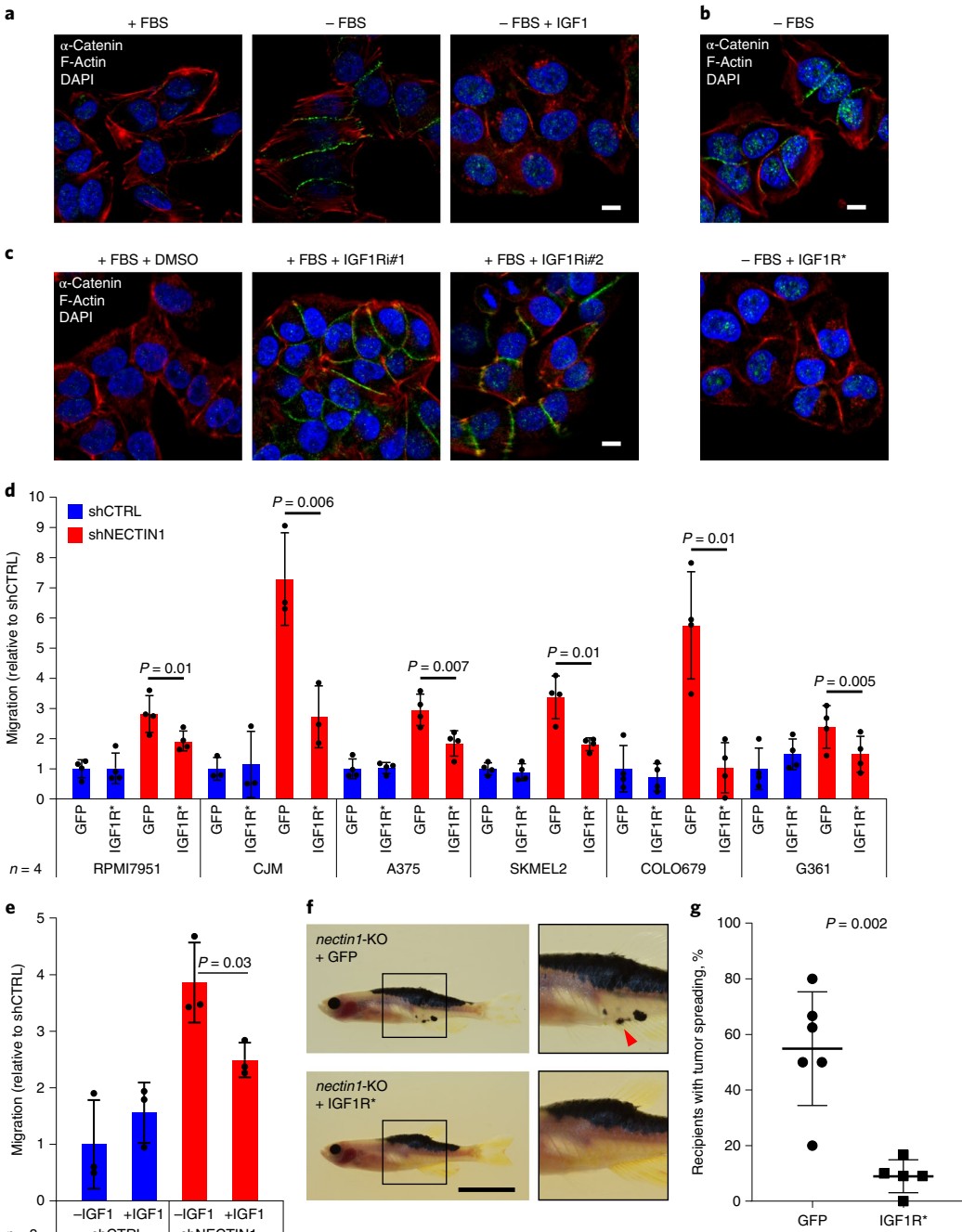

**Fig. 7 | A decrease in IGF1 signaling triggers the formation of adherens junction between *NECTIN1*-wild-type cells or the migration and dissemination of *NECTIN1*-deficient cells. a**, Immunofluorescence analysis of α-catenin (green) and F-actin (red) in A375 human melanoma cells in the presence or absence of serum (FBS) (blue, DAPI), complemented or not with 100 ng ml⁻¹ IGF1 for 12 h. Scale bar, 10 μm. Data are representative of four independent experiments. **b**, Immunofluorescence analysis of α-catenin (green) and F-actin (red) in A375 human melanoma cells expressing or not a constitutively active form of the IGF1 receptor (CD8-IGF1R, IGF1R*), and cultured in the absence of serum (FBS) for 12 h (blue, DAPI). Scale bar, 10 μm. Data are representative of three independent experiments. **c**, Immunofluorescence analysis of α-catenin (green) and F-actin (red) in A375 human melanoma cells cultured in the presence of serum (FBS) and treated with DMSO (vehicle) or with two different IGF1R inhibitors (IGF1Ri#1: Linsitinib, 1 μM; IGF1Ri#2: GSK1838705A, 1 μM) for 12 h (blue, DAPI). Scale bar, 10 μm. Data are representative of four independent experiments. **d**, Migration of six human melanoma cell lines conditionally expressing GFP or a constitutively active form of IGF1R (IGF1R*),

and stably expressing an shRNA directed against *NECTIN1* (shNECTIN1) relative to cells expressing a control shRNA (shCTRL) in a transwell assay. Cells were allowed to migrate for different times depending on the cell line (Extended Data Fig. 4i and Methods). Data represent mean ± s.d. of four independent experiments (paired two-tailed *t*-test). **e**, Migration of A375 human melanoma cells stably expressing either a control shRNA (shCTRL) or an shRNA directed against *NECTIN1* (shNECTIN1) in a transwell assay after 12 h of serum starvation in the presence or absence of 100 ng ml⁻¹ IGF1. Cells were allowed to migrate for 6 h. Data represent mean ± s.d. of three independent experiments (paired two-tailed *t*-test). **f**, Representative images of adult *casper* zebrafish transplanted with *nectin1*-knockout primary zebrafish melanoma cells expressing GFP or a constitutively active form of IGF1R (CD8-IGF1R, IGF1R*), 21 days after injection. Insets show ×2 magnification views. Arrowhead indicates patches of disseminated melanoma cells. Scale bar, 1 cm. **g**, Quantification of the proportion of secondary recipients of *nectin1*-knockout tumors expressing GFP (*n* = 6) or *nectin1*-knockout tumors expressing CD8-IGF1R (IGF1R*) (*n* = 5) showing tumor spreading. Data represent mean ± s.d. (two-tailed *t*-test).

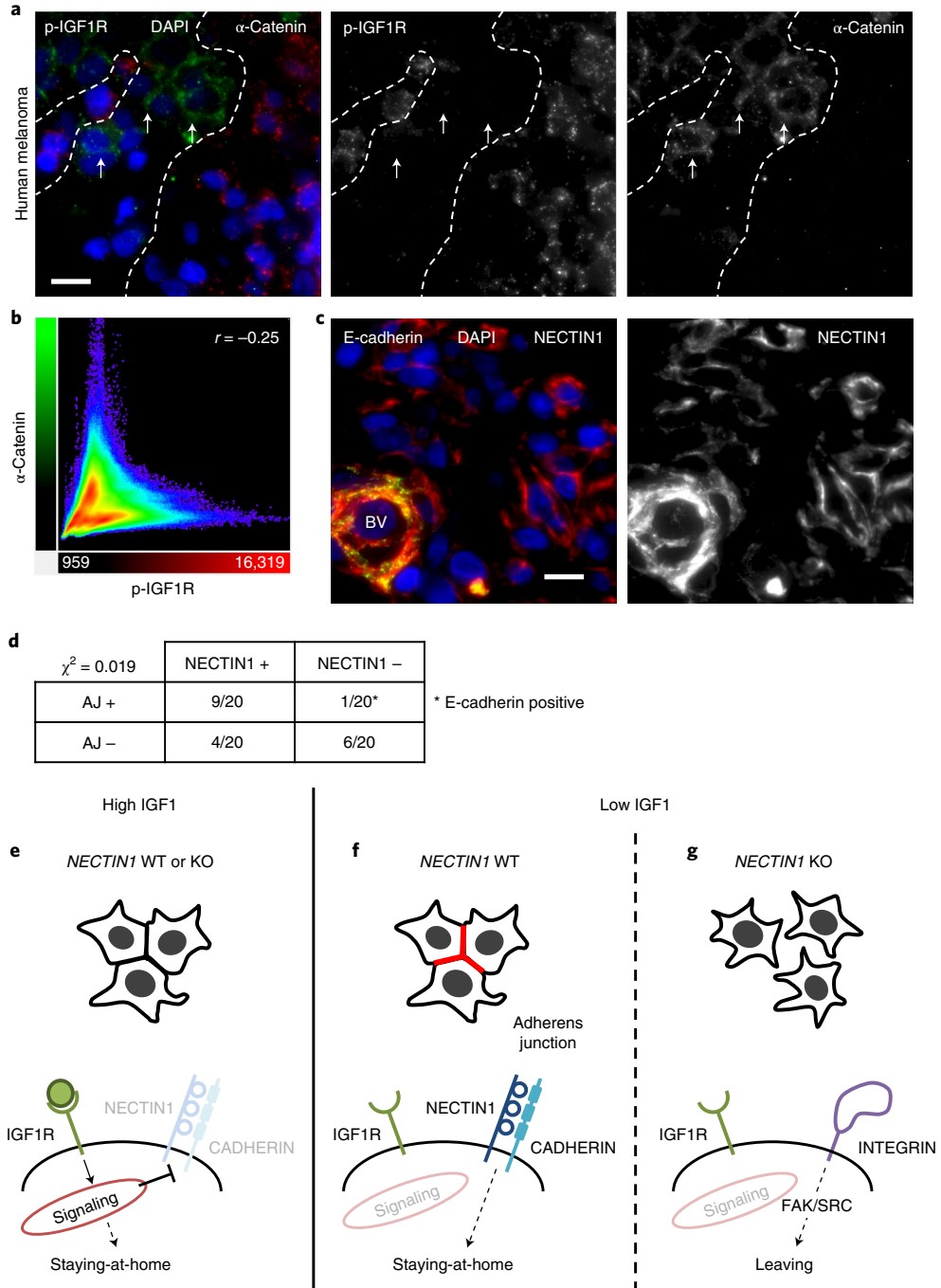

**Fig. 8 | Adherens junctions occur in areas of low IGF1 signaling and specifically in NECTIN1-positive melanomas in human.**
**a**, Immunofluorescence analysis of α-catenin (green) and phospho-IGF1R (red) on a tissue section of human melanoma (blue, DAPI). Scale bar, 10 μm. Dotted line demarcates areas of high and low IGF1 signaling. Arrows point to adherens junctions. A representative example of nine independent tumors is shown. **b**, Diagram representing signal intensities of α-catenin (green) and phospho-IGF1R (red) in the section shown in panel a (Pearson's correlation). **c**, Immunofluorescence analysis of E-cadherin (green) and NECTIN1 (red) (blue, DAPI) on a tissue section of the same tumor as in panel a. Scale bar, 10 μm. BV, blood vessel. **d**, Table showing the distribution of 20 tissue sections of human melanoma depending on the presence or absence of adherens junctions

(as assessed by α-catenin staining) and NECTIN1 by immunofluorescence. Correlation was measured by chi-squared test. Note that the only NECTIN1-negative tumor exhibiting adherens junctions was positive for E-cadherin. **e**–**g**, Model for the role of NECTIN1 in melanoma. **e**, In the presence of IGF1, melanoma cells actively repress the formation of adherens junctions and proliferate, independent of *NECTIN1* status. **f**, In the absence of IGF1, *NECTIN1*-wild-type melanoma cells form robust adherens junctions (red) and rely on cell–cell adhesion to stay in the niche. **g**, In the absence of IGF1, *NECTIN1*-deficient melanoma cells (that are unable to establish adherens junctions) switch to cell–matrix adhesion and activate an integrin/FAK/SRC axis, leading to their migration and dissemination.

external stimuli (Extended Data Fig. 9d and Supplementary Table 3), which allowed us to establish a list of 12 candidate secreted factors (Extended Data Fig. 9e) that we tested for their ability to prevent

adherens junction formation upon serum withdrawal as detected by immunofluorescence for the adherens junction marker α-catenin. Only IGF1 and, to a lesser extent, insulin scored in this assay

(Fig. 7a and Extended Data Fig. 9f). The expression of a constitutively active form of the IGF1 receptor (IGF1R*)[30] recapitulated this effect (Fig. 7b and Extended Data Fig. 9g). Conversely, small-molecule inhibitors of IGF1R triggered adherens junction formation between A375 cells even in the presence of serum (Fig. 7c), demonstrating that the absence of IGF1 signaling is both necessary and sufficient to elicit the behavioral changes observed in melanoma cells. Moreover, forced IGF1 signaling partially reversed the migratory propensity of *NECTIN1*-deficient cells in the modified transwell assay in 6 different human melanoma cell lines (Fig. 7d), similar to direct treatment with IGF1 itself (Fig. 7e). These results were confirmed in vivo by expressing IGF1R* in *nectin1*-deficient zebrafish melanomas, which accelerated tumor onset compared to a GFP-negative control (Extended Data Fig. 10a), in line with the generally protumorigenic role of aberrant IGF1 signaling, although we did not detect significant differences in tumor cell proliferation by phospho-Histone3 staining of zebrafish melanoma sections (Extended Data Fig. 10b). Analysis of tissue sections of adult zebrafish bearing *nectin1*-knockout tumors expressing either GFP or IGF1R* did not reveal any differences in organ involvement (Extended Data Fig. 10c), likely due to IGF1R*-expressing tumors forming earlier than GFP-expressing ones and having more time to invade adjacent tissues and disseminate in the fish. However, expression of IGF1R* abolished the dissemination of *nectin1*-knockout zebrafish melanoma cells upon subcutaneous transplantation into adult recipients (Fig. 7f,g). These data define a functional interaction between IGF1 signaling and adherens junctions in the regulation of melanoma cell behavior, with a decrease in IGF1 eliciting the formation of cell–cell contacts between *NECTIN1*-wild-type adherens junctions-competent cells or the migration of *NECTIN1*-deficient adherens junctions-impaired cells.

### Adherens junctions occur in areas of low IGF1 signaling

In humans, low IGF1R expression was associated with reduced progression-free survival in the TCGA cohort of cutaneous melanoma (Extended Data Fig. 10d). We therefore assessed the interaction between IGF1 signaling and adherens junctions in human melanoma in vivo. We performed immunofluorescence in tissue sections of 20 human melanoma biopsies with markers of active IGF1 signaling (phospho-IGF1R) and adherens junctions (α-catenin) and independently evaluated NECTIN1 expression. We noted heterogeneity in IGF1 signaling, some areas of tumors showing weak or no phospho-IGF1R straining whereas others exhibited strong signals, especially near blood vessels. Adherens junctions were only observed in areas that lacked IGF1 signaling (Fig. 8a and Extended Data Fig. 10e). The patterns of phospho-IGF1R and α-catenin were generally mutually exclusive as demonstrated by the anti-correlation between the fluorescence intensity of the two signals (Fig. 8b and Extended Data Fig. 10f), except in the case of endothelial cells lining blood vessels. Adherens junctions almost exclusively occurred in tumors that expressed NECTIN1 (Fig. 8c,d). The only exception was in a melanoma that stained negative for NECTIN1 but expressed E-cadherin, which likely explains the ability to form adherens junctions. These results indicate that adherens junctions are formed by melanoma cells specifically in response to a drop in IGF1 signaling and suggest that the growth factor variations that elicit changes in melanoma cell behavior happen locally in human tumors.

## Discussion

We identified *NECTIN1* as a major metastasis-suppressor gene in melanoma. The fact that *NECTIN1* downregulation by shRNA and complete *NECTIN1* ablation by CRISPR elicited similar cellular phenotypes indicates that both homozygous and hemizygous deletions of *NECTIN1*, together detected in 55% of patients with melanoma, promote metastasis and that *NECTIN1* acts as a haploinsufficient metastasis-suppressor gene. In addition to melanoma, *NECTIN1* is frequently deleted in bladder, uterine and cervical cancers in TCGA datasets. Whether NECTIN1

controls similar mechanisms in these tumor types remains to be determined.

Our data show that *NECTIN1* status determines the response of melanoma cells to environmental stress, specifically a drop in local IGF1 concentration. In the presence of high levels of IGF1, melanoma cells proliferate within the cancer cell niche regardless of *NECTIN1* status (Fig. 8e), as demonstrated by previous reports examining the role of IGF1 in early melanoma lesions[31]. In low IGF1 conditions and in the presence of NECTIN1, melanoma cells form robust cell–cell contacts through adherens junctions, which prevent them from leaving the niche (Fig. 8f). In the absence of NECTIN1, they switch to cell–matrix adhesion and migrate away by activating an integrin/FAK/SRC pathway, leading to tumor dissemination (Fig. 8g). The switch from cell–cell to cell–matrix adhesion may be explained by a previously suggested balance between NECTIN abundance and integrin activity[32]. Our observations implicating local variations in IGF1 signaling in the regulation of melanoma cell adhesion and behavior are in line with reports indicating that IGF1 signaling decreases cell–cell adhesion and promotes EMT in several epithelial tumors[33,34]. In melanoma, inhibition of IGF1 has been shown to reduce proliferation and invasion by blocking an EMT-like program[35]. That low IGF1 increases epithelial features of melanoma cells is consistent with our data. Additionally, our results provide an explanation to the apparent paradox regarding the effect of IGF1 inhibition on migration and invasion by suggesting the effect is governed by the status of cell–cell adhesion, particularly adherens junctions; in cells capable of establishing cell–cell contacts, low IGF1 signaling forces adherens junction formation and prevents migration whereas it triggers the dissemination of cells that are unable to make cell–cell junctions. By identifying a frequent chromosomal deletion that sensitizes cancer cells to local signals in the tumor microenvironment, our findings reconcile the genetic and non-cell-autonomous views of metastasis in melanoma.

The interplay between cell adhesion and external signals may prove especially relevant to non-epithelial tumors like melanoma. The loss of cell–cell contacts, especially of E-cadherin-dependent adherens junctions, occurs in the early phases of the metastatic progression of epithelial tumors[36]. Melanocytes, however, arise from the neural crest, an embryonic cell layer that delaminates from the neural tube by EMT during vertebrate development and therefore do not usually form adherens junctions[14]. Melanoma cells appear closer in their behavior to the developing neural crest than to organized epithelia[37,38]. Interestingly, *NECTIN1* inactivating mutations in humans cause cleft lip or cleft palate[39] reminiscent of the disruption of craniofacial development in neurocristopathies. Moreover, insulin signaling promotes neural crest induction in vitro[40]. The consequences of NECTIN1 inactivation appear independent of the differentiation state of melanoma cells and the induction of adherens junctions by IGF1 depletion could be observed in cells that expressed N-cadherin and not E-cadherin, a feature usually associated with a more mesenchymal state[41]. Our findings thus uncover a mode of regulation of non-epithelial cancer cell migration.

Previous reports have suggested that melanoma cells can oscillate between proliferative and migratory states, a phenomenon known as plasticity[28]. Our study characterizes genes and mechanisms of the transition to the migratory state, which we found to be initiated by local changes in growth factor availability. The mechanism described here may have evolved to allow cells to survive stress situations, either by establishing contacts with their neighbors to receive supportive signals or by moving away in search of a more favorable environment. This model suggests that melanoma cells successfully metastasize to sites that provide higher levels of IGF1 and that cells that can adapt to variable levels of IGF1 signaling might constitute more efficient metastasizers. The capacity to adapt to local stress may represent a major asset for cancer cells to successfully complete the invasion-metastasis

cascade, as previously shown with oxidative stress[42]. This plasticity of cancer cells can also impinge on their response to treatment as a more migratory state has been associated with drug resistance[43].

Our study identifies the genetic determinants of a mechanism whereby cancer cells integrate two types of external signals from their local environment–physical cues sensed through adhesion molecules and chemical cues sensed by growth factor receptors–to govern cell behaviors. This mechanism sheds light on the regulation of cellular decisions in the cancer cell niche that may prove therapeutically actionable.

## Online content

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

## Methods

The research conducted in the present study complies with all relevant ethical regulations.

### Genomic data analysis

*NECTIN1* was identified in a frequent focal deletion by GISTIC analysis of the TCGA cohort of 363 patients with melanoma[20,21]. The overall survival curves of patients presenting or not *NECTIN1* deletions were generated by stratifying the survival data of the TCGA cohort of cutaneous melanoma according to *NECTIN1* linear copy number with a threshold set at −0.25. The progression-free survival curves of patients with high or low IGF1R expression were generated by stratifying the survival data of the TCGA cohort of cutaneous melanoma according to *IGF1R* expression z-score and comparing the top and bottom quartiles.

### Human melanoma samples

This study was conducted with approval of the institutional review board of Brigham and Women's Partners Human Research Committee, Harvard Medical School. All pathology specimens were archived samples of the Department of Pathology at Brigham and Women's Hospital. All samples are deidentified (that is, personally identifiable information and identifiers were not retained and cannot be retrieved). As such, the institutional review board deemed that seeking further informed consent was not necessary.

All human melanomas were formalin fixed, paraffin embedded, sectioned and stained with hematoxylin and eosin for histopathological evaluation. The hematoxylin and eosin-stained sections were independently reviewed and confirmed by a dermatopathologist (C.G.L.). In total, 122 cases were studied retrospectively for NECTIN1 immunohistochemistry (melanocytic nevi (*n* = 20), primary melanoma (*n* = 72) and metastatic melanoma (*n* = 30)), and 20 tissue sections of primary and metastatic melanoma were used for immunofluorescence studies. In addition, two commercial human melanoma tissue-microarrays were purchased from Biomax (ME1004e and ME2082c).

### Immunohistochemistry staining

Immunohistochemistry studies were performed on paraffin-embedded sections of formalin-fixed tissue. For staining of human melanoma tissues, tissue was deparaffinized, rehydrated and blocked with 3% peroxide. Heat-induced antigen retrieval was completed using Target Antigen Retrieval solution (Dako) in a pressure cooker for 45 min. Sections were blocked with 10% animal serum for 30 min and incubated with NECTIN1 primary antibody (1:50; HPA026846, Sigma-Aldrich) at 4 °C overnight. Sections were then incubated with goat anti-rabbit horseradish peroxidase (HRP) (1:200, Vector Laboratories) secondary antibody for an hour for visualization and then processed with DAB + Chromagen (Dako) and hematoxylin counterstain. For staining of zebrafish melanoma tissues, immunohistochemistry was performed on the Leica Biosystems Bond III automated staining platform. Phospho-Histone3 (Ser10) antibody (#9701, Cell Signaling Technology) or phospho-FAK (Tyr397) antibody (44−624 G, Invitrogen) were run at 1:200 dilution using the Leica Biosystems Refine Detection Kit with citrate antigen retrieval, after bleaching of the melanin pigment. Immunostains were scored as previously described[44] (0 = no staining, 1 = weak and incomplete staining in <10% of targeted cells, 2 = weak/moderate heterogeneous staining in >10% of targeted cells and 3 = strong and complete homogenous staining in >10% of targeted cells). Data analysis was done using GraphPad Prism.

### Human melanoma cell lines

293 T, RPMI7951, CJM, A375, SKMEL2, COLO679, and G361 cell lines were purchased from commercial sources (ATCC CRL-3216, ATCC HTB-66, Creative Bioarray CSC-C6421J, ATCC CRL-1619, ATCC HTB-68, Sigma 87061210, and ATCC CRL-1424, respectively) and cultured in DMEM (Life Technologies), except for G361 (McCoy's media, 10% FBS (Atlanta Biologicals), 1X GlutaMAX (Life Technologies) and 1%

penicillin-streptomycin (Life Technologies)). Cells were grown at 37 °C in 5% $CO_2$. All cultures were regularly checked for the presence of mycoplasma. For cell culture experiments implicating genetic or pharmacological modifications, cells were randomly distributed between the different experimental conditions.

### Gene inactivation

Genes were inactivated in primary zebrafish melanomas using a technique that we previously developed[22,23]. This technique is based on CRISPR MiniCoopR, a DNA plasmid containing one or two U6:gRNA cassettes targeting the gene(s) of interest and a zebrafish Cas9 sequence under the control of the melanocyte-specific *mitf* promoter. The Tol2 transposase technology[45] permits insertion of this vector into the zebrafish genome upon microinjection into one-cell stage embryos. Stable expression subsequently results in the CRISPR inactivation of the gene of interest specifically in melanocytes. Use of this technique in a melanoma-prone zebrafish line, such as Tg(mitfa:*BRAF*$^{V600E}$), *tp53*$^{−/−}$, *mitfa*$^{−/−}$, allows to study the role of a gene in genetically engineered primary tumors.

To inactivate *NECTIN1* in human melanoma cells, we selected an shRNA consistently reducing endogenous NECTIN1 protein levels by over 50% to mimic the effect of deletions observed in human patients (Extended Data Fig. 3g). We also established *NECTIN1* CRISPR knockout lines from individual clones of A375 cells transduced with a lentiviral CRISPR construct targeting early exons of the gene. Knockout was confirmed by the presence of frameshift indels at the target sites by deep-sequencing (Extended Data Fig. 3h) and absence of the protein by western blotting (Extended Data Fig. 3i). Of note, *NECTIN1* mRNA level was also reduced in these knockout lines compared to wild-type parental lines, presumably due to non-sense mediated decay (Extended Data Fig. 3j).

### DNA constructs

Two control vectors were used in this study, one containing a non-targeting gRNA and the other targeting *arhgap11a*, a gene unrelated to *nectin1* that does not impact melanoma initiation or progression in our system. gRNAs targeting *arhgap11a, nectin1a, nectin1b, fak1a, fak1b* (Supplementary Table 4) were introduced into the CRISPR MiniCoopR vector[23] by restriction cloning using the BseRI and AarI enzymes. A pLKO shRNA lentiviral vector (TRCN0000373345, Sigma-Aldrich) was used to down-regulate *NECTIN1* in human melanoma cell lines.

The LentiCRISPR lentiviral vector[46] was used to inactivate *NECTIN1* in human melanoma cell lines (Supplementary Table 4).

The *CD8-IGF1R* ORF was cloned into a middle-entry vector, and assembled by multisite Gateway technology[47] with either the MiniCoopR destination vector[48] for melanocyte-specific expression in zebrafish or the pHAGE vector for doxycycline-inducible expression in human cells. A GFP construct was used as a negative control. The *NECTIN1* ORF was obtained from the Harvard Plasmid repository.

### CRISPR sequencing

Genomic DNA from human cell lines or zebrafish tumors was extracted with QuickExtract solution (Epibio). DNA libraries were prepared by PCR amplification of the CRISPR loci using the primers described in Supplementary Table 4.

Sequencing was performed by the MGH DNA core and analyzed using the Basepair online tool (https://www.basepairtech.com).

### Zebrafish

Zebrafish (*Danio rerio*) were handled according to our vertebrate animal protocol that has been approved by Boston Children's Hospital Animal Care Committee and includes detailed experimental procedures for all in vivo experiments described in this paper. Zebrafish of the *casper* (*mitfa*$^{−/−}$, *roy*$^{−/−}$) and Tg(mitfa:*BRAF*$^{V600E}$), *tp53*$^{−/−}$,

*mitfa*[−/−] strains[48,49] were housed at a density of approximately 10 fish per liter of water at 28.5 °C and bred by pairwise crossing. Embryos were collected for microinjection.

## Zebrafish embryo microinjection

Zebrafish of *casper* (*mitfa*[−/−], *roy*[−/−]) and Tg(mitfa:*BRAF*[V600E]), *tp53*[−/−], *mitfa*[−/−] strains[48,49] were bred and embryos were collected for microinjection. The Tol2 transgenesis technology enables the stable integration of expression vectors into the fish genome[45]. Embryos were injected at the one-cell stage with CRISPR MiniCoopR vectors[23] targeting *nectin1*, *fak* or a control gene and/or MiniCoopR vectors expressing CD8-IGF1R or GFP. *25* pg DNA constructs and 25 pg *Tol2* mRNA were injected into one-cell-stage embryos. After microinjection, embryos were raised in E3 medium at 28.5 °C. Embryos were sorted for melanocyte rescue at 96 h after fertilization and raised to adulthood (15–25 zebrafish per 3.5-liter tank). Tumor formation was monitored between weeks 10 and 22 after injection. Pigmented tumors were collected randomly from adult male and female fish (between 3 and 6 months of age) for transplantation. We did not observe any differences in tumor onset between males and females.

## In vivo tumor spreading assay

Transplantation of primary zebrafish melanoma cells into irradiated adult *casper* recipients was performed as previously described[25]. Specifically, 3- to 6-month-old zebrafish were irradiated at 30 Gy split over 2 days, and 300,000 melanoma cells in suspension in 3 µl PBS were injected subcutaneously into their dorsal cavity using a 10 µl Hamilton syringe with 32-gauge bevel-tipped needle. Each primary tumor was transplanted into 5–10 secondary recipients that were randomly allocated to each genetic condition. No animals were excluded from data analysis. The proportion of females was lower than that of males (approximately 20%) but identical in all experimental groups. We did not observe any influence of sex in our transplantation experiments. Recipient fish were photographed using a Nikon D3100 camera with an AF-S Micro NIKKOR 60 mm lens every 7 days after transplantation for 4 weeks.

## AFM

Atomic force microscopy (AFM) was performed on whole primary melanomas freshly excised from adult zebrafish. Dimension 3100 (Bruker Nano/Veeco) AFM with Nanoscope V controller and nPoint NPXY200Z25 (nPoint) close-loop scanner (200 µm × 200 µm × 25 µm, XYZ) were used in the present study. A standard cantilever holder for operation in liquids was used. The raw indentation data were collected with the vertical ramp size of 15 µm. To minimize viscoelastic effects, the indentation data were recorded at a frequency of 0.75 Hz. The AFM software used was version 7.30 R1Sr3. The data were also pre-processed to align the baseline with SPIP version 6.7.2 (Image Metrology A/S).

The AFM indentation curves were collected using the force-volume mode. 16 × 16 pixels (typically within 80 × 80 µm² area) images were recorded. Standard V-shaped arrow 200 µm AFM tipless cantilevers NP-O10 (Bruker) were used throughout the study. The cantilever spring constant was measured using the thermal tuning method before gluing the spherical probe. Typical spring constant was ~0.06 N m⁻¹. Silica balls 5 µm in diameter (Bangs Labs) were glued to the cantilevers as previously described[50]. The radius of the probe was measured by imaging the inverse grid (TGT1 by NT-NGT, Russia). The use of a rather dull AFM probe is important to attain self-consistency of the models[51]. In each experiment, the AFM cantilever sensitivity was calibrated against a rigid substrate, a small piece of a silicon wafer or coverslip immersed in PBS solution.

In the force-volume mode, one collects both cell sample topography and the indentation data[52,53]. Furthermore, the sample topography can be corrected for the inevitable deformation of the sample by the action of the AFM probe[51]. We processed the indentation curves collected over

the top points on the sample until we reached the inclination angle of 10–15 degrees. The Young's modulus of the sample surface was found using the Hertz model built in the SPIP software (Image Metrology A/S). The specific range of fitting was identified by the requirement of independence of the modulus of the indentation depth[54].

## Primary zebrafish cell cultures

Zebrafish cell lines were established as previously described from *BRAF*[V600E], *tp53*[−/−], *nectin1*[+/+] or *nectin1*[−/−] tumors[25]. Whole tumors were dissociated with liberase, plated in 48-well plates coated with fibronectin and cultured in the presence of zebrafish media (DMEM:F12 supplemented with 15% FBS, 10% zebrafish embryo extract, HEPES, sodium pyruvate, glutamine, insulin, holotransferrin, selenous acid, endothelin 1, chemically defined lipids, non-essential amino acids, penicillin/streptomycin and primocin). After three or four passages (or when cells expanded from a six-well plate to a T25 flask), cells were gradually transitioned to DMEM + 10% FBS. All cultures were regularly checked for the presence of mycoplasma.

## Transfection and viral transduction

Lentiviral particles were produced by co-transfection of 293 T cells with pLKO or pHAGE vectors and packaging plasmids pVSV-G and psPAX2 using FuGENE HD (Promega). Viral particles were harvested 48 and 72 h after transfection, concentrated by overnight PEG precipitation, resuspended in PBS and stored at −80 °C. Human melanoma cell lines were overlaid with viral particles diluted in medium supplemented with 5 µg ml⁻¹ polybrene (Sigma-Aldrich) for 12 h at 37 °C. Then, 48 h after transduction, infected cells were selected with 500 µg ml⁻¹ G418 (Gibco) or with 1 µg ml⁻¹ puromycin (Gibco) and maintained under selection by replacing the antibiotics every 48 h.

A375 cells were transfected with pLentiCRISPR vectors using FuGENE HD (Promega). Stable knockout lines were established from single-cell colonies.

siRNA against FAK (SASI_Hs01_00035697), SRC (Hs01_00112907), ITGB1 (SASI_Hs02_00333437), ITGB2 (SASI_Hs01_00245442), ITGB3 (SASI_Hs01_00174219), ITGB4 (SASI_Hs01_00230929), ITGB5 (SASI_Hs02_00333438) were purchased from Millipore and transfected using the N-TER Nanoparticle siRNA Transfection System according to the manufacturer's instructions.

## Adhesion and migration assays

Spheroid formation capacity was evaluated by plating 100–10,000 human melanoma cells in the wells of a Round Bottom Ultra Low Attachment 96-well Microplate (Corning) in the presence or absence of FBS. Spheres were imaged after 4–8 days using a Nikon SMZ18 stereomicroscope or a Nikon C2-si confocal microscope. Circularity was measured using ImageJ and multiplied by 100 for ease of representation. A perfect circle would have a circularity of 100 (Fig. 3b).

For the adhesion assay, wells of a 12-well plate were coated with Collagen-I (3440, R&D Systems), fibronectin (F0895, Sigma-Aldrich), laminin (LN521-02, Biolamina), vitronectin (5051, Sigma-Aldrich) at 37 °C for 3 h and then at 4 °C overnight. Human melanoma cells were transduced with a lentiviral vector stably expressing GFP, harvested and resuspended in the absence of FBS. Then, 25,000 cells were seeded in ECM protein-coated well. After 15-, 30- or 45-min incubations, wells were rinsed with PBS and immediately imaged using a Nikon SMZ18 microscope equipped with a DS-Ri2 camera. The number of attached cells was quantified with ImageJ.

The Transwell assay was performed with 6.5 mm Transwell with 8.0 µm Pore Polycarbonate Membrane Insert (Corning) according to the manufacturer's instructions. For invasion assays, Matrigel (2515251, Corning) containing the cells was added to the transwells. 40,000 cells were seeded in the absence of FBS for 12 h. The membranes were then transferred to a well containing media with FBS. Cells were allowed to migrate for 6 h (A375, CJM), 10 h (RPMI7951, G361) or 24 h (SKMEL2,

COLO679), fixed, stained with DAPI and imaged using a Nikon SMZ18 microscope equipped with a DS-Ri2 camera. The number of cells that have migrated through the membrane was quantified with ImageJ.

For colony-forming assays in semi-solid media, 1,000 A375 human melanoma cells were seeded in 250 µl 50% Matrigel (Corning) or 1x Geltrex Collagen-I matrix (R&D Systems) on a layer of 250 µl 100% Matrigel or 10x Geltrex Collagen-I matrix in 24-well plates. Then, 1 ml culture media was added to wells containing Matrigel. Spread was measured with ImageJ as the number of individual particles detected in each colony (Fig. 3d).

## Migration inhibitors

The following compounds were tested for toxicity with the CellTiter-Glo assay after a 4-day treatment in A375 melanoma cells and used at the maximal nontoxic dose in the transwell assay: Integrin inhibitor (Echistatin, 100 nM, R&D Systems), FAK inhibitor (PF562271 besylate, 500 nM, Cayman Chemicals), SRC inhibitor (Dasatinib, 10 nM, LC Laboratories), RAC1/CDC42 inhibitor (MBQ-167, 50 nM, Selleckchem), Wnt inhibitor (XAV939, 1 µM, Cayman Chemicals), JNK inhibitor (SP600125, 1 µM, LC Laboratories), ROCK inhibitor (RKI-1447, 500 nM, Cayman Chemicals), PKC inhibitor (Staurosporine, 1 nM, LC Laboratories), AMPKa inhibitor (Dorsomorphin, 1 µM, LC Laboratories), integrin inhibitor (SB273005, 100 nM, Selleckchem), FAK inhibitor (GSK2256098, 1 µM, Cayman Chemicals) and SRC inhibitor (Bosutinib, 100 nM, LC Laboratories). Echistatin was resuspended in water. All other compounds were dissolved in DMSO.

In addition, CD49f (Integrin alpha 6) monoclonal antibody (GoH3) (14-0495-85, eBioscience) was used at 40 µg ml$^{-1}$ to block integrin α6β4.

## Screen for growth factors and hormones

The following factors were tested for their ability to prevent the formation of adherens junctions between melanoma cells upon serum withdrawal, as assessed by immunofluorescence for α-catenin: EGF (1 ng ml$^{-1}$, StemCell Technologies), bFGF (10 ng ml$^{-1}$, StemCell Technologies), IGF1 (100 ng ml$^{-1}$, StemCell Technologies), SCF (25 ng ml$^{-1}$, StemCell Technologies), PDGF-BB (100 ng ml$^{-1}$, Gibco), TGFβ (10 ng ml$^{-1}$, StemCell Technologies), insulin (10 µg ml$^{-1}$, R&D Systems), IFN-α (100 pg mll$^{-1}$, StemCell Technologies), NTN1 (1 µg ml$^{-1}$, R&D Systems), retinoic acid (100 ng ml$^{-1}$, Sigma-Aldrich), β-estradiol (100 pg ml$^{-1}$, Sigma-Aldrich) and progesterone (20 ng ml$^{-1}$, Sigma-Aldrich).

## Immunofluorescence

Cells were grown on glass coverslips for 2 days, fixed with 4% PFA for 20 min, permeabilized with PBS 0.2% Triton for 30 min at 4 °C, blocked with 1% BSA in PBS, 0.05% Triton for 1 h at room temperature, hybridized with primary antibody in blocking solution for 2 h at room temperature, hybridized with fluorescently-labeled secondary antibody in blocking solution for 2 h at room temperature in the dark, rinsed in PBS and mounted on glass slides using VECTASHIELD Antifade Mounting Medium with DAPI (Vector Laboratories). Primary antibodies: α-E-Catenin (1:200, #3236, Cell Signaling Technology), N-Cadherin (1:200, #13116, Cell Signaling Technology), α-Actinin (1:100, #69758, Cell Signaling Technology), fibronectin (1:400, #26836, Cell Signaling Technology) and laminin (1:100, L9393, Sigma-Aldrich). Secondary antibodies used were Alexa Fluor 488 goat anti-rabbit (1:400, A11008, Life Technologies) and Alexa Fluor 568 goat anti-mouse (1:400, A11004, Life Technologies). F-actin was stained using Alexa Fluor 568 Phalloidin (1:2000, A12380, Life Technologies). Cells were imaged using a Nikon C2-si confocal microscope.

For immunofluorescence on human melanoma tissue or zebrafish tissue, sections were deparaffinized through a xylene/ethanol gradient and rehydrated. Heat-induced antigen retrieval was completed using Antigen Unmasking Solution, Citric Acid Based (H-3300, Vector) in a microwave for 20 min. Sections were blocked with 5% BSA in PBS for 30 min and incubated with α-catenin (1:200, 13-9700,

Invitrogen), phospho-IGF1R beta (Tyr1161) (1:200, PA5-37601, Invitrogen), E-cadherin (4A2) (1:200, #14472, Cell Signaling Technology), NECTIN1 (1:200, HPA026846, Sigma-Aldrich), laminin (1:100, L9393, Sigma-Aldrich) and FNDC3A (1:200, PA5-109309, Invitrogen) primary antibodies at 4 °C overnight. Sections were then incubated with Alexa Fluor Plus 488 goat anti-Mouse IgG (1:400, A32723, Life Technologies) and Alexa Fluor 568 goat anti-rabbit IgG (1:400, A11036, Life Technologies) secondary antibodies for 2 h and mounted using VECTASHIELD Antifade Mounting Medium with DAPI (Vector Laboratories).

## Protein analysis

For western blot, cells were lysed with RIPA lysis buffer containing protease inhibitors (cOmplete, Roche) and phosphatase inhibitors (PhosSTOP, Roche). Lysates were incubated for 20 min on ice and spun down for 10 min at 14,000 r.p.m. at 4 °C. Protein concentrations were normalized using the DC protein assay (BioRad). Samples were denatured by adding Laemmli sample buffer (BioRad) with 5% β-mercaptoethanol (Sigma-Aldrich) and boiled at 95 °C for 5 min before loading. Proteins were separated on a 10% mini-PROTEAN TGX (BioRad) precast gel, and transferred onto a PVDF membrane using the iBlot2 system (Thermo Fisher Scientific). Primary antibodies: NECTIN1 (1:1,000, ab66985, Abcam), phospho-FAK (1:1,000, #3283, Cell Signaling Technology), FAK (1:1,000, #71433, Cell Signaling Technology), phospho-SRC (1:1,000, #2101, Cell Signaling Technology), SRC (1:1,000, #2109, Cell Signaling Technology), integrin β1 (1:1,000, #9699, Cell Signaling Technology), integrin β2 (1:1,000, #73663, Cell Signaling Technology), Integrin β3 (1:1,000, #13166, Cell Signaling Technology), integrin β4 (1:1,000, #14803, Cell Signaling Technology), integrin β5 (1:1,000, #3629, Cell Signaling Technology), GAPDH (1:2,000, ab8245, Abcam) and β-ACTIN (1:5,000, A2228, Sigma-Aldrich). Secondary antibodies used were HRP anti-mouse and HRP anti-rabbit (1:2,000, Cell Signaling Technology). Membranes were developed using Pierce ECL Western Blotting Substrate (Thermo Fisher Scientific) or ECL Prime Western Blotting Detection Reagent (Amersham). Activity of intracellular signaling cascades was measured using a Proteome Profiler Human Phospho-Kinase Array Kit (R&D Systems) following the manufacturer's instructions. Signal from the immunoblot was quantified using ImageJ.

## Purification and analysis of cell-surface proteins

Cell-surface proteins were purified using the EZ-Link Sulfo-NHS-SS-Biotin system (Thermo Fisher Scientific). Cells were rinsed with PBS (+ CaCl$_2$/MgCl$_2$) and incubated in a 0.5 mg ml$^{-1}$ biotin solution for 1 h on ice. The free biotin was quenched by three 5-min washes with a 50 mM glycine solution. Cells were rinsed with PBS and lyzed in an immunoprecipitation buffer (Cell Signaling Technology) containing protease inhibitors. After centrifugation of the lysate, Neutravidin beads (Thermo Fisher Scientific) were added 1:2 to the supernatant. The beads/lysate mixture was incubated at 4 °C for 3 h under constant rotation. Beads were washed once with the immunoprecipitation buffer and then three times for 5 min with a 50 mM Tris-HCl solution. After on-bead digestion with trypsin, purified peptides were labeled and analyzed by tandem mass spectrometry using a 180-min MS3 method on an Orbitrap Fusion Lumos mass spectrometer.

## Gene expression analyses

Read length was 150 bp pair ended. Quality control of RNA-sequencing datasets was performed by FastQC and Cutadapt to remove adaptor sequences and low-quality regions. The high-quality reads were aligned to UCSC build hg19 of human genome using Tophat 2.0.11 without novel splicing form calls. Transcript abundance and differential expression were calculated with Cufflinks 2.2.1 (ref. [55]). Raw counts were not normalized but FPKM values were used to normalize and quantify each transcript. Genes were filtered by log fold change and q-value. Differentially regulated pathways were analyzed by Gene Ontology[56,57] and gene set enrichment analysis[58,59].

## Statistics

Statistical tests used in this study are indicated in figure legends. Comparison of Kaplan–Meier survival curves was performed by a log-rank (Mantel–Cox) test. Statistical differences in cell culture experiments were estimated using paired two-tailed $t$-test. For other experiments, a two-tailed $t$-test was used. Data distribution was assumed to be normal, but this was not formally tested. An F test was used to compare variances. For gene expression analyses, statistical methods are describes in the original Gene Ontology[56,57] and gene set enrichment analysis[58,59] papers. For each statistical test, the $P$ value is reported on the figure. Significance was defined as $P$ value < 0.05. Dispersion of the data and precision measures are specified in figure legends. Generally, graphs presented in this study show the mean ± s.d. The number of replicates indicated on figures or in figure legends refers to biological replicates. For cell line work, no statistical methods were used to predetermine sample size, but our sample sizes are similar to those reported in previous publications. Experiments were repeated four times to ensure robustness, unless statistical significance could be reached with three replicates only. For zebrafish transplantation experiments, a target sample size of 16 primary tumors per arm was calculated such that twofold differences in tumor spreading could be observed with a power of 80% and with statistical significance (5% type I error) using two-sided Student's $t$-test. This sample size was never attained because statistical significance was reached with smaller samples due to the larger size of the observed effects. For human melanoma sections, sample size was limited by availability. Analyses of the screens for growth factors and hormones and migration inhibitors were blinded. Data collection was also performed blinded in the case of transplantation experiments in zebrafish. Blinding was only lifted at the time of data analysis.

## Reporting summary

Further information on research design is available in the Nature Research Reporting Summary linked to this article.

## Data availability

The TCGA dataset for cutaneous melanoma used in this study is available at https://portal.gdc.cancer.gov/projects/TCGA-SKCM. Copy-number analyses were performed with GISTIC 2.0 through the interface of the Broad Institute (https://portals.broadinstitute.org/tcga/home) using 2014-04-28 stddata release.
The RNA-sequencing data of the consequences of NECTIN1 loss and the effect of serum depletion on A375 human melanoma cells generated during this study have been deposited in the Gene Expression Omnibus under the accession code GSE174150 and are publicly available.
The tandem mass spectrometry data of the cell-surface proteins of NECTIN1-deficient and NECTIN1-proficient A375 human melanoma cells upon serum depletion generated during this study have been deposited in MassIVE (Center for Computational Mass Spectrometry, UCSD) under the accession code MSV000090170 and are publicly available. Source data are provided with this paper.

## Code availability

All data processing and analysis were performed by existing software packages, which have been included in the relevant Methods section. No custom code or software was used for any aspect of data processing or analysis.

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

## Acknowledgements

We thank A. Weng and S. Gusscott for sharing their CD8-IGF1R plasmid, C. Yuan for technical support with immunohistochemistry, S. Rowbotham for comments on the manuscript, N. Osmani for advice regarding integrin signaling and S. Avagyan for critical reading of the manuscript, discussions on the data and logistical support between the Lyon and Boston laboratories. This work was supported by grants from the National Institutes of Health/National Cancer Institute (R01 CA103846 and P01 CA163222 to L.I.Z. and K99 CA201465 to J.A.), the Sloan Kettering Institute for Cancer Research (BD527651 to L.I.Z.), the Melanoma Research Alliance, the National Science Foundation (CMMI 1435655 and CMMI 1937373 to I.S.), the Centre National de la Recherche Scientifique (ATIP Avenir 2019 to J.A.), the Institut National de la Santé et de la Recherche Médicale, the Ligue contre le Cancer, the Fondation ARC pour la recherche sur le cancer and the Fondation Tourre. L.I.Z. is a Howard Hughes Medical Institute Investigator.

## Author contributions

J.A. designed the study, performed cell culture and zebrafish experiments, analyzed results, discussed findings and wrote the manuscript, A.A. performed cell culture experiments and analyzed results, H.R. helped with cell culture and zebrafish experiments, M.P. and S.Y. performed RNA-sequencing analyses, S.A. participated in cell culture experiments, M.D. performed initial zebrafish transplantation experiments, S.X. and C.G.L. performed and analyzed

immunohistochemistry experiments and M.E.D. and I.S. performed and analyzed atomic force microscopy experiments. L.I.Z. designed the study, discussed findings and wrote the manuscript.

## Competing interests

L.I.Z. is a founder and stock holder of Fate Therapeutics, CAMP4 Therapeutics, Scholar Rock, and Amagma, outside of the present work. The other authors declare no competing interests.

## Additional information

**Extended Data** are available for this paper at https://doi.org/10.1038/s41588-022-01191-z.

**Correspondence and requests for materials** should be addressed to Julien Ablain.

**a**

| Region | q-value | Frequency (%) | # genes | Genes |
|---|---|---|---|---|
| chr9:21963430-21996996 | 3.89E-164 | 76.6 | 2 | C9ORF53, **CDKN2A** |
| chr10:89617401-90033423 | 4.53E-27 | 62.9 | 2 | KLLN, **PTEN** |
| chr11:119375857-119978951 | 2.76E-21 | 53.8 | 1 | **NECTIN1** |
| chr4:178913816-183059542 | 6.88E-11 | 28.4 | 1 | LINC00290 |
| chr2:239464310-242085734 | 9.12E-11 | 24.7 | 34 | |
| chr1:3218610-6843928 | 3.07E-9 | 23.1 | 139 | |
| chr16:5725109-7764030 | 7.89E-6 | 18.7 | 2 | MIR8065, RBFOX1 |
| chr1:79013045-119424027 | 8.65E-5 | 25.4 | 304 | |
| chr6:161695943-163152563 | 9.69E-5 | 61.5 | 1 | PARK2 |
| chr10:415240-849979 | 1.16E-4 | 54.5 | 7 | |
| chr15:33027254-33439414 | 1.2E-4 | 24.4 | 0 | |
| chr16:82219441-89379319 | 4.4E-4 | 35.8 | 110 | |
| chr10:134945927-135092590 | 0.0029 | 64.5 | 6 | |
| chr19:284018-2152203 | 0.0029 | 27.8 | 84 | |
| chr5:914233-14580576 | 0.0054 | 20.4 | 92 | |
| chr14:78403322-105986740 | 0.0069 | 33.8 | 306 | |
| chr15:38244770-38745783 | 0.0076 | 24.4 | 1 | SPRED1 |
| chr12:33053294-55339329 | 0.0085 | 21.1 | 239 | |

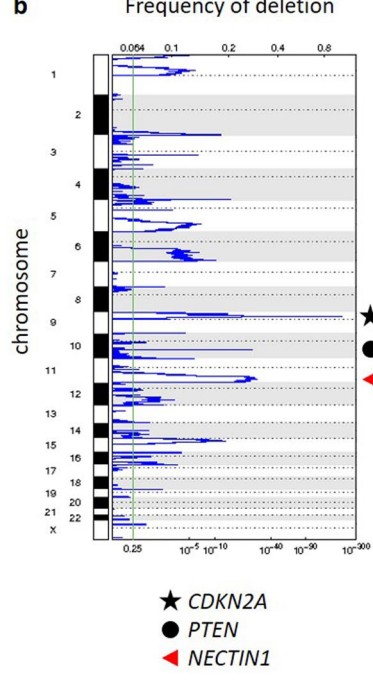

**b** Frequency of deletion

★ CDKN2A
● PTEN
◄ NECTIN1

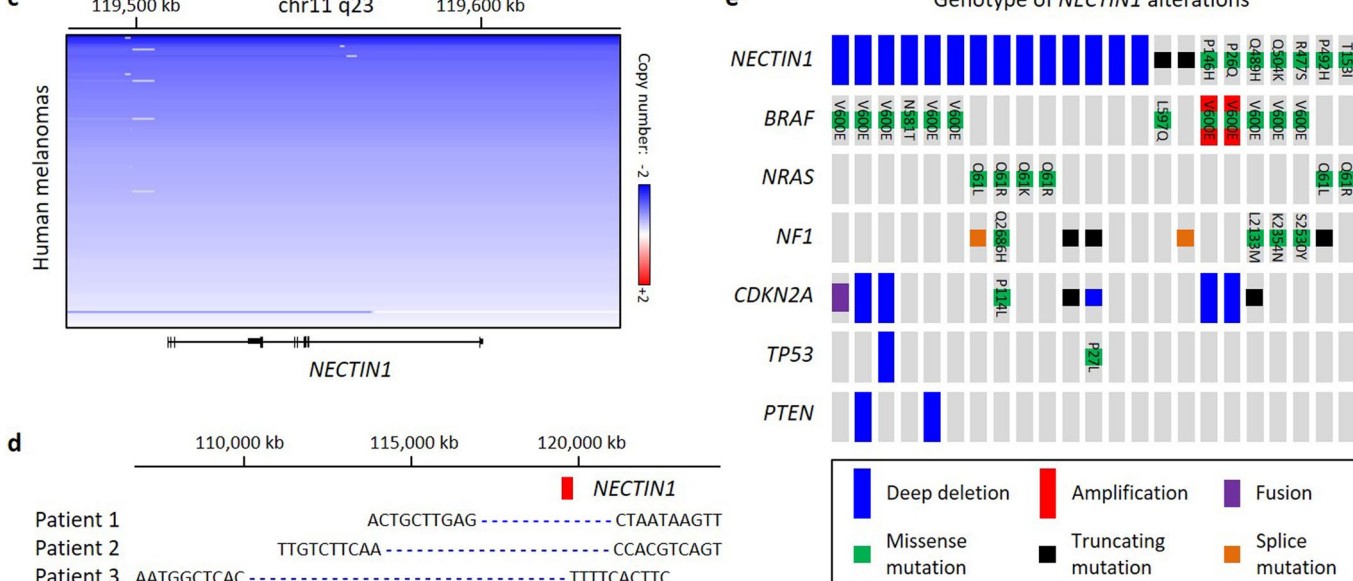

**c** chr11 q23

**d**

**e** Genotype of NECTIN1 alterations

Deep deletion — Amplification — Fusion
Missense mutation — Truncating mutation — Splice mutation

**Extended Data Fig. 1 | Genetics of *NECTIN1* in human melanoma. (a)** List of the 18 most significant chromosomal deletions found in the Cancer Genome Atlas (TCGA) cohort of human cutaneous melanomas (363 samples). **(b)** Diagram representing the frequency of chromosomal deletions in the TCGA cohort of human cutaneous melanomas (363 samples). Frequent loss of *CDKN2A* on chromosome 9, and *PTEN* on chromosome 10 are highlighted. *NECTIN1* was identified by GISTIC analysis in a recurrent focal deletion on chromosome 11. **(c)** Linear copy number in 202 human melanoma samples exhibiting deep or shallow deletions of the NECTIN1 locus. **(d)** Sequences of shallow deletions affecting NECTIN1 in three examples of human melanoma samples. In TCGA-D9-A148 (Patient 1) and TCGA-FS-A1ZQ (Patient 2), the deletion encompasses the entire NECTIN1 gene, while in TCGA-FW-A3TU (Patient 3), the deletion starts in exon 2. **(e)** Driver lesions in oncogenes and tumor-suppressor genes occurring with *NECTIN1* alterations in the TCGA cohort of human cutaneous melanomas (363 samples).

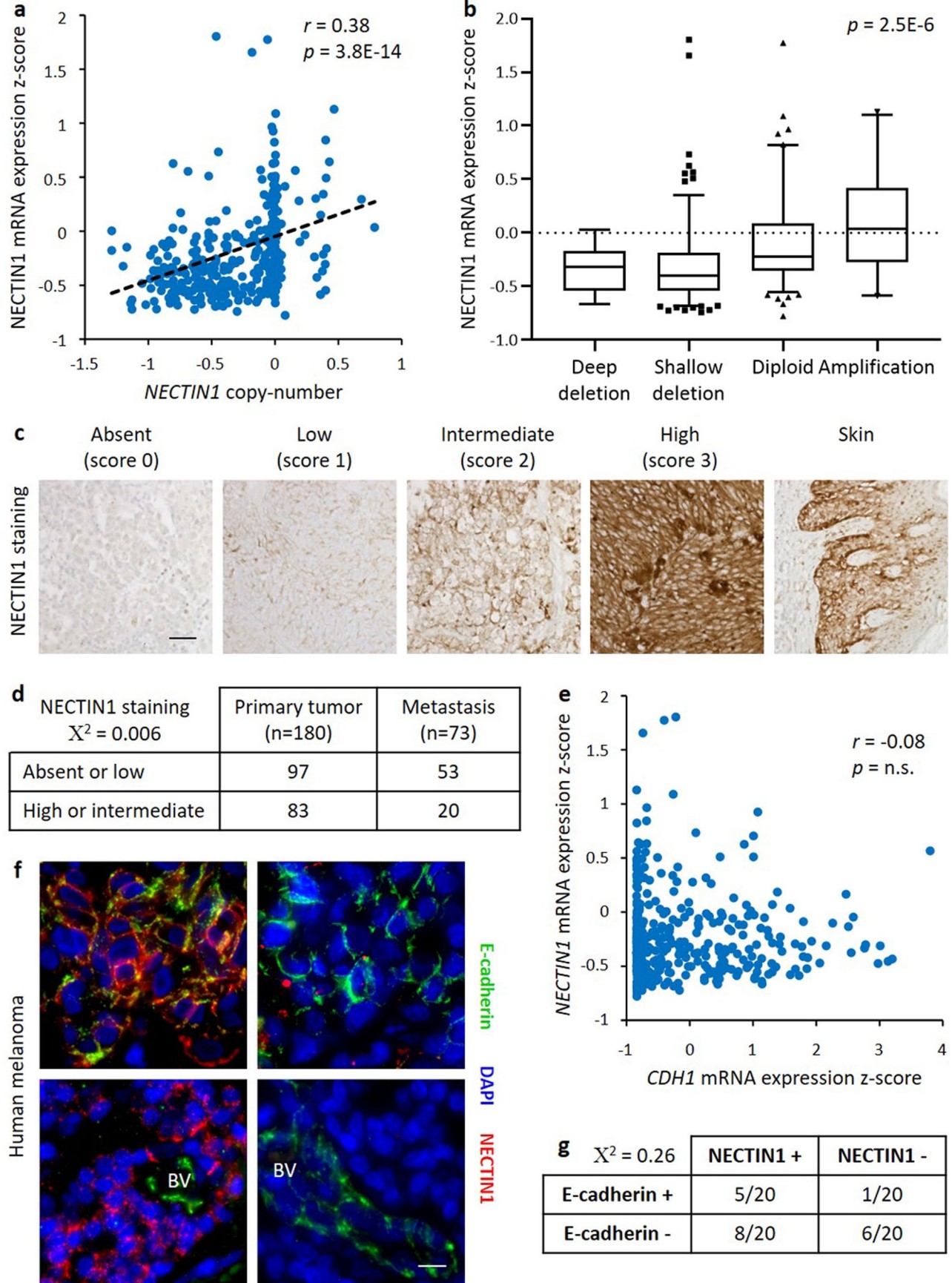

**Extended Data Fig. 2 | See next page for caption.**

**Extended Data Fig. 2 | *NECTIN1* expression in human melanoma. (a)** Dot plot representing *NECTIN1* mRNA expression and linear copy number in the TCGA cohort of human cutaneous melanomas (363 samples). Spearman correlation: r = 0.38, p = 3.8E-14. Four outliers with high *NECTIN1* expression were removed from the graph but retained in the analysis. **(b)** Box-and-whisker plot representing *NECTIN1* mRNA expression depending on NECTIN1 copy-number status (363 samples). Min-max, 5 and 95 percentiles, median. Brown-Forsythe ANOVA. Unpaired two-tailed Welch's t-test: deep deletion vs diploid: p = 0.0013, shallow deletion vs diploid: p = 0.0006. Four outliers with high *NECTIN1* expression were removed from the graph but retained in the analysis. **(c)** Representative histology and scoring system used for the evaluation of NECTIN1 expression by immunohistochemistry, related to Fig. 1d. 0 = Negative (<1% tumor cells immunoreactive); 1= weak and incomplete staining in <10% of cells, 2 = weak/moderate heterogeneous staining in >10% of cells, or 3 = strong and complete homogenous staining in >10% of cells. Scale bar: 50 µm. Representative of 3 independent tissue-microarrays. **(d)** Distribution of NECTIN1 staining level by immunohistochemistry in 253 tissue sections of human primary melanomas or metastases. Correlation was measured by Chi-square test. **(e)** Dot plot representing *NECTIN1* and *CDH1* mRNA expression in the TCGA cohort of human cutaneous melanomas (363 samples). Spearman correlation: r = −0.08, p = 0.107. Four outliers with high *NECTIN1* expression were removed from the graph but retained in the analysis. **(f)** Immunofluorescence analysis of E-cadherin (green) and NECTIN1 (red) on tissue sections of 4 different human melanomas exhibiting different staining patterns: from left to right and top to bottom: double positive, single E-cadherin-positive, single NECTIN1-positive, double negative. Scale bar: 10 µm. BV, blood vessel. Representative examples of the cases described in **g**. **(g)** Distribution of NECTIN1 and E-cadherin positivity by immunofluorescence in 20 sections of human melanoma. Correlation was measured by Chi-square test.

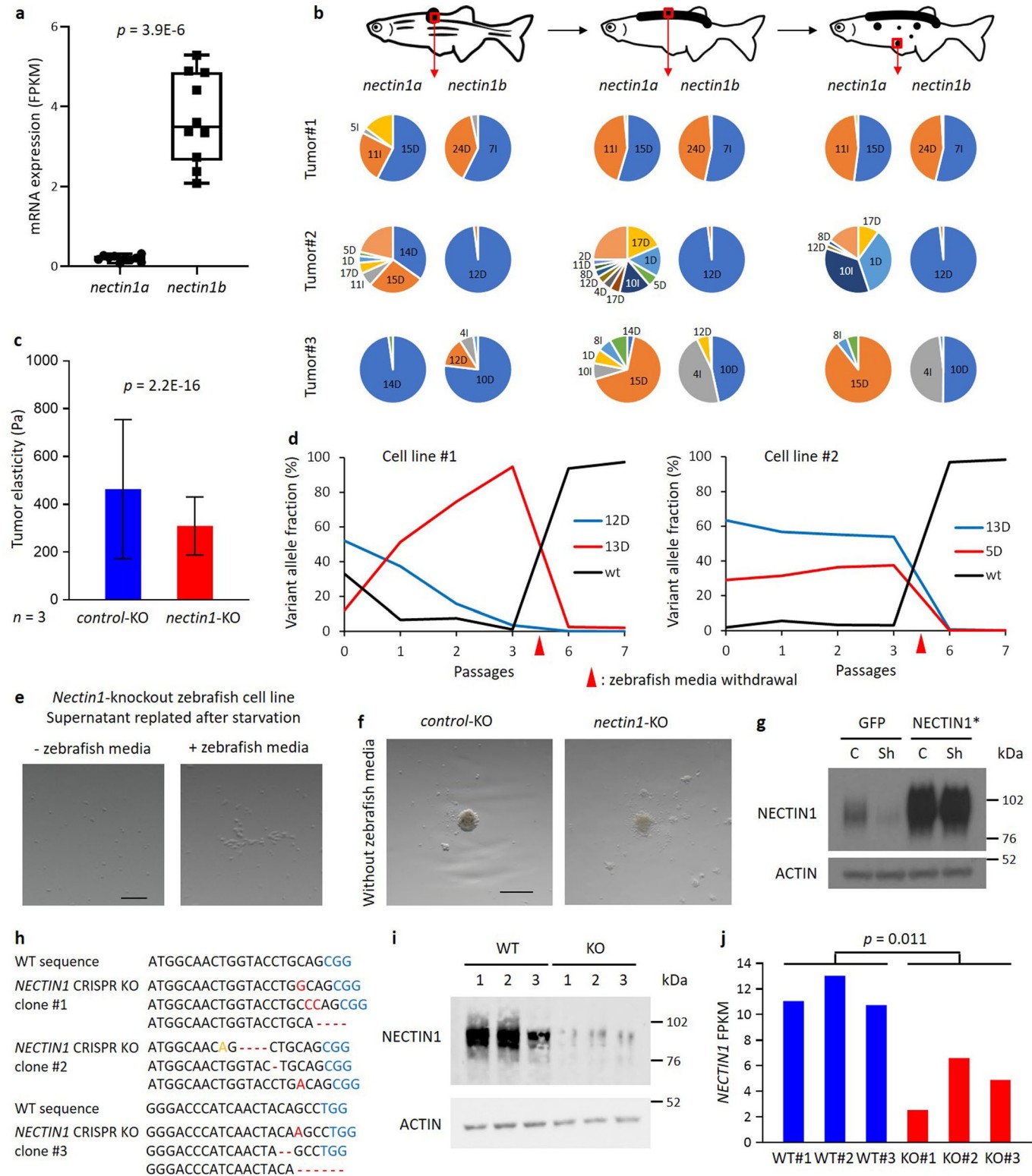

**Extended Data Fig. 3 | See next page for caption.**

**Extended Data Fig. 3 | *NECTIN1* inactivation in zebrafish and human melanoma.** (**a**) Box-and-whisker plot representing *nectin1a* and *nectin1b* expression levels in primary zebrafish melanomas. Min-max, 1st and 4th quartiles, median, n = 10. Two-tailed t-test. (**b**) Type and proportion of *nectin1a* and *nectin1b* mutant alleles in 3 primary zebrafish melanomas, their corresponding allografts in *casper* zebrafish, and matched disseminated patches. D: deletion, I: insertion, numbers denote affected base pairs. Unlabeled slice represents all other minor alleles. (**c**) Elasticity (Young's modulus) of primary zebrafish melanomas, as measured by atomic force microscopy. Mean ± SD of 4184 and 3134 measurements performed on 20 and 26 areas of 3 *control*-knockout and 3 *nectin1*-knockout tumors, respectively. Two-tailed Kolmogorov-Smirnov test. (**d**) Evolution of *nectin1b* variant allele fraction in 2 zebrafish melanoma cell lines over 7 passages in culture in the presence or absence of zebrafish media. wt: wild-type; 5D: 5 base-pair deletion. (**e**) Images of cultures of the supernatant of a *nectin1*-knockout cell line subjected to zebrafish media withdrawal. Scale bar: 50 µm. Representative of 3 independent experiments. (**f**) Spheroids formed by *control* or *nectin1*-knockout (KO) zebrafish melanoma cells after 4 days in low-attachment conditions in the absence of zebrafish media. Scale bar: 200 µm. Representative of 3 independent experiments. (**g**) Western blot analysis of NECTIN1 in A375 cells stably expressing a control shRNA (C) or an shRNA against *NECTIN1* (Sh), and either GFP or a *NECTIN1* construct mutated in the shRNA target site (NECTIN1\*). Representative of 3 independent experiments. (**h**) Sequences of the *NECTIN1* CRISPR loci in wild-type (WT) A375 cells and 3 *NECTIN1*-knockout (KO) cell lines derived from A375. Blue: PAM sequence; red: insertion/deletion; yellow: mutation. (**i**) Western blot analysis of NECTIN1 in 3 *NECTIN1*-wild-type and 3 *NECTIN1*-knockout (KO) cell lines derived from A375. Representative example of 2 independent experiments. (**j**) *NECTIN1* mRNA expression levels (FPKM) in the cells described in **i**. Two-tailed *t*-test.

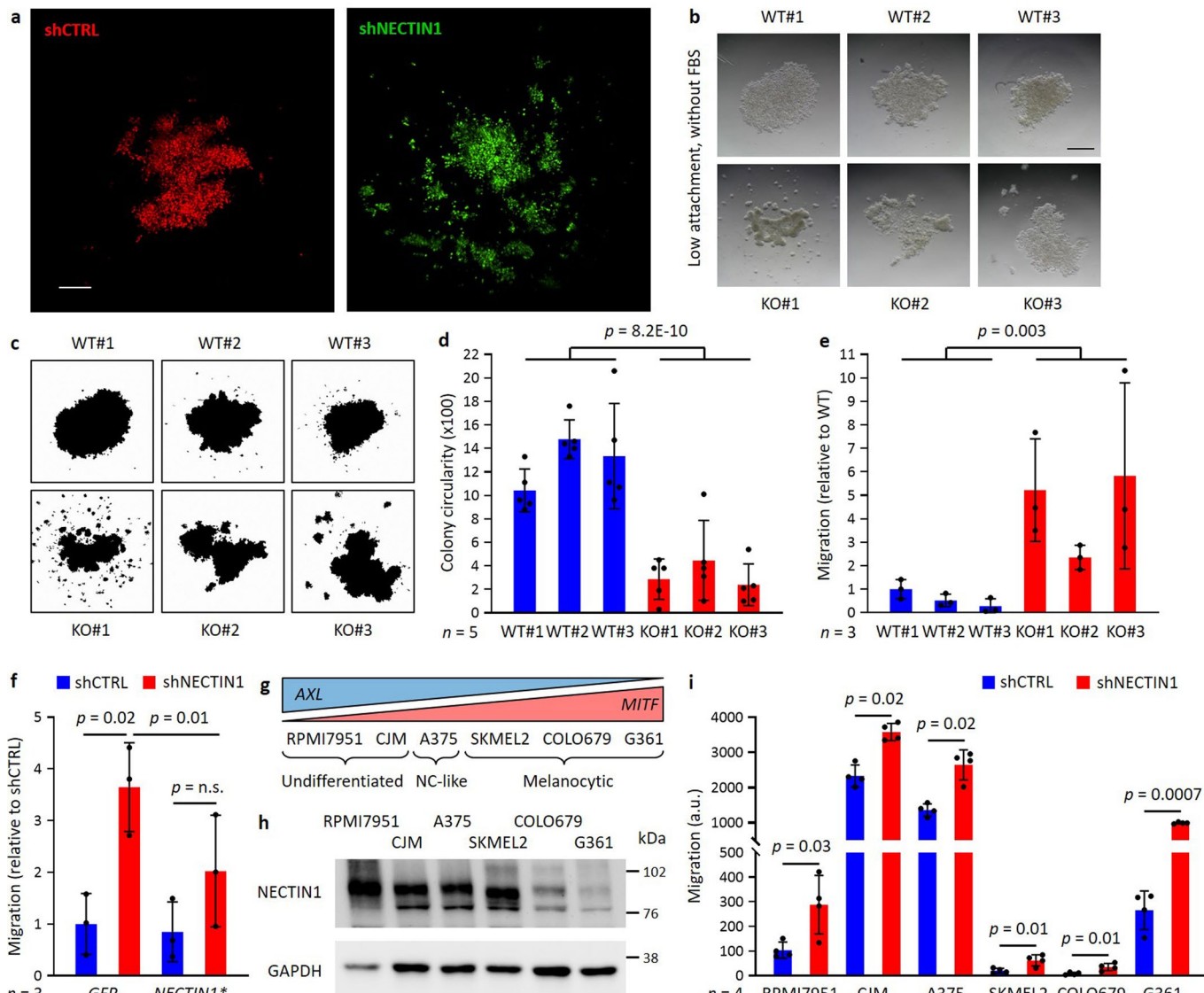

**Extended Data Fig. 4 | Migratory phenotype associated with *NECTIN1* inactivation in human melanoma cell lines.** (**a**) Structures formed by 5,000 A375 cells stably expressing a control shRNA (shCTRL) or an shRNA against *NECTIN1* (shNECTIN1) and stained with fluorescent dyes, after 4 days in low-attachment conditions in the absence of serum. Scale bar: 300 μm. Representative of 3 independent experiments. (**b**) Spheroids formed by 20,000 cells of 3 *NECTIN1*-wild-type (WT) and 3 *NECTIN1*-knockout (KO) cell lines derived from A375 cells, after 4 days in low-attachment conditions in the absence of serum. Scale bar: 500 μm. Representative of 5 independent experiments. (**c**) Analysis of the images shown in **b**. (**d**) Circularity of the colonies described in **b**, as measured with ImageJ. Mean ± SD of 5 independent colonies per condition. Two-tailed t-test. (**e**) Migration of the cells described in **b** in a transwell assay after 12 hours of serum starvation. Cells were allowed to migrate for 6 hours. Mean ± SD of 3 independent experiments. Two-tailed t-test. (**f**) Migration of the cells described in **a** stably expressing either GFP or an shRNA-resistant NECTIN1 gene (NECTIN1*) in a transwell assay after 12 hours of serum starvation. Cells were allowed to migrate for 6 hours. Mean ± SD of 3 independent experiments. Paired two-tailed t-test. (**g**) Diagram representing the position of 6 human melanoma cell lines on the spectrum of differentiation states from undifferentiated, migratory, AXL^high/MITF^low to melanocytic, proliferative, AXL^low/MITF^high. NC-like, neural crest-like. (**h**) Western Blot analysis of NECTIN1 in 6 human melanoma cell lines. Representative example of 3 independent experiments. (**i**) Migration of 6 human melanoma cell lines stably expressing a control shRNA (shCTRL) or an shRNA against NECTIN1 (shNECTIN1) in a transwell assay after 12 hours of serum starvation. Cells were allowed to migrate for 10 hours and were counted in a 2.35 ×2.35 mm field (a.u.). Mean ± SD of 4 independent experiments. Paired two-tailed t-test.

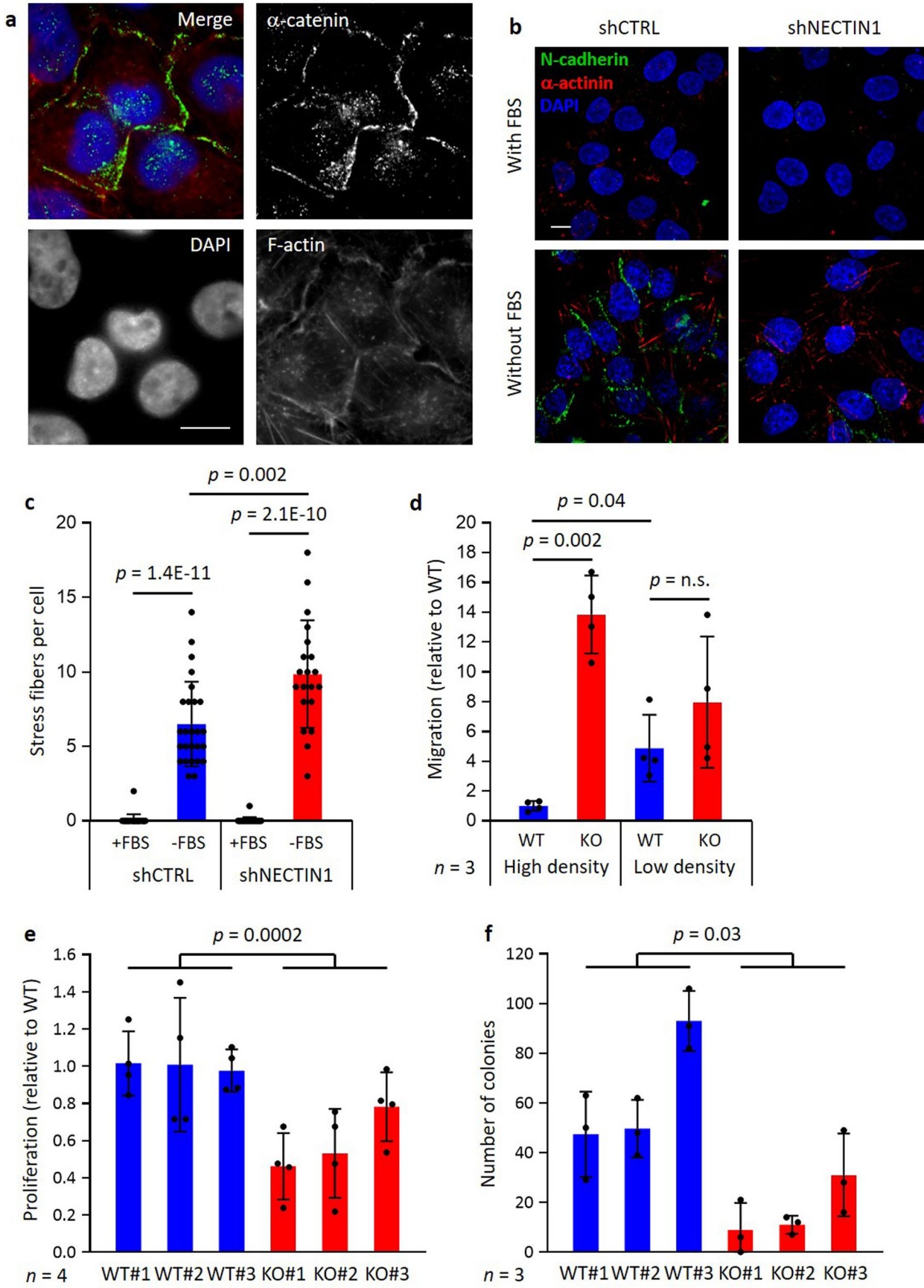

**Extended Data Fig. 5 | See next page for caption.**

**Extended Data Fig. 5 | Changes in cell adhesion and proliferation upon _NECTIN1_ loss in human melanoma cell lines.** (**a**) Immunofluorescence analysis of α-catenin (green) and F-actin (red) in A375 human melanoma cells stably expressing a control shRNA in the absence of serum for 12 hours. Scale bar: 10 μm. Representative images of 5 independent experiments. (**b**) Immunofluorescence analysis of N-cadherin (green) and α-actinin (red) in A375 human melanoma cells stably expressing a control shRNA (shCTRL) or an shRNA directed against _NECTIN1_ (shNECTIN1) and cultured in the presence or absence of serum (FBS) for 12 hours. Scale bar: 10 μm. Representative images of 5 independent experiments. (**c**) Quantification of the number of stress fibers per cell based on α-actinin staining on immunofluorescence images as shown in **b**. Mean ± SD of n = 29, 26, 24, 20 cells for shCTRL+FBS, shCTRL-FBS, shNECTIN1+FBS, and shNECTIN1-FBS, respectively. Paired two-tailed t-test. (**d**) Migration of _NECTIN1_-deficient cells depending on cell density in a transwell assay after 12 hours of serum starvation. Data was pooled from 3 _NECTIN1_-knockout (KO) cell lines relative to 3 _NECTIN1_-wild-type (WT) cell lines derived from A375 cells. Cells were seeded at high density (40,000 cells) or low density (4,000 cells) and allowed to migrate for 6 hours. Mean ± SD of 3 independent experiments. Paired two-tailed t-test. (**e**) Proliferation of the cell lines described in **d**, measured as the cell count 4 days after seeding relative to the average cell count in wild-type lines. Mean ± SD of 4 independent experiments. Two-tailed _t_-test. (**f**) Number of colonies formed in matrigel by the cells described in **e** after 14 days in culture. Mean ± SD of 3 independent experiments. Two-tailed _t_-test.

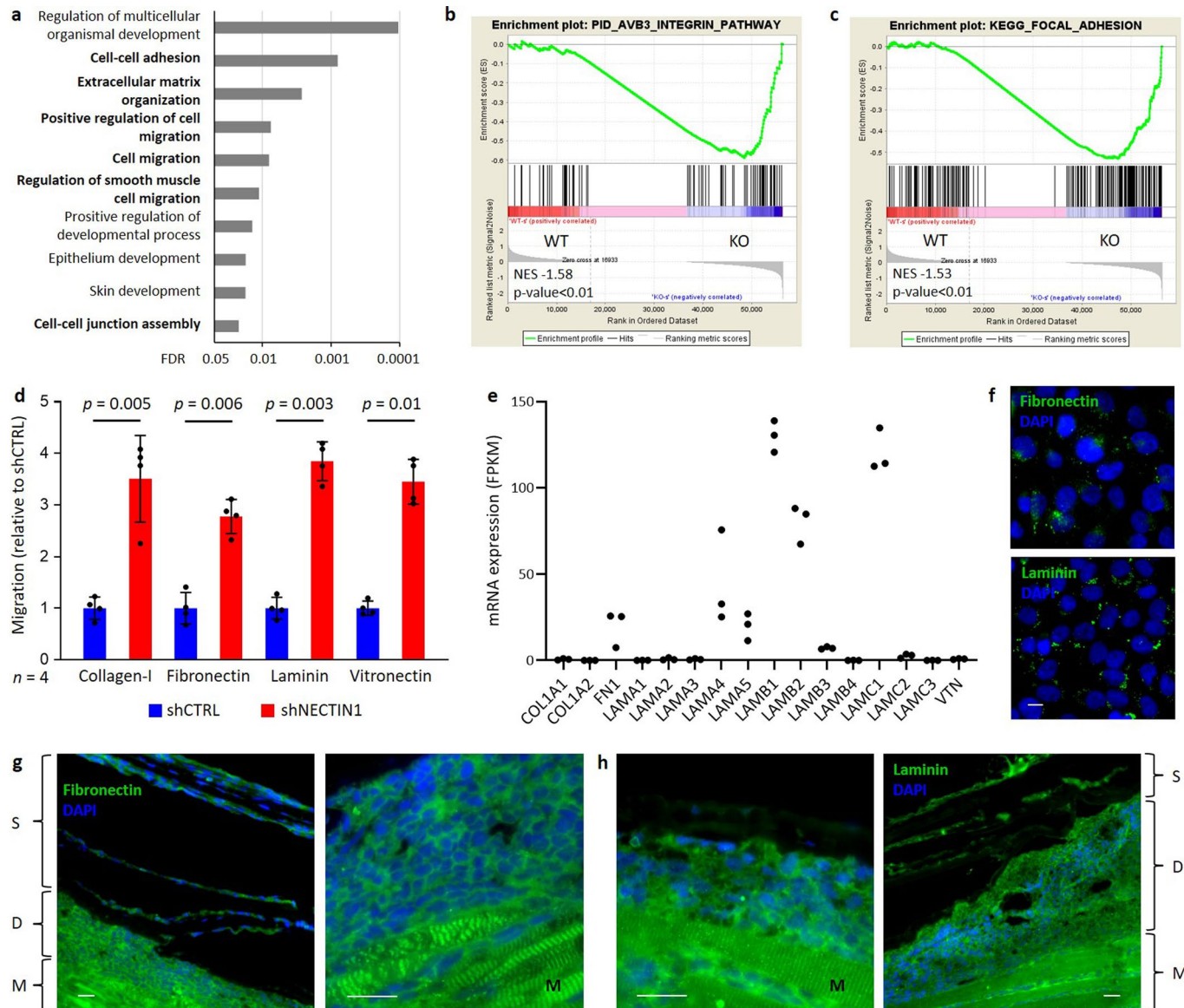

**Extended Data Fig. 6 | Adhesion and motility program expressed by *NECTIN1*-deficient cells.** (**a**) Most represented categories of biological processes in a Gene Ontology (GO) analysis of the set of genes more significantly differentially expressed between *NECTIN1*-knockout and *NECTIN1*-wild-type A375 human melanoma cells in response to 12 hours of serum starvation. The GO terms in bold correspond to adhesion and migration signatures. (**b-c**) Enrichment plots of Integrin and Focal adhesion gene signatures found by Gene Set Enrichment Analysis (GSEA) among the most significantly enriched pathways in *NECTIN1*-knockout cells compared to *NECTIN1*-wild-type cells in the A375 human melanoma line under serum starvation. FDR p-values were calculated by GSEA using a permutation test. (**d**) Migration of A375 human melanoma cells stably expressing an shRNA directed against *NECTIN1* (shNECTIN1) relative

to cells expressing a control shRNA (shCTRL) in transwells coated with the indicated Extracellular Matrix (ECM) proteins and after 12 hours of serum starvation. Cells were allowed to migrate for 6 hours. Mean ± SD of 4 independent experiments. Paired two-tailed t-test. (**e**) Expression levels of various ECM genes in 3 clones of A375 human melanoma cells, as measured by RNA-sequencing. (**f**) Immunofluorescence analysis of fibronectin or laminin (green) in A375 cells cultured in the absence of serum (FBS) for 12 hours. Scale bar: 10 µm. Representative of 3 independent experiments. (**g-h**) Immunofluorescence analysis of fibronectin (**g**) and laminin (**h**) in adult *casper* zebrafish transplanted with primary zebrafish melanoma cells. Scale bar: 20 µm. S: skin; D: dorsal cavity; M: muscle. Representative of 8 independent zebrafish tissue sections.

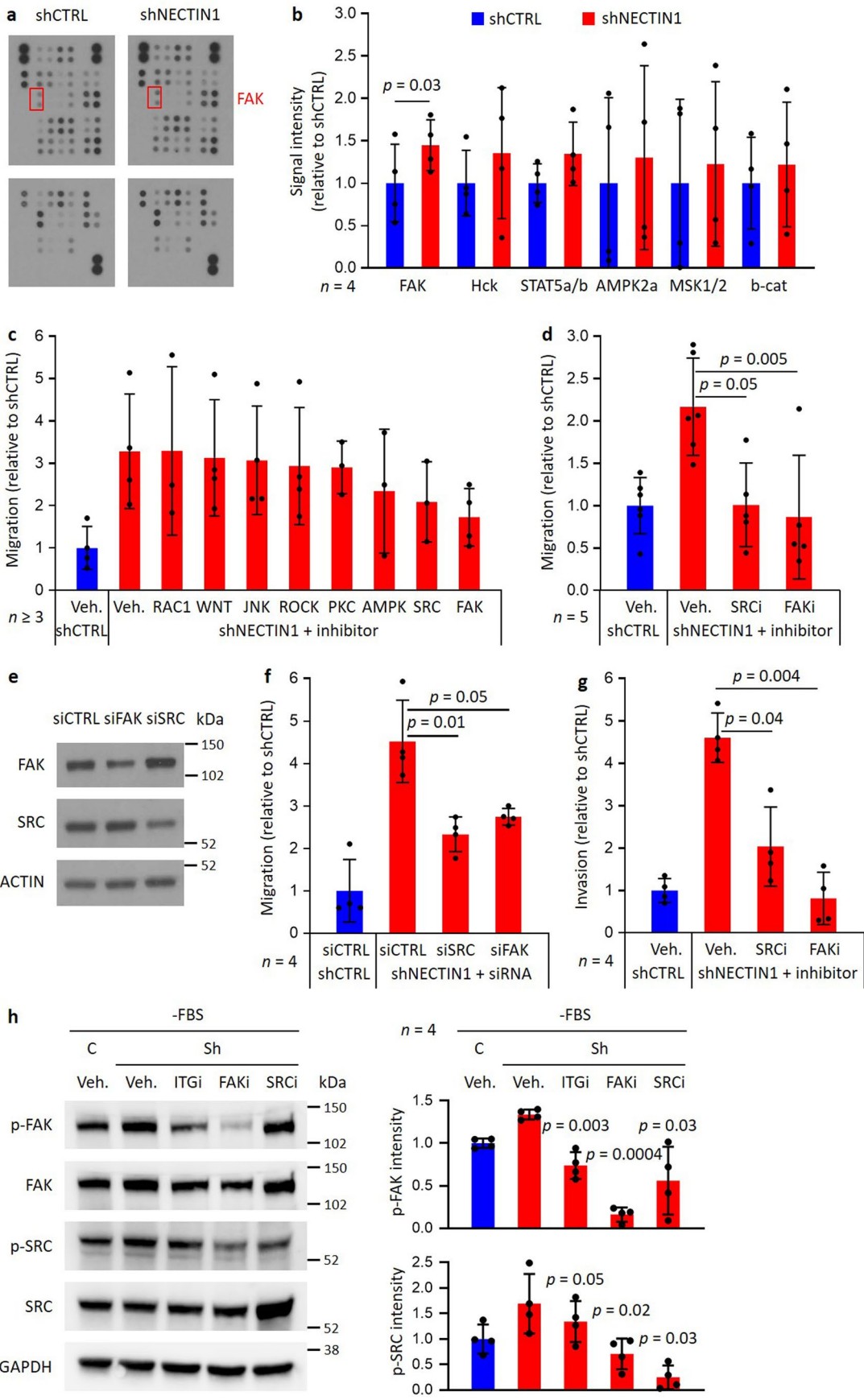

**Extended Data Fig. 7 | See next page for caption.**

**Extended Data Fig. 7 | Implication of a FAK/SRC axis in the migration of _NECTIN1_-deficient melanoma cells.** (**a**) Phospho-kinase array immunoblot of A375 cells stably expressing a control shRNA (shCTRL) or an shRNA against _NECTIN1_ (shNECTIN1) upon serum depletion. Boxes indicate phospho-FAK. Representative of 4 independent experiments. (**b**) Most differentially activated proteins between NECTIN1-proficient and deficient cells, as measured by phospho-kinase array immunoblot. Mean ± SD of 4 independent experiments. Paired two-tailed t-test. (**c**) Relative migration of the cells described in **a** in a transwell assay after 12 hours of serum starvation and upon treatment with a panel of inhibitors. Cells were allowed to migrate for 6 hours. Mean ± SD of 3 independent experiments. (**d**) Relative migration of cells treated with a FAK inhibitor (FAKi, GSK2256098) or a SRC inhibitor (SRCi, bosutinib) as described in **c**. Mean ± SD of 5 independent experiments. Paired two-tailed t-test. (**e**) Western blot analysis of FAK and SRC in A375 cells transfected with a control siRNA (siCTRL) or siRNAs targeting _FAK_ (siFAK) or _SRC_ (siSRC). Representative example of 4 independent experiments. (**f**) Relative migration of the cells described in **e** in the conditions described in **c**. Mean ± SD of 4 independent experiments. Paired two-tailed t-test. (**g**) Invasion through matrigel of cells treated with a FAK inhibitor (FAKi, PF562271) or a SRC inhibitor (SRCi, dasatinib) as described in **c**. Cells were allowed to invade for 10 hours. Mean ± SD of 4 independent experiments. Paired two-tailed t-test. (**h**) Western blot analysis of FAK and SRC phosphorylation in the cells described in **a** serum-starved for 12 hours and treated with an integrin inhibitor (ITGi, echistatin), a FAK inhibitor (FAKi, PF562271) or a SRC inhibitor (SRCi, dasatinib) for 6 hours. C: shCTRL; Sh: shNECTIN1; veh: vehicle-treated. Representative example of 4 independent experiments. Signal intensity was normalized to GAPDH. Mean ± SD. Paired two-tailed t-test. P-values are shown for comparisons to vehicle-treated shNECTIN1.

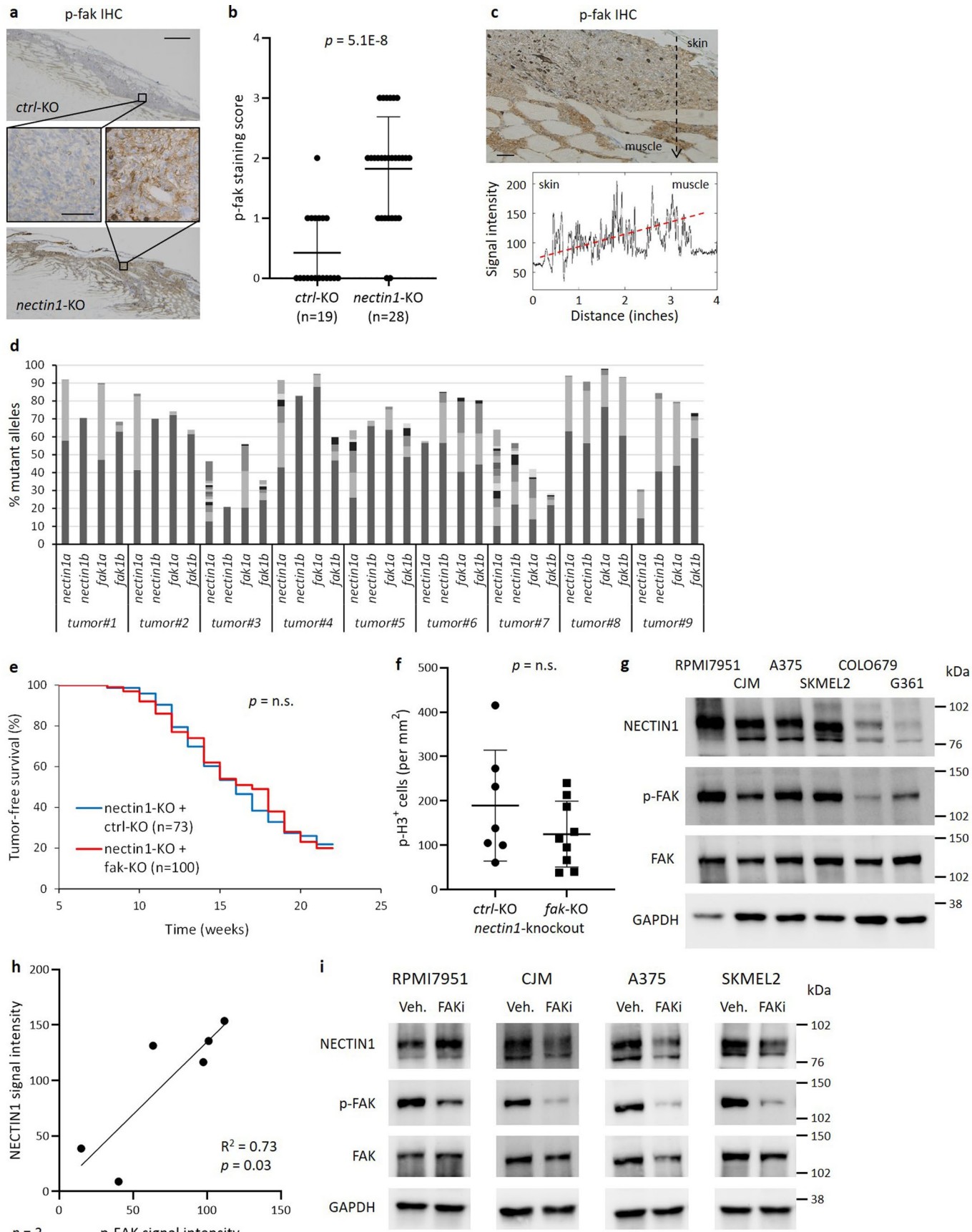

**Extended Data Fig. 8 | See next page for caption.**

**Extended Data Fig. 8 | Impact of FAK signaling in *NECTIN1*-deficient melanoma *in vivo*.** (**a**) Representative images of sections of *nectin1*-wildype or *nectin1*-knockout zebrafish melanomas transplanted into adult *casper* recipients, stained for phospho-fak by immunohistochemistry. Scale bars: 50 and 500 μm. (**b**) Scoring of overall phospho-fak staining intensity in secondary *casper* recipients transplanted with *nectin1*-wildype (n = 19) or *nectin1*-knockout (n = 28) zebrafish melanomas. Mean ± SD. Two-tailed *t*-test. (**c**) Top: section of adult *casper* recipient transplanted with *nectin1*-knockout zebrafish melanoma cells, stained for phospho-fak by immunohistochemistry. Orientation from skin to muscle is indicated. Scale bar: 50 μm. Representative example of 10 independent zebrafish tissue sections displaying a gradient of phospho-fak staining. Bottom: histogram representing phospho-fak signal intensity along the black dotted line on the image shown on top. The red dotted line materializes the spatial increase in fak activity. (**d**) Proportion of *nectin1a*, *nectin1b*, *fak1a*, and *fak1b* mutant alleles in 9 primary zebrafish melanomas expressing CRISPR vectors targeting these genes. Each section of each bar represents a different mutant allele. (**e**) Tumor-free survival curves of *Tg(mitf:BRAF^{V600E});tp53^{−/−}* zebrafish injected with vectors targeting either *nectin1* and a control gene, or *nectin1* and *fak*. Pooled data of 2 independent experiments. Log-rank test. (**f**) Quantification of overall p-H3 staining intensity in 7 *nectin1/ctrl*-knockout and 9 *nectin1/fak*-knockout primary zebrafish melanomas. Mean ± SD. Two-tailed t-test. (**g**) Western Blot analysis of NECTIN1 and phospho-FAK in 6 human melanoma cell lines. Representative example of 3 independent experiments. (**h**) Correlation plot of NECTIN1 and phospho-FAK levels in 6 human melanoma cell lines. Average values of 3 independent experiments. Pearson's correlation: $R^2 = 0.73$, p = 0.03. (**i**) Western Blot analysis of NECTIN1 and phospho-FAK in 4 human melanoma cell lines treated with vehicle (Veh.) or a FAK inhibitor (FAKi, PF562271) for 12 hours. Representative examples of 3 independent experiments.

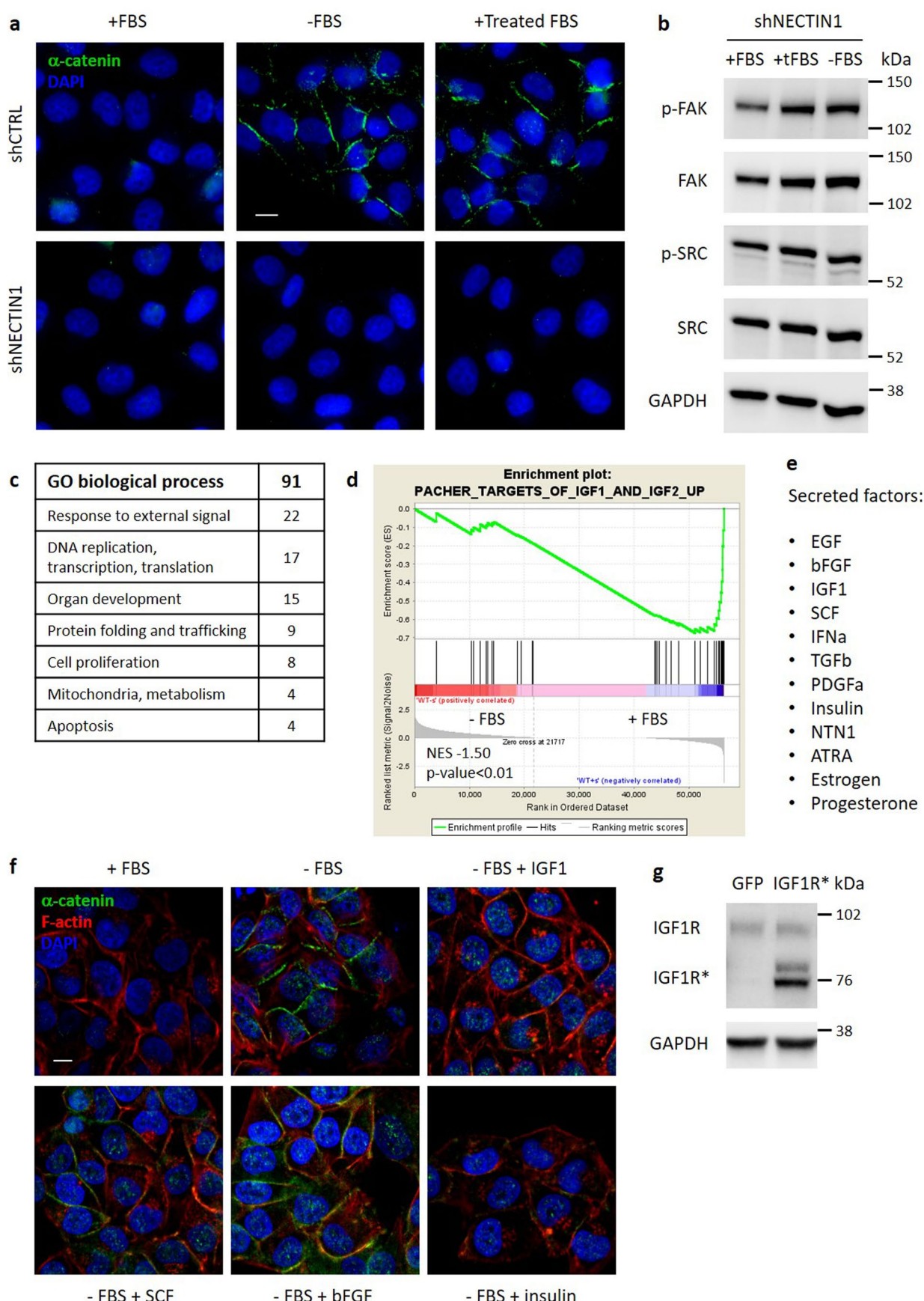

**Extended Data Fig. 9 | See next page for caption.**

**Extended Data Fig. 9 | Implication of IGF1 in the regulation of adherens junctions between melanoma cells.** (**a**) Immunofluorescence analysis of α-catenin (green) in A375 human melanoma cells stably expressing a control shRNA (shCTRL) or an shRNA against *NECTIN1* (shNECTIN1) and cultured in the presence or absence of serum (FBS), or in the presence of dextran-treated/charcoal-stripped serum (treated FBS) for 12 hours. Scale bar: 10 μm. Representative of 4 independent experiments. (**b**) Western Blot analysis of FAK and SRC phosphorylation levels in the cells described in **a** cultured in the presence or absence of serum (+FBS or -FBS), or in the presence of dextran-treated/charcoal-stripped serum (tFBS) for 12 hours. Representative example of 3 independent experiments. (**c**) Most represented categories of biological processes in a Gene Ontology (GO) analysis of the transcriptional response to serum starvation in A375 human melanoma cells. The right column indicates the number of significant GO terms assigned to each category. (**d**) Enrichment plot of an IGF1/IGF2 signature found among the most significantly enriched pathways in A375 cells cultured in the presence vs absence of serum by GSEA. The FDR p-value was calculated by GSEA using a permutation test. (**e**) List of secreted factors tested for their ability to prevent the formation of adherens junctions between melanoma cells upon serum withdrawal. (**f**) Immunofluorescence analysis of α-catenin (green) and F-actin (red) in A375 cells cultured in the presence or absence of serum (FBS), complemented or not with various growth factors, for 12 hours. Scale bar: 10 μm. Representative of 3 independent experiments. (**g**) Western Blot analysis of IGF1R in A375 cells stably transduced with a doxycycline-inducible vector expressing either GFP or a constitutively active form of IGF1R (CD8-IGF1R, IGF1R*) and treated with doxycycline for 12 hours. CD8-IGF1R (estimated size: 70 kDa) appears below endogenous IGF1R (estimated size: 95 kDa). Representative example of 3 independent experiments.

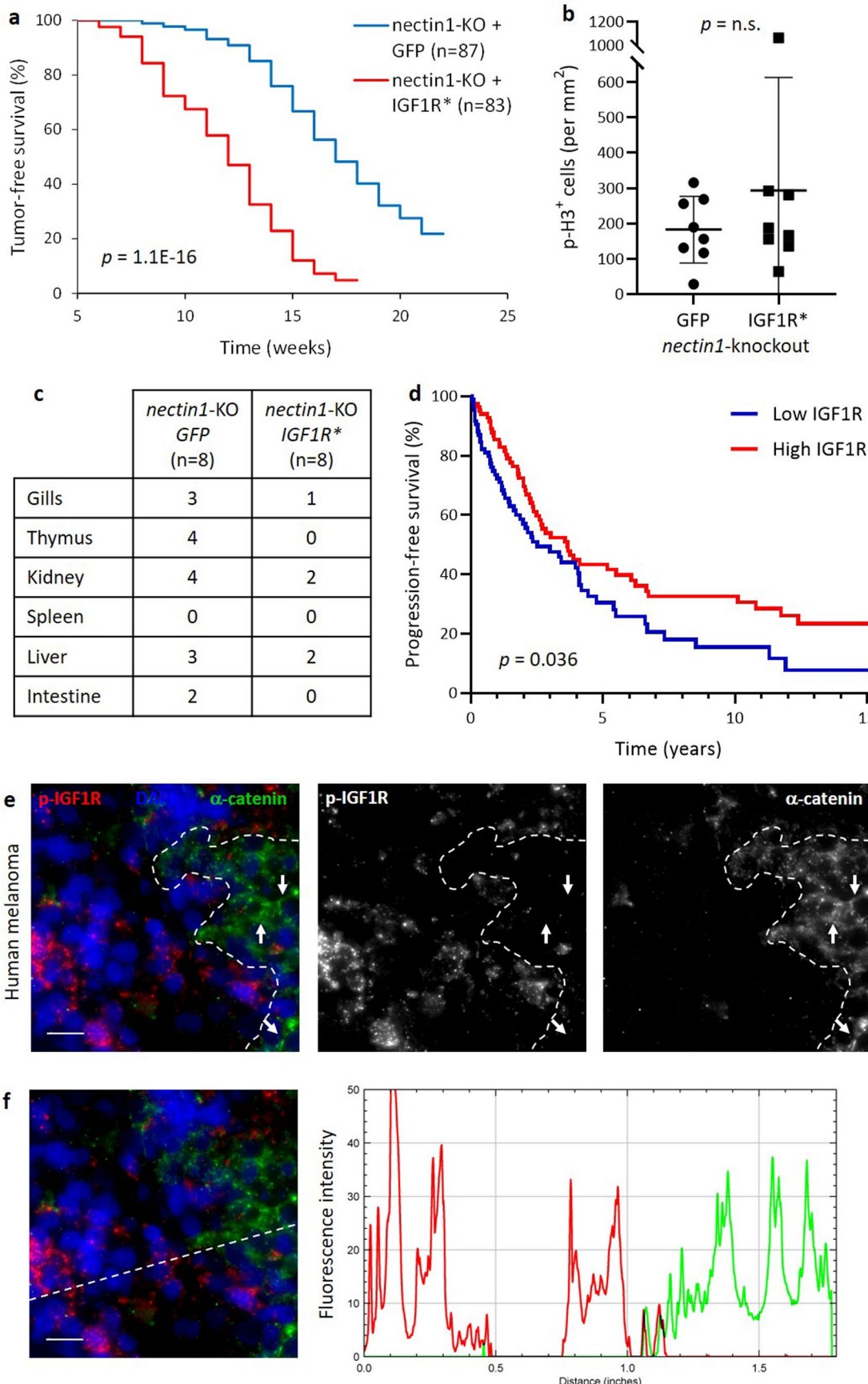

**Extended Data Fig. 10 | See next page for caption.**

**Extended Data Fig. 10 | Role of IGF1 signaling in zebrafish and human melanoma.** (**a**) Tumor-free survival curves of *Tg(mitf:BRAF^{V600E});tp53^{−/−}* zebrafish injected with a vector targeting *nectin1* and vectors expressing either GFP or CD8-IGF1R (IGF1R*). Pooled data of 2 independent experiments. Log-rank test. (**b**) Quantification of overall p-H3 staining intensity in 8 *nectin1*-knockout/GFP and 8 *nectin1*-knockout/IGF1R* primary zebrafish melanomas. Mean ± SD. Two-tailed t-test. (**c**) Quantification of organ involvement in 16-week-old zebrafish bearing *nectin1*-knockout/GFP (n = 8) or *nectin1*-knockout/IGF1R* (n = 8) primary melanoma. Two-tailed t-test. (**d**) Progression-free survival curves of patients with melanoma stratified according to IGF1R mRNA expression (top and bottom n = 104 samples). TCGA cohort. Log-rank test. (**e**) Immunofluorescence analysis of α-catenin (green) and phospho-IGF1R (red) on a tissue section of human melanoma. Scale bar: 10 μm. Dotted line demarcates areas of high and low IGF1 signaling. Arrows point to adherens junctions. Representative example of 9 independent tumors. (**f**) Histogram representing signal intensities of α-catenin (green) and phospho-IGF1R (red) along the dotted line in the human melanoma tissue section shown on the left. Note the mutual exclusivity of the two signals.

# Reporting Summary

## Statistics

For all statistical analyses, confirm that the following items are present in the figure legend, table legend, main text, or Methods section.

| n/a | Confirmed | |
|-----|-----------|---|
| ☐ | ☒ | The exact sample size (*n*) for each experimental group/condition, given as a discrete number and unit of measurement |
| ☐ | ☒ | A statement on whether measurements were taken from distinct samples or whether the same sample was measured repeatedly |
| ☐ | ☒ | The statistical test(s) used AND whether they are one- or two-sided *Only common tests should be described solely by name; describe more complex techniques in the Methods section.* |
| ☒ | ☐ | A description of all covariates tested |
| ☐ | ☒ | A description of any assumptions or corrections, such as tests of normality and adjustment for multiple comparisons |
| ☐ | ☒ | A full description of the statistical parameters including central tendency (e.g. means) or other basic estimates (e.g. regression coefficient) AND variation (e.g. standard deviation) or associated estimates of uncertainty (e.g. confidence intervals) |
| ☐ | ☒ | For null hypothesis testing, the test statistic (e.g. *F*, *t*, *r*) with confidence intervals, effect sizes, degrees of freedom and *P* value noted *Give P values as exact values whenever suitable.* |
| ☒ | ☐ | For Bayesian analysis, information on the choice of priors and Markov chain Monte Carlo settings |
| ☒ | ☐ | For hierarchical and complex designs, identification of the appropriate level for tests and full reporting of outcomes |
| ☐ | ☒ | Estimates of effect sizes (e.g. Cohen's *d*, Pearson's *r*), indicating how they were calculated |

*Our web collection on statistics for biologists contains articles on many of the points above.*

## Software and code

Policy information about availability of computer code

| Data collection | No software was used for data collection |
|-----------------|-------------------------------------------|
| Data analysis | FastQC version 0.11.9, Cutadapt version 2.0, Tophat version 2.0.11, Cufflinks version 2.2.1, GISTIC 2.0, Gene Ontology (GO Ontology database Released 2019-02-02) using PANTHER Overrepresentation Test (Released 20190308), Gene Set Enrichment Analysis GSEA 3.0 (released July 2017) using MSigDB version 6.2 (released July 2018), ImageJ version 1.51 (released 23 April 2018), and GraphPad Prism 9 softwares were used for data analysis. No custom code or software was use for any aspect of data processing or analysis. |

For manuscripts utilizing custom algorithms or software that are central to the research but not yet described in published literature, software must be made available to editors and reviewers. We strongly encourage code deposition in a community repository (e.g. GitHub). See the Nature Portfolio guidelines for submitting code & software for further information.

## Data

Policy information about availability of data

All manuscripts must include a data availability statement. This statement should provide the following information, where applicable:

- Accession codes, unique identifiers, or web links for publicly available datasets
- A description of any restrictions on data availability
- For clinical datasets or third party data, please ensure that the statement adheres to our policy

The Cancer Genome Atlas (TCGA) dataset for cutaneous melanoma used in this study is available at https://gdc.cancer.gov/about-data/publications/pancanatlas. Copy-number analyses were performed with GISTIC 2.0 through the interface of the Broad Institute (https://portals.broadinstitute.org/tcga/home), using 2014-04-28 stddata release.
The RNA sequencing data of the consequences of NECTIN1 loss and the effect of serum depletion on A375 human melanoma cells generated during this study have

been deposited in GEO under the accession code GSE174150 and are publicly available.

The tandem mass spectrometry data of the cell-surface proteins of NECTIN1-deficient and NECTIN1-proficient A375 human melanoma cells upon serum depletion generated during this study have been deposited in MassIVE (Center for Computational Mass Spectrometry, UCSD) under the accession code MSV000090170 and are publicly available.

# Field-specific reporting

Please select the one below that is the best fit for your research. If you are not sure, read the appropriate sections before making your selection.

☒ Life sciences ☐ Behavioural & social sciences ☐ Ecological, evolutionary & environmental sciences

For a reference copy of the document with all sections, see nature.com/documents/nr-reporting-summary-flat.pdf

# Life sciences study design

All studies must disclose on these points even when the disclosure is negative.

| Sample size | For cell line work, no calculation was performed to predetermine sample size; experiments were repeated four times to ensure robustness, unless statistical significance could be reached with 3 replicates only. For zebrafish transplantation experiments, a target sample size of 16 primary tumors per arm was calculated such that 2 fold differences in tumor spreading could be observed with a power of 80% and with statistical significance (5% type I error) using two-sided student's t-test. This sample size was never attained because statistical significance was reached with smaller samples due to the larger size of the observed effects. For human melanoma sections, sample size was limited by availability. |
|---|---|
| Data exclusions | No data were excluded from the analyses. |
| Replication | All experiments were independently repeated at least times and the aggregated data reported in the manuscript include all replicates. All attempts at replication were successful for experiments presented in the manuscript and that showed statistically significant differences. |
| Randomization | For injection and transplantation experiments in zebrafish, embryos and secondary recipients were randomly allocated to each genetic condition. For cell culture experiments implicating genetic or pharmacological modifications, cells were randomly distributed between the different experimental conditions. Randomization was irrelevant to all other experiments, which did not involve allocation of samples or organisms to different experimental groups. |
| Blinding | Data collection was performed blinded in the case of tumor-initiation and transplantation experiments in zebrafish, as well as for immunofluorescence, immunohistochemistry and H&E staining of human or zebrafish tissue-sections. In the case of cell culture experiments, blinding was performed whenever possible, depending on research personnel availability. Blinding was only lifted at the time of data analysis. |

# Reporting for specific materials, systems and methods

We require information from authors about some types of materials, experimental systems and methods used in many studies. Here, indicate whether each material, system or method listed is relevant to your study. If you are not sure if a list item applies to your research, read the appropriate section before selecting a response.

## Materials & experimental systems

| n/a | Involved in the study |
|---|---|
| ☐ | ☒ Antibodies |
| ☐ | ☒ Eukaryotic cell lines |
| ☒ | ☐ Palaeontology and archaeology |
| ☐ | ☒ Animals and other organisms |
| ☐ | ☒ Human research participants |
| ☒ | ☐ Clinical data |
| ☒ | ☐ Dual use research of concern |

## Methods

| n/a | Involved in the study |
|---|---|
| ☒ | ☐ ChIP-seq |
| ☒ | ☐ Flow cytometry |
| ☒ | ☐ MRI-based neuroimaging |

# Antibodies

| Antibodies used | All antibodies used in this study are reported in the Methods section.<br>NECTIN1 primary antibody (HPA026846, Sigma-Aldrich) , dilution 1:50 for IHC, 1:200 for IF<br>goat-anti-rabbit HRP secondary antibody (PI-1000, Vector Laboratories) , dilution 1:200 for IHC<br>Phospho-Histone3 (Ser10) primary antibody (#9701, Cell Signaling Technology), dilution 1:200 for IHC<br>Phospho-FAK (Tyr397) primary antibody (44-624G, Invitrogen), dilution 1:200 for IHC<br>α-E-Catenin primary antibody (#3236, Cell Signaling Technology), dilution 1:200 for IF<br>N-Cadherin primary antibody (#13116, Cell Signaling Technology), dilution 1:200 for IF<br>α-Actinin primary antibody (#69758, Cell Signaling Technology), dilution 1:100 for IF<br>α-Catenin primary antibody (13-9700, Invitrogen), dilution 1:200 for IF<br>Phospho-IGF1R beta (Tyr1161) primary antibody (PA5-37601, Invitrogen), dilution 1:200 for IF |
|---|---|

E-Cadherin (4A2) primary antibody (#14472, Cell Signaling Technology), dilution 1:200 for IF
Alexa Fluor 488 goat anti-rabbit secondary antibody (A11008, Life Technologies), dilution 1:400 for IF
Alexa Fluor 568 goat anti-mouse secondary antibody (A11004, Life Technologies), dilution 1:400 for IF
Alexa Fluor Plus 488 Goat anti-Mouse secondary antibody (A32723, Life Technologies), dilution 1:400 for IF
Alexa Fluor 568 Goat anti-Rabbit secondary antibody (A11036, Life Technologies), dilution 1:400 for IF
NECTIN1 primary antibody (ab66985, Abcam), dilution 1:1000 for WB
phospho-FAK primary antibody (#3283, Cell Signaling Technology), dilution 1:1000 for WB
FAK primary antibody (#71433, Cell Signaling Technology), dilution 1:1000 for WB
phospho-SRC primary antibody (#2101, Cell Signaling Technology), dilution 1:1000 for WB
SRC primary antibody (#2109, Cell Signaling Technology), dilution 1:1000 for WB
ACTIN primary antibody (A2228, Sigma-Aldrich), dilution 1:5000 for WB
anti-mouse HRP (#7076, Cell Signaling Technology), dilution 1:2000 for WB
anti-rabbit HRP (#7074, Cell Signaling Technology), dilution 1:2000 for WB
Fibronectin (#26836, Cell Signaling Technology), dilution 1:400 for IF
Laminin (L9393, Sigma-Aldrich), dilution 1:100 for IF
FNDC3A (PA5-109309, Invitrogen) , dilution 1:200 for IF
Integrin β1 (#9699, Cell Signaling Technology), dilution 1:1000 for WB
Integrin β2 (#73663, Cell Signaling Technology), dilution 1:1000 for WB
Integrin β3 (#13166, Cell Signaling Technology), dilution 1:1000 for WB
Integrin β4 (#14803, Cell Signaling Technology), dilution 1:1000 for WB
Integrin β5 (#3629, Cell Signaling Technology), dilution 1:1000 for WB
GAPDH (ab8245, Abcam), dilution 1:1000 for WB
CD49f (Integrin alpha 6) (14-0495-85, eBioscience), 40ug/mL for inhibition

| Validation | All Cell Signaling Technology and Invitrogen antibodies were validated for use in human samples by the companies, as described on the suppliers' websites. NECTIN1 primary antibody is a Prestige Antibody developed and validated by the Human Protein Atlas (HPA) project and characterization data is accessible via the Human Protein Atlas portal. We validated NECTIN1 primary antibody (ab66985, Abcam) by knockdown and overexpression experiments as shown in Extended Data Fig. 3g. We validated Phospho-FAK (Tyr397) primary antibody (44-624G, Invitrogen) for use in zebrafish by comparing IHC staining in fak wild-type and knockout zebrafish tumors. Laminin (L9393, Sigma-Aldrich) and FNDC3A (PA5-109309, Invitrogen) primary antibodies were validated by other groups for use in zebrafish as documented on the ZFIN website (https://zfin.org/ZDB-ATB-090304-3#summary and https://zfin.org/ZDB-ATB-200313-1#summary). |

## Eukaryotic cell lines

Policy information about cell lines

| Cell line source(s) | 293T, RPMI-7951, CJM, A375, SKMEL2, COLO679, and G-361 cell lines were purchased from commercial sources (ATCC CRL-3216, ATCC HTB-66, Creative Bioarray CSC-C6421J, ATCC CRL-1619, ATCC HTB-68, Sigma 87061210, and ATCC CRL-1424, respectively) |
| Authentication | The cell lines used in this study were purchased directly from vendors that certify their authenticity but we did not authenticate them ourselves. |
| Mycoplasma contamination | All cell lines were tested negative for mycoplasma contamination throughout the study. |
| Commonly misidentified lines (See ICLAC register) | No commonly misidentified cell lines were used in the study. |

## Animals and other organisms

Policy information about studies involving animals; ARRIVE guidelines recommended for reporting animal research

| Laboratory animals | Zebrafish (Danio rerio) of the casper (mitfa-/-, roy-/-) and Tg(mitfa:BRAFV600E), tp53-/-, mitfa-/- strains were used in this study. Both males and females were used between 3 and 6 months of age. |
| Wild animals | The study did not involve wild animals. |
| Field-collected samples | The study did not involve samples collected from the field. |
| Ethics oversight | Zebrafish were handled according to our vertebrate animal protocol that has been approved by Boston Children's Hospital Animal Care Committee. |

Note that full information on the approval of the study protocol must also be provided in the manuscript.

## Human research participants

Policy information about studies involving human research participants

| Population characteristics | Since all tissue samples used in this study were deidentified archived samples, no information about subjects was available. |
| Recruitment | Human melanoma tissue-sections were obtained from archived cases of the Brigham and Women's Hospital and patients were not recruited specifically for this study. |

| Ethics oversight | The human melanoma tissue-sections were used in this study with approval of the Institutional Review Board of Brigham and Women's Partners Human Research Committee, Harvard Medical School. |

Note that full information on the approval of the study protocol must also be provided in the manuscript.

