## [Peer Review File · Nature Genetics]

Peer Review Information

Manuscript Title: Loss of NECTIN1 triggers melanoma dissemination upon local IGF1 depletion

Corresponding author name(s): Dr Julien Albain

Reviewer Comments & Decisions:

Decision Letter, initial version:
--

15th Sep 2021

Dear Dr Zon,

Your Article, "Frequent deletion of adhesion gene NECTIN1 triggers melanoma dissemination upon local IGF1 depletion" has now been seen by 3 referees. You will see from their comments below that while they find your work of interest, some important points are raised. We are interested in the possibility of publishing your study in Nature Genetics, but would like to consider your response to these concerns in the form of a revised manuscript before we make a final decision on publication.

To guide the scope of the revisions, the editors discuss the referee reports in detail within the team, including with the chief editor, with a view to identifying key priorities that should be addressed in revision and sometimes overruling referee requests that are deemed beyond the scope of the current study. You'll see that all reviewers have asked for further methodological detail (including information about which zebrafish nectin1 paralog was used) and clarity around your figures. Considering the reports as a whole, we would encourage you to address all the reviewer concerns, experimentally where possible, or textually as a minimum. Please do not hesitate to get in touch if you would like to discuss these issues further.

We therefore invite you to revise your manuscript taking into account all reviewer and editor comments. Please highlight all changes in the manuscript text file. At this stage we will need you to upload a copy of the manuscript in MS Word .docx or similar editable format.

*1) Include a "Response to referees" document detailing, point-by-point, how you addressed each

referee comment. If no action was taken to address a point, you must provide a compelling argument. This response will be sent back to the referees along with the revised manuscript.

*2) If you have not done so already please begin to revise your manuscript so that it conforms to our Article format instructions, available

here.

*3) Include a revised version of any required Reporting Summary:

Please be aware of our guidelines on digital image standards.

[REDACTED]

We hope to receive your revised manuscript within six months. If you cannot send it within this time, please let us know.

Nature Genetics is committed to improving transparency in authorship. As part of our efforts in this direction, we are now requesting that all authors identified as 'corresponding author' on published papers create and link their Open Researcher and Contributor Identifier (ORCID) with their account on the Manuscript Tracking System (MTS), prior to acceptance. ORCID helps the scientific community achieve unambiguous attribution of all scholarly contributions. You can create and link your ORCID from the home page of the MTS by clicking on 'Modify my Springer Nature account'. For more information please visit please visit www.springernature.com/orcid.

Sincerely,

Safia Danovi
Editor
Nature Genetics

Referee expertise:

Referee #1: zebrafish models of cancer

Referee #2: melanoma metastasis

Referee #3: melanoma signaling

Reviewers' Comments:

Reviewer #1:
Remarks to the Author:
Summary:

The study by Ablain and colleagues identifies NECTIN1 as a new melanoma tumor suppressor gene involved in tumor cell migration. The authors use a combination of human cell-based models and zebrafish in vivo models to show that loss of NECTIN1 promotes melanoma cell migration in vitro and spreading in vivo. The study further shows that low IGF1 signaling promotes the migration of NECTIN1-deficient melanoma cells through activation of an Integrin/FAK/SRC-dependent mechanism that in turn down regulated adherens junction formation. The combination of human cell and patient samples with relevant animal modeling provides a compelling case that NECTIN1 is an important new melanoma tumor suppressor that may lead to a better understanding of how melanoma metastasizes, which is the leading cause of death in melanoma patients and a significant clinical challenge. Thus, the study has significant potential to impact the field and should appeal to a general audience that is appropriate for this journal. However, the study could be improved by including more details of the approaches/methods used, in vivo analysis of primary tumors and controls.

Suggestions:

- 1) Figure 1. The molecular nature (sequence) of the deep and shallow deletions in NECTIN1 are not fully described. Examples of shallow deletions should be included.
- 2) Figure 1. While overall survival of patients with NECTIN1 deletion is important, if possible, these data should also be analyzed to show if there are any significant differences in metastasis-free survival and/or melanoma stage, as it is expected that NECTIN1 deletions would be significantly over represented in advanced melanoma stages and/or metastasis.
- 3) Figure 2. There are two nectin1 paralogs in zebrafish, nectin1a and nectin1b. The authors do not make this distinction in the main text, which therefore glosses over the technical difficulties in performing the loss-of-function experiments in vivo, as 4 alleles are being targeted at once to generate the nectin1-KO melanomas. This brings up a number of issues. First, it is important to know if one or both nectin1 genes are actually expressed in primary zebrafish melanomas. Second, there

3are no details showing the mutational efficiency of Cas9/Crisper cutting using their system- how often are all 4 alleles mutated and are there differences in the mutational frequency in nectin1a/b in melanoma cells that spread versus those that remain at the dorsal injection site? Third, the conclusions in the zebrafish model would be strengthened by the addition of appropriate controls, such as other molecules involved in cell-cell adhesion (e.g., E-cadherin). In addition, the control-KO used is not well described in the main text, nor is the reasoning behind using this as a control. Such data would help to address the concern that knocking down any cell-cell adhesion molecule may cause melanoma cell spreading in the transplantation system.

4) Figure 2. Details of tumor cell behavior in primary tumors is lacking. Is there evidence of metastasis or localized cell spreading in the primary tumors, as the authors have previously demonstrated for other melanoma oncogenes in the zebrafish model?

5) Figure 4. The analysis of p-fak by IHC should be quantified and/or inclusion of a immunoblot added to quantify the extent to which p-fak is increased in nectin1-knockout tumors.

6) Figure 4. The authors are targeting 6 different alleles in panel 4h, which again is not fully described in the main text or methods. How efficient are the Cas9/gRNA's at targeting/mutating the expected alleles?

7) Figure 4 and 5. There are no details describing the impact of fak knockout or ectopic IGF1R expression in the primary tumors in vivo (see comments above). Do these tumors have differences in localized spreading compared to nectin1a/1b alone and/or evidence of dissemination to other organ sites? Inclusion of primary tumor cell behaviors would bolster the in vivo conclusions that currently rely on using a transplantation approach in irradiation/immune suppressed recipients which may also impact the behavior of these cells.

Reviewer #2:

Remarks to the Author:

In this manuscript, Ablain et al describe a frequent deletion of adhesion gene NECTIN1 found in melanomas. Then they go on to investigate if this deletion triggers melanoma dissemination upon local IGF1 depletion.

The part of the data related to melanoma genetics is very interesting since this gene has not been linked to melanoma progression before. The functional validation needs further work and therefore dampens overall enthusiasm.

General comment

This is a manuscript about melanoma dissemination. Part of this dissemination occurs via invasion into the dermis (complex matrix with a high content of collagen I). It is therefore surprising that authors perform all of their functional validation in transwell assays. While this is a valid model for understanding some aspects of dissemination (chemotaxis, confinement, nuclear deformation, etc), authors claim this mechanism works via cell-matrix adhesion signaling- via integrin signaling- but they never perform any validation in a matrix based migration-invasion type assay (they only do one adhesion assay to laminin). On the other hand, 2D migration and 3D migration mechanisms are quite different, therefore authors could have missed important signalling in the nectin-deficient type melanomas. 3D inverted invasion assays, spheroids grown and invading into collagen or even transwells with some type of matrix overlaid in them, could have been an alternative more convincing approach. For example, Figure 4c, there is no matrix in this transwell so why inhibition of integrins has

4an effect? It may be completely independent of cell-matrix interactions. In fact, many 3D- invasive programs rely on low levels of adhesion to the matrix- to achieve higher speed, so these connections between matrix-nectin and invasion have not been fully developed.

Specific points

-The list of integrins upregulated after nectin loss are : beta4 , alpha 5, beta 1 and alpha 6, why do they not use matrices for these receptors in their migration/invasion assays? (collagen I, fibronectin, laminin) Moreover why do they use an integrin inhibitor that targets other integrins -alpha V beta3 and alpha V beta 5 (SB273005) which would affect vitronectin adhesion. It suggests lack of a specific cell-matrix type mechanism. Maybe is more a mechanism related to the secretion of chemokines and that is why they see effects in a transwell with no matrix.

- In general, in all the figures, there is a lack of information on migration assays and western blots: how many times were the experiments done. There are no quantification of the western blots- and no statistical analysis of these western blots either.

- Migration assays (according to materials and methods): these assays were performed with different endpoints (6, 20, 24 h depending on cell line). If you represent all this data together, the assays need to have the same endpoint. On the same lines, all assays seem to be normalised to 1 when comparing different cell lines. Moreover, what are the migratory abilities of the control cells (their basal migration) without normalising to 1? Does it relate to their endogenous nectin levels? For that again, same endpoint assays need to be chosen. Can authors rescue their effects on migration with an RNAi resistant version of nectin?

-When looking for the possible FAK related mechanism they base their directions in two data sets:

1. a protein phospho-array from which they chose FAK, while there is only a very modest fold increase in phosphorylation (1.2 if I understood this data correctly). This array does not seem to have produced optimal fold changes. Can the authors show original data, the dot blot? You would ideally want a phospho-change of at least 2-fold.

2. When screening different inhibitors (rac, wnt, jnk inhibitors) in the migration assay (Ext Data 5e), authors argue they use a concentration that does not affect viability of cells. Many of the migration-related pathways also drive the survival of cancer cells, therefore how can they make sure the drugs are hitting the target and they are not just using a very low concentration that only mildly reduces pathway activity? Authors should confirm by Western blot that downstream signalling is affected after drug treatment at the concentrations they have chosen of all those drugs. Therefore, authors may not have chosen the best targets (or at least not the only possible targets). In their OMICs there are many more changes taking place that could be relevant for the migration of the nectin deficient melanomas. In fact, that FAK is required for 2D migration, is not a surprise but what happens in 3D migration and how matrix is connected to nectin downstream signalling is the key question that has not been addressed. In those lines, in vivo staining of p-FAK, are they more prominent in the invasive areas within the tumours? what is the matrix found there? Can they correlate in vivo invasion to FAK activity levels? In the panel of 6 melanoma lines is there an anti-correlation between nectin levels and active FAK?

-The data on nectin +/- tumours being softer is interesting but authors have not linked this feature to anything, if anything stiffer tumours have been linked to worse prognosis. It is not clear that this piece of data adds to their argument, unless authors look at the cytoskeleton of the cancer cells properly.

With regards to that, all the imaging performed in vitro needs improvement. Authors say there is a cell shape change but it is really difficult to see in the presented images (specially F-actin). Higher resolution and higher quality imaging would help. Moreover, quantitative image analysis of both Actin

structures and of AJ needs to be performed, not just show one picture.

-Regarding the IGF connections. If IGF1 induces junctions, what is the effect of adding IGF (or an inhibitor) in the migration assay? Does it reduce it? Other labs have seen a pro-migratory effect of this factor in melanoma. Also, can authors show the level of over expression of IGFR*, I could not find it. Data in figure 6, How do the levels of IGF/IGFR relate to any prognostic feature of melanomas? In those lines where the nectin high melanomas more proliferative (high ki67), and the IGF high?

Minor points

-in western blots if you are altering cytoskeleton, you should use other loading control-not Actin

- E-cadherin to N-cadherin switch has been linked to melanoma invasion, but authors say that N-cadherin is downregulated in the migratory process they describe, can authors discuss this apparent discrepancy?

Reviewer #3:

Remarks to the Author:

In this study, the authors investigated the role of NECTIN1 loss, which was found to occur in 55% of human melanoma cases in TCGA, in melanoma metastasis. NECTIN1 loss seemed to occur predominantly in metastatic melanoma lesions and was found to be associated with worse overall survival. Knockout of NECTIN1 in a zebrafish model was found to promote tumor spread and was associated with greater tumor elasticity. In vitro, NECTIN1 knockdown was found to promote a migratory phenotype in serum-free conditions despite slower proliferation of the NECTIN1 knockdown cells. The increase in migratory capacity was dependent on an increase in Integrin/SRC/FAK signaling, diminished focal adhesion formation and was shown to occur in the context of low IGF1. Knockout of IGFR in the NECTIN1-knockout zebrafish blocked tumor spreading capacity of the cells. Finally, the authors show that NECTIN negative melanomas typically lack focal adhesions and that focal adhesions typically occur in NECTIN1 positive melanomas, in areas of low IGF1.

The authors present a very compelling study of great significance to the melanoma research community showing a novel genetic driver of melanoma metastasis. Overall, the study is thorough, well conducted, and mechanistically strong. The manuscript is written well, is very clear and provides the appropriate context and references. There are a few minor concerns.

1. The figures could be improved for clarity. The details in Figure 1b are difficult to see and several figures (for example Figure 1 and Figure 3) have panels which appear in random order (panel order should follow the left to right, top to bottom sequence).

2. Could the authors clarify how heterogeneous NECTIN1 was scored in the human melanoma TMA? They mentioned in the manuscript that some tissues (mostly primary) had areas of high and low expression, so what numerical score would this be? Mainly, how does the staining scores depicted in Figure 1d relate to the Extended Figure 1c images (which score would each example have)?

3. For the patients whose survival was measured in Figure 1e, are these the primary tissues or metastatic tissues or both? If NECTIN1 deletion is assessed in primary tumors only, would you see similar results in regards to survival (ie is NECTIN1 loss in primary tumors prognostic of worse

6survival)?

4. The spheroid experiments are not especially convincing. I would argue that based on the images, the shCTRL also don't really form spheroids without FBS (they look very flat compared to typical spheroids. It may be helpful to quantify these in some way.

5. Please include the +FBS control for NECTIN knockdown in Extended data Figure 4D.

6. The western blot for pSRC in Extended Data Fig. 6f could use more clarity, the decrease in pSRC with integrin inhibition is not obvious or striking. Please include quantification of key western blots.

7. Does dextran-treated charcoal-stripped serum also fail to activate adherens junctions and instead increase integrin/FAK/SRC signaling in NECTIN1-shRNA cells like serum-free media does?

8. It may help to have a summary diagram of the presented mechanism moved to the main Figure 7 as opposed to the Extended Data Figure 8.

Author Rebuttal to Initial comments

Reviewers' Comments:

Reviewer #1:

Remarks to the Author:

Summary:

The study by Ablain and colleagues identifies NECTIN1 as a new melanoma tumor suppressor gene involved in tumor cell migration. The authors use a combination of human cell-based models and zebrafish in vivo models to show that loss of NECTIN1 promotes melanoma cell migration in vitro and spreading in vivo. The study further shows that low IGF1 signaling promotes the migration of NECTIN1-deficient melanoma cells through activation of an Integrin/FAK/SRC-dependent mechanism that in turn down regulated adherens junction formation. The combination of human cell and patient samples with relevant animal modeling provides a compelling case that NECTIN1 is an important new melanoma tumor suppressor that may lead to a better understanding of how melanoma metastasizes, which is the leading cause of death in melanoma patients and a significant clinical challenge. Thus, the study has significant potential to impact the field and should appeal to a general audience that is appropriate for this journal. However, the study could be improved by including more details of the approaches/methods used, in vivo analysis of primary tumors and controls.

We thank the reviewer for their positive assessment of our work. In our revised manuscript, we have significantly expanded the characterization of the tissue-specific CRISPR targeting system that we used to generate primary *nectin1*-knockout melanomas in adult zebrafish. We are now providing a comprehensive description of the nature and frequency of the mutant alleles found in these tumors by deep-sequencing, as well as evidence supporting a more prominent role of the *nectin1b* paralog compared to *nectin1a* (Figure 2). We have also added a thorough analysis of the histopathological features of all genetically-defined melanomas generated in our study. In particular, this analysis revealed a greater dissemination of primary *nectin1*-knockout tumors in adult zebrafish (Figure 2), an effect reversed by genetic inactivation of *fak* (Figure 4). These data demonstrate an extremely high CRISPR targeting efficiency in zebrafish melanoma and consolidate our *in vivo* conclusions regarding the role of *NECTIN1* loss and FAK activity in tumor dissemination.

Suggestions:

1) Figure 1. The molecular nature (sequence) of the deep and shallow deletions in *NECTIN1* are not fully described. Examples of shallow deletions should be included.

Figure 1b showed copy-number alterations encompassing the *NECTIN1* locus in cutaneous melanoma. We are now providing a more focused view of deep and shallow deletions affecting the *NECTIN1* gene and present in 55% of human cutaneous melanomas in Extended Data Fig. 1c. We are also showing 3 examples of shallow deletions containing part or all of the *NECTIN1* coding sequence in Extended Data Fig. 1d.

(Extended Data Figure 1c) Linear copy-number in 202 human melanoma samples exhibiting deep or shallow deletions of the *NECTIN1* locus. (Extended Data Figure 1d) Sequences of shallow deletions affecting *NECTIN1* in three examples of human melanoma samples. In TCGA-D9-A148 (Patient 1) and TCGA-FS-A1ZQ (Patient 2), the deletion encompasses the entire *NECTIN1* gene, while in TCGA-FW-A3TU (Patient 3), the deletion starts in exon 2.

2) Figure 1. While overall survival of patients with *NECTIN1* deletion is important, if possible, these data should also be analyzed to show if there are any significant differences in metastasis-free survival and/or melanoma stage, as it is expected that *NECTIN1* deletions would be significantly over represented in advanced melanoma stages and/or metastasis.

NECTIN1 is deleted in over half of the melanoma samples of the TCGA cohort, which makes it difficult to detect significant associations with tumor stage. Unfortunately, the TCGA data for melanoma does not contain information about metastasis-free survival and a large fraction of samples actually come from lymph node metastases. We have, however, looked at *NECTIN1* copy-number variations across disease stages in this cohort. Although samples with low *NECTIN1* copy-number appeared enriched in advanced melanoma (stages III and IV), this observation was not statistically significant (see figure for reviewers only below).

NECTIN1 copy-number values across disease stages in samples from the TCGA cohort of cutaneous melanoma (n = 329). Bar: median.

3) Figure 2. There are two *nectin1* paralogs in zebrafish, *nectin1a* and *nectin1b*. The authors do not make this distinction in the main text, which therefore glosses over the technical difficulties in performing the loss-of-function experiments in vivo, as 4 alleles are being targeted at once to generate the *nectin1*-KO melanomas.

We initially did not distinguish between *nectin1a* and *nectin1b* and only referred to “nectin1” in order to simplify the presentation of the zebrafish experiments, although we mentioned targeting both paralogs in the Methods section. We agree, however, that this point deserves clarification and we have now added details in both the main text and Methods section.

This brings up a number of issues. First, it is important to know if one or both *nectin1* genes are actually expressed in primary zebrafish melanomas.

Nectin1b is the paralog expressed in zebrafish melanoma. We now provide in Extended Data Fig. 3a the FPKM values for *nectin1a* and *nectin1b* expression obtained by RNA-sequencing in 10 primary zebrafish melanomas. We nonetheless decided to target both paralogs in zebrafish tumors using our tissue-specific CRISPR system to prevent potential compensation.

(Extended Data Figure 3a) Box-and-whisker plot representing the expression levels of *nectin1a* and *nectin1b* in 10 primary zebrafish melanomas. Min-max, 1st and 4th quartiles, median, n=10. Two-tailed t-test.

Second, there are no details showing the mutational efficiency of Cas9/Crisper cutting using their system- how often are all 4 alleles mutated?

We are now showing in new Figure 2b the next-generation sequencing results of *nectin1a* and *nectin1b* CRISPR loci in 65 individual primary zebrafish melanomas. Overall, the average and median proportion of mutant alleles were 65.0 and 77.7 for *nectin1a*, and 75.9 and 85.2 for *nectin1b*, respectively. Given that not all cells are melanoma cells in a tumor, we can consider that a proportion of mutant alleles greater than 75% likely corresponds to complete targeting, while a proportion comprised between 25 and 75% likely reflects partial gene inactivation (possibly mimicking heterozygous loss). We found 52% of tumors (34/65) with near-complete targeting (> 75% mutant alleles) in *nectin1a* versus 68% (44/65) in *nectin1b*. 40% of tumors (26/65) had near-complete targeting of both genes. 14% of tumors (9/65) had low or no targeting (< 25% mutant alleles) in *nectin1a* versus only 8% (5/65) in *nectin1b*.

We are also showing in new Figure 2c the proportions of mutant alleles found in these tumors. Only mutant alleles accounting for at least 2% of all mutant alleles in a specific tumor were considered. In total, in 65 individual tumors, we detected 124 and 70

different mutant alleles in *nectin1a* and *nectin1b*, respectively. The vast majority of CRISPR-induced indels (insertions or deletions) resulted in frameshifts. Furthermore, we observed that targeting *nectin1b* resulted in significantly lower allele diversity in each tumor than targeting *nectin1a*, with an average of 3.2 and 4.5 mutant alleles per tumor, respectively (median of 2 and 4 mutant alleles, respectively).

Finally, new Figure 2d displays the nature and proportions of mutant alleles of *nectin1a* and *nectin1b* in 10 individual primary zebrafish melanomas as a representative example.

Together, these data demonstrate a high targeting efficiency of the CRISPR MiniCoopR system in primary zebrafish melanomas and further support a more prominent role of *nectin1b* compared to *nectin1a* in the phenotypes observed *in vivo*. All results from the transplantation experiments shown in Figure 2 were obtained from tumors with near-complete targeting of *nectin1b*.

(Figure 2b) Proportion (in %) of mutant alleles of *nectin1a* and *nectin1b* in 65 primary zebrafish melanomas resulting from the injection of a CRISPR MiniCoopR vector targeting *nectin1a* and *nectin1b* into one-cell stage *Tg(mitf:BRAF^{V600E});tp53^{-/-}* zebrafish embryos as shown in (a). Bar: median. Paired two-tailed t-test. (Figure 2c) Box-and-whisker plot representing the number of mutant alleles of *nectin1a* and *nectin1b* per primary zebrafish melanoma described in (b). Min-max, 1st and 3rd quartiles, median, n=65. Paired two-tailed t-test. (Figure 2d) Type and proportion (in %) of mutant alleles of *nectin1a* and *nectin1b* in 10 representative primary zebrafish melanomas, as described in (b). D: deletion, I: insertion, the number represents the number of base pairs affected by the mutation. The last section in each bar represents all other minor alleles (accounting for <2% of mutant alleles in that tumor).

Are there differences in the mutational frequency in nectin1a/b in melanoma cells that spread versus those that remain at the dorsal injection site?

Thank you for this excellent question. We were able to compare the mutational pattern in 3 primary zebrafish melanomas, their corresponding allografts (dorsal injection site)

and disseminated patches. Due to near-complete gene targeting in primary tumors, we could not observe further selection of *nectin1* knockout clones upon transplantation or in vivo dissemination. We did, however, observe some degree of clonal selection in disseminated patches originating from tumors with mosaic *nectin1a* targeting, suggesting that our spreading assay offers favorable conditions for clone competition. We now show this data as new Extended Data Figure 3b.

(Extended Data Figure 3b) Type and proportion of mutant alleles of *nectin1a* and *nectin1b* in 3 examples of primary zebrafish melanomas resulting from the injection of a CRISPR MiniCoopR vector targeting *nectin1a* and *nectin1b* into one-cell stage *Tg(mitf:BRAF^{V600E});tp53^{-/-}* zebrafish embryos, their corresponding allografts in the dorsal region of adult recipient *casper* zebrafish, and matched disseminated patches. D: deletion, I: insertion, the number represents the number of base pairs affected by the mutation. The last unlabeled slice of each pie chart represents all other minor alleles (accounting for <2% of mutant alleles in that sample).

Third, the conclusions in the zebrafish model would be strengthened by the addition of appropriate controls, such as other molecules involved in cell-cell adhesion (e.g., E-cadherin). In addition, the control-KO used is not well described in the main text, nor is the reasoning behind using this as a control. Such data would help to address the concern that knocking down any cell-cell adhesion molecule may cause melanoma cell spreading in the transplantation system.

Although knocking-out other adhesion molecules in zebrafish melanoma and measuring the impact of these manipulations on cell dissemination *in vivo* would be of great interest, these experiments would take 9-12 months to complete and would exceed the

scope of the current study focusing on the role of NECTIN1 in melanoma. In view of the published data on E-cadherin in cancer metastasis, it is conceivable that increased melanoma dissemination would also be observed with this particular gene. We have now described the controls used in the present study in greater details. Two CRISPR MiniCoopR vectors were used as controls, one containing a non-targeting gRNA and the other targeting *arhgap11a*, an unrelated gene that does not impact melanoma initiation or progression in our system.

4) Figure 2. Details of tumor cell behavior in primary tumors is lacking. Is there evidence of metastasis or localized cell spreading in the primary tumors, as the authors have previously demonstrated for other melanoma oncogenes in the zebrafish model?

This point is well taken. We have now examined 21 longitudinal sections of 16-week-old adult zebrafish bearing control or *nectin1*-knockout primary melanomas stained with hematoxylin/eosin (H&E). We have assessed local tumor invasion and dissemination to distant organs including gills, thymus, kidney, spleen, liver, and intestine. Although we did not detect differences in local tumor invasion into adjacent muscle, we found significantly higher organ involvement with *nectin1*-knockout tumors than with control tumors. We now show examples of tumor cell seeding of distant organs in new Figure 2h and quantification of organ involvement in new Figure 2j, with details of the organs involved in Figure 2i.

(Figure 2h) Pictures of different organs presenting with disseminated tumor cells (white arrows) in Hematoxylin/Eosin-stained longitudinal sections of 16-week-old zebrafish bearing *nectin1*-knockout melanoma. Scale bar: 20 μ m. (Figure 2i) Number of 16-week-old zebrafish presenting with disseminated melanoma cells in the indicated organs. (Figure 2j) Quantification of organ involvement in 11 16-week-old zebrafish bearing *control*-knockout melanoma and 10 16-week-old zebrafish bearing *nectin1*-knockout melanoma. Two-tailed t-test.

5) Figure 4. The analysis of p-fak by IHC should be quantified and/or inclusion of a immunoblot added to quantify the extent to which p-fak is increased in nectin1-knockout tumors.

The p-fak IHC was quantified in Extended Data Fig. 6e (now Extended Data Figure 6j) using a scoring system reflecting the extent and level of the IHC signal. We have now quantified the intensity of p-fak in individual fields of each of the 47 tissue-sections (19 control-knockout and 28 nectin1-knockout tumors) using ImageJ. This quantification is shown in new Figure 4k.

(**Figure 4j**) Representative images of sections of *control*-knockout (*ctrl*-KO) or *nectin1*-knockout (*nectin1*-KO) primary zebrafish melanomas transplanted into adult casper recipients, stained for phospho-fak by immunohistochemistry. Scale bar: 50 μ m. (**Figure 4k**) Quantification of phospho-fak staining intensity in 19 secondary *casper* recipients transplanted with *nectin1*-wildtype zebrafish melanomas and 28 secondary *casper* recipients transplanted with *nectin1*-knockout zebrafish melanomas. Mean \pm SD. Two-tailed *t*-test.

6) Figure 4. The authors are targeting 6 different alleles in panel 4h, which again is not fully described in the main text or methods. How efficient are the Cas9/gRNA's at targeting/mutating the expected alleles?

FAK1 (*PTK2*) also has 2 orthologs in zebrafish, *fak1a* (also known as *ptk2ab*) and *fak1b* (also known as *ptk2aa*), which we collectively referred to as *fak* for ease of presentation. As for *nectin1*, we decided to target both at the same time using our tissue-specific CRISPR vector. The gRNA sequences used in the CRISPR vector were specified in the Methods section. We have sequenced both *nectin1* and *fak1* alleles in 9 primary zebrafish melanomas by next-generation sequencing. *Nectin1a* and *nectin1b* targeting efficiency was consistent with our previous observations with an average of 69 and 70%,

respectively. We found similar levels of targeting for *fak1a* and *fak1b*, with an average proportion of mutant alleles per tumor of 77 and 63%, respectively. These results, now shown in Extended Data Figure 6l, indicate that our CRISPR system achieves high levels of gene targeting and allows the generation of compound knockout tumors in zebrafish. Note that the proportion of mutant alleles was generally similar for all 4 genes in each tumor, suggesting that the samples exhibiting relatively lower gene targeting might contain a higher proportion of non-tumor cells.

(Extended Data Figure 6l) Proportion (in %) of mutant alleles of *nectin1a*, *nectin1b*, *fak1a*, and *fak1b* in 9 primary zebrafish melanomas resulting from the injection of 2 melanocyte-specific CRISPR vectors targeting *nectin1a* and *nectin1b* or *fak1a* and *fak1b* into one-cell stage *Tg(mitf:BRAF^{V600E});tp53^{-/-}* zebrafish embryos. Each section of each bar represents a different mutant allele (accounting for >2% of mutant alleles in that tumor).

7) Figure 4 and 5. There are no details describing the impact of *fak* knockout or ectopic IGF1R expression in the primary tumors in vivo (see comments above). Do these tumors have differences in localized spreading compared to *nectin1a/1b* alone and/or evidence of dissemination to other organ sites? Inclusion of primary tumor cell behaviors would bolster the in vivo conclusions that currently rely on using a transplantation approach in irradiation/immune suppressed recipients which may also impact the behavior of these cells.

We agree that this information could add to the paper. We now provide data about the impact of *fak* knockout and constitutive IGF1R expression on zebrafish melanoma onset, proliferation, and dissemination. In new Extended Data Figure 6m, the melanoma-free survival curves of zebrafish with melanocyte-specific inactivation of *nectin1* or both *nectin1* and *fak* are undistinguishable, whereas new Extended Data Figure 7h shows that the expression of constitutive IGF1R (IGF1R*) significantly accelerates the onset of

nectin1-knockout tumors *in vivo* compared to a GFP control, in line with the generally pro-tumorigenic role of aberrant IGF1 signaling. We also measured cancer cell proliferation by staining zebrafish melanoma sections with an antibody directed against phospho-H3 but the data did not show significant differences between any of the genetic conditions tested (new Extended Data Figures 6n and 7i). Finally, we assessed melanoma dissemination to distant organs on zebrafish tissue-sections stained with hematoxylin/eosin (H&E). Interestingly, we found a significantly reduced organ involvement in fish bearing *nectin1*-knockout primary melanomas upon *fak* knockout (new Figure 4m). Details on the organs involved are displayed in new Figure 4l. In contrast, we did not observe differences in organ involvement between *nectin1*-knockout tumors expressing GFP or IGF1R* (new Extended Data Figure 7j). This is likely due to the fact that IGF1R*-expressing tumors form earlier than GFP-expressing ones and have therefore more time to invade the whole body of the fish. These data are now presented in the main text and Methods section of our revised manuscript.

(Extended Data Figure 6m) Tumor-free survival curves of *Tg(mitf:BRAF^{V600E});tp53^{-/-}* zebrafish injected with vectors targeting either *nectin1* and a control gene, or *nectin1* and *fak*. Pooled data of 2 independent experiments. Log-rank test. **(Extended Data Figure 6n)** Quantification of overall phospho-H3 staining intensity in 7 *nectin1/ctrl*-knockout and 9 *nectin1/fak*-knockout primary zebrafish melanomas. Mean \pm SD. Two-tailed *t*-test. **(Figure 4l)** Number of 16-week-old zebrafish presenting with disseminated melanoma cells in the indicated organs. **(Figure 4m)** Quantification of organ involvement in 9 16-week-old zebrafish bearing *nectin1/ctrl*-knockout melanoma and 9 16-week-old zebrafish bearing *nectin1/fak*-knockout melanoma. Two-tailed *t*-test.

(Extended Data Figure 7h) Tumor-free survival curves of *Tg(mif:BRAF^{V600E});tp53^{-/-}* zebrafish injected with a vector targeting *nectin1* and vectors expressing either GFP or CD8-IGF1R (IGF1R*). Pooled data of 2 independent experiments. Log-rank test. (Extended Data Figure 7i) Quantification of overall phospho-H3 staining intensity in 8 *nectin1*-knockout/GFP and 8 *nectin1*-knockout/IGF1R* primary zebrafish melanomas. Mean \pm SD. Two-tailed *t*-test. (Extended Data Figure 7j) Number of 16-week-old zebrafish presenting with disseminated melanoma cells in the indicated organs.

Reviewer #2:

Remarks to the Author:

In this manuscript, Ablain et al describe a frequent deletion of adhesion gene NECTIN1 found in melanomas. Then they go on to investigate if this deletion triggers melanoma dissemination upon local IGF1 depletion.

The part of the data related to melanoma genetics is very interesting since this gene has not been linked to melanoma progression before. The functional validation needs further work and therefore dampens overall enthusiasm.

General comment

This is a manuscript about melanoma dissemination. Part of this dissemination occurs via invasion into the dermis (complex matrix with a high content of collagen I). It is therefore surprising that authors perform all of their functional validation in transwell assays. While this is a valid model for understanding some aspects of dissemination (chemotaxis, confinement, nuclear deformation, etc), authors claim this mechanism works via cell-matrix adhesion signaling- via integrin signaling- but they never perform any validation in a matrix based migration-invasion type assay (they only do one adhesion assay to laminin). On the other hand, 2D

migration and 3D migration mechanisms are quite different, therefore authors could have missed important signalling in the nectin-deficient type melanomas. 3D inverted invasion assays, spheroids grown and invading into collagen or even transwells with some type of matrix overlaid in them, could have been an alternative more convincing approach.

For example, Figure 4c, there is no matrix in this transwell so why inhibition of integrins has an effect? It may be completely independent of cell-matrix interactions. In fact, many 3D- invasive programs rely on low levels of adhesion to the matrix- to achieve higher speed, so these connections between matrix-nectin and invasion have not been fully developed.

We agree with the reviewer's comment regarding the role of the extra cellular matrix (ECM) in cancer dissemination and we now provide additional data on: 1) the role of specific ECM components in NECTIN1-dependent phenotypes, 2) the specific integrins involved in these phenotypes.

To further characterize the behavior of *NECTIN1*-deficient cells in a 3D environment, we seeded a small number of A375 human melanoma cells expressing either a control shRNA or an shRNA directed against *NECTIN1* in collagen-rich matrix in the absence of serum. We observed that after 7 days, *NECTIN1*-deficient cells were significantly more spread out than *NECTIN1*-control cells. We quantified this observation by measuring the number of spatially separated groups of cells present in each colony using the "particle analysis" function of ImageJ. These results, displayed in new Figure 3c-d, support the idea that *NECTIN1* increases the spreading capacity of melanoma cells in a 3D environment.

(Figure 3c) Top: Images of the colonies formed by 1,000 A375 human melanoma cells stably expressing a control shRNA (shCTRL) or an shRNA directed against *NECTIN1* (shNECTIN1) after 7 days in collagen-rich matrix in the absence of serum (FBS). Scale bar: 50 μ m. Representative of 3 independent experiments. Bottom: Analysis of the images shown on top using ImageJ. **(Figure 3d)** Spread of the colonies formed by A375 human melanoma cells in collagen-rich matrix as shown in **c**, bottom. Spread was measured with ImageJ as the number of individual particles detected in each colony. Mean \pm SD. Two-tailed *t*-test.

Moreover, by plating human melanoma cells in Matrigel, we found that the difference in invasion capacity between *NECTIN1*-proficient and deficient cells was similar (approximately 4-fold) to the difference in migration capacity that we previously observed using a simple transwell assay (new Figure 3f). These results demonstrate that *NECTIN1* loss increases both 2D migration and 3D invasion of melanoma cells.

(Figure 3f) Invasion through matrigel of A375 human melanoma cells stably expressing an shRNA directed against *NECTIN1* (shNECTIN1) relative to cells expressing a control shRNA (shCTRL) in a transwell assay after 12 hours of serum starvation. Mean \pm SD of 4 independent experiments. Paired two-tailed *t*-test.

We also added different types of matrices to transwells and observed similar differences between *NECTIN1*-proficient and *NECTIN1*-deficient cells as in simple transwell assays (new Extended Data Figure 5d, please see specific point 1 below). This similarity of behavior in 2D and 3D assays may be explained by the fact that melanoma cells express ECM proteins. We show in new Extended Data Figure 5e that A375 human melanoma cells express high levels of fibronectin and laminin. In the modified transwell assay, since the cells are plated 12 hours before establishing the chemotactic gradient, it is likely that they have already secreted some ECM. Indeed, we confirmed by immunofluorescence the production of both fibronectin and laminin by A375 cells in that setting (new Extended Data Figure 5f, please see specific point 2 below). Interestingly, we also

detected strong laminin and fibronectin networks in the dorsal cavity of recipient zebrafish in which primary zebrafish melanoma cells were transplanted (new Extended Data Figures 5g-h, please see specific point 11 below).

We have now expanded our analysis of the adhesion of *NECTIN1*-deficient cells to different types of matrices. In particular, we show that *NECTIN1*-deficient cells adhere to laminin, fibronectin, and, to a lesser extent, vitronectin, but not collagen I, significantly more than *NECTIN1*-proficient cells (new Figure 4b, please see specific point 1 below).

Finally, we systematically down-regulated each beta integrin by siRNA in *NECTIN1*-deficient cells to formally identify the integrins responsible for increased cell migration. We observed that knocking down integrins beta3, beta4 and beta5 significantly reduced the migration of *NECTIN1*-deficient cells (new Figures 4c-d, please see specific point 2 below). In addition to the abolition of *NECTIN1*-deficient cell migration upon treatment with integrin avb3/avb5 inhibitors (Figure 4f), we now show that an antibody known to block integrin a6b4 also significantly reduces *NECTIN1*-induced migration (new Figure 4e).

Together with our transcriptomic analyses implicating integrin avb3 signaling (Extended Data Figure 5b), our proteomic analyses showing changes in integrins a6b4 and a5b1 at the cell surface (Figure 4a), and our ECM adhesion experiments suggesting a role for laminin, fibronectin, and, to a lesser extent, vitronectin (new Figure 4b), our new data point to integrins avb3, avb5 and a6b4 as the main mediators of *NECTIN1*-deficient cell migration and invasion. We thank the reviewer for their suggestions as we believe that our new data strengthens the mechanistic part of our study.

Specific points

1) The list of integrins upregulated after nectin loss are : beta4 , alpha 5, beta 1 and alpha 6, why do they not use matrices for these receptors in their migration/invasion assays? (collagen I, fibronectin, laminin)

We have now performed both migration and adhesion assays with a wider range of matrices. Similar differences in migration were measured between *NECTIN1*-deficient and proficient cells when coating transwells with collagen-I, fibronectin, laminin or vitronectin (new Extended Data Figure 5d). This result is not surprising since *NECTIN1*-

deficient cells already migrate more than their wildtype counterparts even in the absence of added matrix.

(Extended Data Figure 5d) Migration of A375 human melanoma cells stably expressing an shRNA directed against *NECTIN1* (shNECTIN1) relative to cells expressing a control shRNA (shCTRL) in transwells coated with the indicated ECM proteins and after 12 hours of serum starvation. Mean \pm SD of 4 independent experiments. Paired two-tailed *t*-test.

However, adhesion assays did reveal differences between matrices in their ability to promote the adhesion of *NECTIN1*-deficient cells. We found that *NECTIN1*-deficient melanoma cells adhered significantly more to laminin, fibronectin, and to a lesser extent, vitronectin, but not collagen-I, than *NECTIN1*-proficient cells. These data, displayed in new Figure 4b, expand the characterization of the adhesion changes occurring in melanoma cells upon *NECTIN1* loss and inform the repertoire of integrins potentially implicated in the migration of *NECTIN1*-deficient cells (please see response to point 2).

(Figure 4b) Adhesion of A375 human melanoma cells stably expressing an shRNA directed against *NECTIN1* (shNECTIN1) relative to cells expressing a control shRNA (shCTRL) to collagen-I-, fibronectin-, laminin-, or vitronectin-coated surfaces at various timepoints after seeding. Mean \pm SD of 4 independent experiments. Paired two-tailed *t*-test.

2) Moreover why do they use an integrin inhibitor that targets other integrins - alpha V beta3 and alpha V beta 5 (SB273005) which would affect vitronectin adhesion. It suggests lack of a specific cell-matrix type mechanism. Maybe is more a mechanism related to the secretion of chemokines and that is why they see effects in a transwell with no matrix.

We thank the reviewer for this question that allowed us to clarify the part of our manuscript related to integrins. We concluded from our proteomic data (Figure 4a) that integrins might be involved in the migration difference observed between *NECTIN1*-deficient and proficient cells but we did not focus specifically on the two integrins, $\alpha 5\beta 1$ and $\alpha 6\beta 4$, that came out of these analyses since integrin signaling as a whole is likely altered in melanoma. We used $\alpha v\beta 3/\alpha v\beta 5$ inhibitors because they were some of the rare commercially available integrin inhibitors and because $\alpha v\beta 3$ signaling had also come out of our transcriptomic analysis. We have now performed additional experiments to determine the types of integrins responsible for the increase in melanoma cell migration caused by *NECTIN1* loss. We downregulated each beta integrin with siRNAs in *NECTIN1*-deficient cells and measured the impact of this downregulation on their migration (new Figure 4c). We observed that genetic inactivation of integrin $\beta 5$, and to a lesser extent, integrins $\beta 3$ and $\beta 4$, inhibited the migration of *NECTIN1*-deficient cells in a transwell assay (new Figure 4d). These results are consistent with the findings that *NECTIN1*-deficient cells show increased adhesion to laminin, vitronectin, and fibronectin that has also been identified as a potential ligand for integrin $\alpha v\beta 3$ (Xiong et al, Science 2002).

(Figure 4c) Western blot analysis of ITGB levels in A375 cells stably expressing a control shRNA (shCTRL) or an shRNA directed against *NECTIN1* (shNECTIN1) and transfected with a control siRNA (C) or siRNAs targeting *ITGB1* (1), *ITGB2* (2), *ITGB3* (3), *ITGB4* (4) or *ITGB5* (5). Representative of 4 independent experiments. **(Figure 4d)** Migration of the cells described in **c** relative to cells stably expressing a control shRNA (shCTRL) and transfected with a control siRNA in a transwell assay after 12 hours of serum

starvation. Mean \pm SD of 3 independent experiments. Paired two-tailed *t*-test. *: $p < 0.05$. For ITGB3, $p = 0.053$.

Additionally, since integrin $\alpha 6 \beta 4$ was one of the most upregulated proteins on the surface of melanoma cells upon *NECTIN1* loss, and since downregulation of integrin $\beta 4$ reduced the migration of *NECTIN1*-deficient cells, we evaluated the effect of GoH3, an antibody known to block integrin $\alpha 6 \beta 4$, on melanoma cell migration. We found that pharmacological inhibition of integrin $\alpha 6 \beta 4$ reduced the migratory capacity of *NECTIN1*-deficient cells by 33% (new Figure 4e).

(Figure 4e) Migration of A375 human melanoma cells stably expressing an shRNA directed against *NECTIN1* (shNECTIN1) relative to cells expressing a control shRNA (shCTRL) in a transwell assay upon treatment with an integrin $\alpha 6 \beta 4$ blocking antibody (GoH3, 40 $\mu\text{g}/\text{mL}$). Mean \pm SD of 4 independent experiments. Paired two-tailed *t*-test.

Our functional data thus indicates that changes in cell adhesion are at least in part responsible for increasing cell migration and invasion upon *NECTIN1* loss and IGF1 signaling inhibition. Moreover, taken together, our results support a role of integrins $\alpha 6 \beta 4$ and $\alpha v \beta 3 / \alpha v \beta 5$ and their cognate ligands laminin and vitronectin/fibronectin, respectively, in the migration of *NECTIN1*-deficient cells.

We agree that the fact that these effects can be seen in transwell assays without the addition of specific matrices is puzzling. However, this could be explained by the synthesis of several ECM proteins by melanoma cells themselves. Indeed, A375 human melanoma cells express high levels of fibronectin (FN1) and several laminin genes, as now shown in new Extended Data Figure 5e. We also observed strong staining of these cells cultured in transwell conditions with antibodies directed against human fibronectin or human laminin (new Extended Data Figure 5f). Our data with integrin inhibitors and siRNAs against integrins thus support the idea that *NECTIN1*-deficient cells at least in

part rely on interactions with the ECM, which they may have deposited themselves, to migrate and disseminate.

(Extended Data Figure 5e) Expression levels of various ECM genes in 3 clones of A375 human melanoma cells, as measured by RNA-sequencing. (Extended Data Figure 5f) Immunofluorescence analysis of fibronectin or laminin (green) in A375 cells cultured in the absence of serum (FBS) for 12 hours. Scale bar: 10 μ m.

3) In general, in all the figures, there is a lack of information on migration assays and western blots: how many times were the experiments done. There are no quantification of the western blots- and no statistical analysis of these western blots either.

The number of biological replicates was indicated in the bottom left corner of each figure panel presenting results of *in vitro* experiments (e.g. migration, adhesion or proliferation assays). We are now mentioning it in figure legends as well. This information, as well as a quantification of signal intensities, has also been added for Western blots. We are showing here the Western Blots presented in Figure 4i and Extended Data Figure 6h. We also quantified the Western Blots associated with siRNA experiments but these results are not shown in the paper in the interest of space.

(Figure 4i) Western Blot analysis of FAK and SRC phosphorylation levels in A375 human melanoma cells stably expressing a control shRNA (C) or an shRNA directed against *NECTIN1* (sh) in the absence of serum. Representative example of 5 independent experiments. Signal intensity was normalized to GAPDH and presented as a ratio to the vehicle-treated shCTRL condition. Paired two-tailed *t*-test. *: < 0.05; **: < 0.01. **(Extended Data Figure 6h)** Western blot analysis of FAK and SRC phosphorylation levels in A375 human melanoma cells stably expressing a control shRNA (C) or an shRNA directed against *NECTIN1* (sh), serum-starved for 12 hours and treated with an integrin avb3/avb5 inhibitor (ITGi, echistatin), a FAK inhibitor (FAKi, PF562271) or a SRC inhibitor (SRCi, dasatinib) for 6 hours. veh: vehicle-treated. Representative example of 4 independent experiments. Signal intensity was normalized to GAPDH and presented as a ratio to the vehicle-treated shCTRL condition. Paired two-tailed *t*-test. *: < 0.05; **: < 0.01; ***: < 0.001.

4) Migration assays (according to materials and methods): these assays were performed with different endpoints (6, 20, 24 h depending on cell line). If you represent all this data together, the assays need to have the same endpoint. On the same lines, all assays seem to be normalised to 1 when comparing different cell lines. Moreover, what are the migratory abilities of the control cells (their basal migration) without normalising to 1?

In these experiments, we investigate the effects of *NECTIN1* inactivation on the migration of human melanoma cells. The most relevant measure of these effects is the fold change in migration between *NECTIN1*-deficient and *NECTIN1*-proficient cells. That is why we normalized to 1 the migration of cells expressing a control shRNA in our figures. Moreover, the evaluation of this fold change is only faithful if the basal level of migration of *NECTIN1*-wildtype cells can itself be accurately measured. That is why we had to let different cell lines migrate for different times depending on their intrinsic migratory capacity. Our data actually suggest that *NECTIN1* loss increases the migration of melanoma cells independent of their initial migratory capacity. To avoid any confusion regarding the representation of data from different cell lines on the same figure panel, we are now also mentioning in the legends of Figures 3g, 4h and 5d the fact that

different endpoints were used for different cell lines (which initially appeared in the Methods section).

We understand, however, that the representation used in Figures 3g, 4h and 5d, masks differences between melanoma cell lines in their intrinsic migratory capacity. In order to also provide this information, we have measured the migration of all 6 human melanoma cell lines at a single timepoint (10 hours) and represented the data without normalization in new Extended Data Figure 4i.

(Extended Data Figure 4i) Migration of 6 human melanoma cell lines stably expressing a control shRNA (shCTRL) or an shRNA directed against NECTIN1 (shNECTIN1) in a transwell assay after 12 hours of serum starvation. Cells were allowed to migrate for 10 hours. Data represented as the number of cells detected in a 2.35 x 2.35 mm field of transwell (a.u.). Mean \pm SD of 4 independent experiments. Paired two-tailed t-test. *: < 0.05; **: < 0.01; ***: < 0.001.

5) Does it relate to their endogenous nectin levels? For that again, same endpoint assays need to be chosen.

Our data suggest that reducing the levels of NECTIN1 in established melanoma cells further increases their migratory capacity specifically in the absence of IGF1 signaling, a parameter that may not have influenced the history of these cell lines. The basal migratory capacity of human melanoma cell lines depends on many factors including their differentiation state and is likely to have been determined by the strong selective pressures at work during the establishment of the cell lines. The absence of a clear correlation between a complex determinant of metastasis, such as NECTIN1, and the basal migratory capacity of the 6 human melanoma cell lines that we have used *in vitro*, was therefore not surprising. We analyzed NECTIN1 levels in our panel by Western

Blotting. Two of the cell lines with the strongest basal migration (A375 and CJM) expressed relatively high levels of NECTIN1, as did the cell line with the weakest basal migration (SKMEL2), whereas cell lines exhibiting intermediate migratory capacities had variable NECTIN1 levels (RPMI7951, G361). We are showing this new data in Extended Data Figure 4h.

(Extended Data Figure 4h) Western Blot analysis of NECTIN1 levels in 6 human melanoma cell lines. Representative example of 3 independent experiments.

6) Can authors rescue their effects on migration with an RNAi resistant version of nectin?

We have performed rescue experiments with inducible expression of a *NECTIN1* construct mutated in the shRNA recognition sequence and have observed a partial rescue of the migration phenotype induced by *NECTIN1* loss (new Extended Data Figure 4f). Interestingly, this rescue was the strongest when shRNA-resistant *NECTIN1* was expressed before the shRNA (data not shown), supporting the idea that changes induced by *NECTIN1*-loss may be implemented by a cell state program that gradually becomes irreversible in culture.

(Extended Data Figure 4f) Migration of A375 human melanoma cells stably expressing either *GFP* or an shRNA-resistant *NECTIN1* gene (*NECTIN1**), and either a control shRNA (shCTRL) or an shRNA directed against *NECTIN1* (shNECTIN1) in a transwell assay after 12 hours of serum starvation. Mean \pm SD of 3 independent experiments. Paired two-tailed t-test.

7) When looking for the possible FAK related mechanism they base their directions in two data sets: a protein phospho-array from which they chose FAK, while there is only a very modest fold increase in phosphorylation (1.2 if I understood this data correctly). This array does not seem to have produced optimal fold changes. Can the authors show original data, the dot blot? You would ideally want a phospho-change of at least 2-fold.

We are now showing the dot blots in Extended Data Figure 6a. While we agree that the changes are small, large changes are typically observed with drug treatment whereas here, we are comparing two genetic conditions and are thus expecting rather subtle changes. Moreover, the phospho-kinase assay was used as a discovery tool to generate hypotheses that could then be verified by Western Blotting analyses. To consolidate our results, we pooled the data from 4 individual phospho-kinase assays and performed a statistical analysis comparing differences in protein phosphorylation between *NECTIN1*-deficient and *NECTIN1*-proficient cells. The results, shown in new Extended Data Figure 6b, confirmed the increase in phospho-FAK (1.5 fold), phospho-MSK1/2, and b-catenin, while also revealing increases in phospho-Hck, phospho-STAT5, and phospho-AMPK2a.

(Extended Data Figure 6a) Representative images of a phospho-kinase array immunoblot of A375 human melanoma cells stably expressing a control shRNA (shCTRL) or an shRNA directed against *NECTIN1* (shNECTIN1) upon serum depletion. The red boxes indicate the dots corresponding to phospho-FAK. (Extended Data Figure 6b) Most differentially activated proteins in A375 human melanoma cells stably expressing an shRNA directed against *NECTIN1* (shNECTIN1) relative to cells expressing a control shRNA (shCTRL), upon serum depletion, as measured by signal intensity on a phospho-kinase array immunoblot. Mean \pm SD of 4 independent experiments. Paired two-tailed t-test. *: p < 0.05.

8) When screening different inhibitors (rac, wnt, jnk inhibitors) in the migration assay (Ext Data 5e), authors argue they use a concentration that does not affect viability of cells. Many of the migration-related pathways also drive the survival of cancer cells, therefore how can they make sure the drugs are hitting the target and they are not just using a very low concentration that only mildly reduces pathway activity? Authors should confirm by Western blot that downstream signalling is affected after drug treatment at the concentrations they have chosen of all those drugs.

This experiment was a mini-screen with inhibitors of common signaling pathways involved in cell migration. Its goal was to hint at pathways potentially implicated in the migration of *NECTIN1*-deficient cells in order to generate new hypotheses, not to test a specific hypothesis regarding a pathway in particular. This is the reason why we did not perform extensive characterization of these inhibitors. We took the approach of using drugs at the highest non-toxic concentration, as defined as the concentration reducing the viability of A375 human melanoma cells by less than 10% after 4 days in culture. When not affecting cell viability, small molecule inhibitors were used at 1 μ M, a relatively high concentration often exceeding their reported IC50, thus ensuring target inhibition. Below are the viability curves for all the drugs used in the screen. The viability of melanoma cells mainly depends on the activity of the MAPK pathway so we do not expect inhibitors solely targeting migration pathway to significantly affect cell viability in the absence of MAPK-targeted therapy, unless they are used at a concentration inducing non-specific toxicity.

As an example, we have now validated the efficacy of 3 of these drugs on the phosphorylation status of known downstream targets: increase of S144 phosphorylation of PAK1 downstream of RAC1 inhibition, decrease of JNK phosphorylation downstream of ROCK1 inhibition, and decrease of c-Jun phosphorylation downstream of JNK inhibition (see figure for reviewers only below).

(Left) Cell viability curves of A375 human melanoma cells after 4 days of treatment with various small molecule inhibitors, relative to untreated cells. **(Right)** Western Blot analysis of PAK1, JNK and c-Jun phosphorylation levels in A375 human melanoma cells treated for 6 hours with RAC1, ROCK or JNK inhibitors. Representative example of 3 independent experiments.

9) Therefore, authors may not have chosen the best targets (or at least not the only possible targets). In their OMICs there are many more changes taking place that could be relevant for the migration of the nectin deficient melanomas. In fact, that FAK is required for 2D migration, is not a surprise but what happens in 3D migration and how matrix is connected to nectin downstream signalling is the key question that has not been addressed.

We agree that the drugs that we have tested do not inhibit all the targets potentially involved in the migration of *NECTIN1*-deficient cells but they block some of the most common migration pathways. In addition, we present several lines of evidence supporting the idea that the FAK/SRC pathway contributes to melanoma dissemination downstream of *NECTIN1* loss, including the fact that its genetic inactivation blocks the dissemination of *NECTIN1*-deficient melanoma cells *in vitro* and *in vivo*. Regarding the comment about 2D versus 3D migration, we have now repeated the experiments assessing the effects of FAK and SRC inhibitors on the migration of *NECTIN1*-deficient cells, this time in Matrigel. We obtained similar results as in simple transwell assays, indicating that the FAK/SRC axis is also essential for the invasion of *NECTIN1*-deficient cells in 3D (new Extended Data Figure 6g).

(Extended Data Figure 6g) Invasion through matrigel of A375 human melanoma cells stably expressing an shRNA directed against *NECTIN1* (shNECTIN1) relative to cells expressing a control shRNA (shCTRL) in a transwell assay upon treatment with a FAK inhibitor (FAKi, PF562271) or a SRC inhibitor (SRCi, dasatinib). Mean \pm SD of 4 independent experiments. Paired two-tailed *t*-test.

10) In those lines, in vivo staining of p-FAK, are they more prominent in the invasive areas within the tumours?

Thank you for this interesting question. We have now looked at tissue-sections of adult zebrafish transplanted with *nectin1*-knockout zebrafish melanoma cells stained by IHC with an antibody against phospho-FAK. Out of 28 individual fish, 10 (36%) displayed a staining pattern in the form of a gradient with stronger signal towards the invasive edge than directly under the skin, as shown in new Extended Data Figure 6k.

(Extended Data Figure 6k) Top: section of adult *casper* recipient transplanted with *nectin1*-knockout zebrafish melanoma cells, stained for phospho-fak by immunohistochemistry. Orientation from skin to

muscle is indicated. Scale bar: 50 μm . Bottom: histogram representing phospho-fak signal intensity along the black dotted line on the image shown on top. The red dotted line materializes the spatial increase in fak activity.

11) what is the matrix found there?

The type of matrix present in the dorsal cavity under the skin of adult zebrafish is unknown. Considering our results (please see response to points 1 and 2) relative to the types of matrices that *NECTIN1*-deficient cells adhere to, we assessed the presence of laminin and fibronectin in the transplantation area using antibodies previously shown to recognize the fish proteins. In all the recipient fish we looked at ($n=8$), we found strong laminin and fibronectin staining around melanoma cells, suggesting that these matrices could also play a role in this *in vivo* spreading assay (Extended Data Figures 5g-h).

(Extended Data Figure 5g) Immunofluorescence analysis of fibronectin in adult *casper* zebrafish transplanted with primary zebrafish melanoma cells. Scale bar: 20 μm . M: muscle. (Extended Data Figure 5h) Immunofluorescence analysis of laminin in adult *casper* zebrafish transplanted with primary zebrafish melanoma cells. Scale bar: 20 μm . M: muscle.

12) Can they correlate in vivo invasion to FAK activity levels?

We performed this analysis in the context of *nectin1*-wildtype or *nectin1*-knockout primary zebrafish melanoma cells transplanted subcutaneously into adult secondary recipients, knowing that *nectin1*-knockout cells disseminate significantly more than *nectin1*-wildtype cells in this setting. A p-fak IHC staining of zebrafish tissue-sections was quantified in Extended Data Fig. 6e (now Extended Data Figure 6j) using a scoring system reflecting the extent and level of the IHC signal. We have now quantified the intensity of p-fak in individual fields of each of the 47 tissue-sections (19 control-knockout and 28 *nectin1*-knockout tumors) using ImageJ. This quantification is shown in new Figure 4k and demonstrates significantly higher levels of FAK activity in the more migratory *nectin1*-knockout cells.

(Figure 4j) Representative images of sections of *control*-knockout (*ctrl*-KO) or *nectin1*-knockout (*nectin1*-KO) primary zebrafish melanomas transplanted into adult casper recipients, stained for phospho-fak by immunohistochemistry. Scale bar: 50 μ m. **(Figure 4k)** Quantification of phospho-fak staining intensity in 19 secondary *casper* recipients transplanted with *nectin1*-wildtype zebrafish melanomas and 28 secondary *casper* recipients transplanted with *nectin1*-knockout zebrafish melanomas. Mean \pm SD. Two-tailed *t*-test.

13) In the panel of 6 melanoma lines is there an anti-correlation between nectin levels and active FAK?

Our functional data indicate that *NECTIN1* inactivation increases FAK phosphorylation in the absence of serum. Surprisingly, a Western Blot analysis of *NECTIN1* and phospho-FAK levels revealed a positive correlation between FAK activity and *NECTIN1* levels in our panel of 6 human melanoma cell lines (new Extended Data Figures 6o, p, Pearson's correlation $R^2 = 0.73$, *p*-value = 0.03), in apparent contrast with our observations following the modulation of *NECTIN1* levels in melanoma. However, we also noticed that in 3 out of 4 cell lines exhibiting relatively high levels of *NECTIN1* (RPMI7951, CJM, A375 and SKMEL2), pharmacological inhibition of FAK for 12h induced a marked decrease in *NECTIN1* levels, as shown by Western Blotting (new Extended Data Figure 6q). This result points to a regulation of *NECTIN1* expression by FAK activity and suggests the existence of a feedback mechanism downstream of *NECTIN1* modulation. Note that this regulation loop is likely hindered in cases of *NECTIN1* genetic inactivation, which may explain this apparent discrepancy. We now discuss this interesting point in our revised manuscript.

(**Extended Data Figure 6o**) Western Blot analysis of NECTIN1 and phospho-FAK levels in 6 human melanoma cell lines. Representative example of 3 independent experiments. (**Extended Data Figure 6p**) Correlation plot of NECTIN1 and phospho-FAK levels in 6 human melanoma cell lines. Average values of 3 independent experiments. Pearson's correlation. (**Extended Data Figure 6q**) Western Blot analysis of NECTIN1 and phospho-FAK levels in 4 human melanoma cell lines treated with vehicle (Veh.) or a FAK inhibitor (FAKi, PF562271) for 12 hours. Representative example of 3 independent experiments.

14) The data on nectin -/- tumours being softer is interesting but authors have not linked this feature to anything, if anything stiffer tumours have been linked to worse prognosis. It is not clear that this piece of data adds to their argument, unless authors look at the cytoskeleton of the cancer cells properly.

We presented this piece of data as an additional characterization of the biophysical properties of *nectin1*-knockout primary zebrafish tumors. While elevated stiffness of the ECM has been linked to increased tumor cell aggressiveness, it was interesting to note that genetic modulation of a cell-cell adhesion molecule like NECTIN1 could also modify tumor elasticity. In the absence of formal evidence of a direct link between this observation and the consequences of NECTIN1 loss on tumor dissemination, we have now moved this piece of data to Extended Data Figure 3.

15) With regards to that, all the imaging performed in vitro needs improvement. Authors say there is a cell shape change but it is really difficult to see in the presented images (specially F-actin). Higher resolution and higher quality imaging would help. Moreover, quantitative image analysis of both Actin structures and of AJ needs to be performed, not just show one picture.

We apologize for the low quality of the images that was likely due to a loss of resolution during file transfer at initial submission. We have now added a higher magnification picture of adherens junctions as new Extended Data Figure 4j.

(Extended Data Figure 4j) Immunofluorescence analysis of α -catenin (green) and F-actin (red) in A375 human melanoma cells stably expressing a control shRNA in the absence of serum for 12 hours. Scale bar: 10 μ m. Representative images of 5 independent experiments.

We have also quantified AJ formation based on alpha-catenin signal intensity (new Figure 3i), cell area based on F-actin staining (new Figure 3j) and number of stress fibers per cell based on alpha-actinin staining (new Extended Data Figure 4l) using ImageJ on our immunofluorescence images. As expected, AJ junctions are formed specifically in *NECTIN1*-proficient cells upon serum withdrawal. Stress fibers are essentially absent in normal serum conditions. While they form in both *NECTIN1* conditions in response to serum withdrawal, *NECTIN1*-deficient cells displayed significantly more of them. Additionally, we measured a larger cell size in the *NECTIN1*-deficient condition, suggestive of a cytoskeletal reorganization induced by *NECTIN1* loss.

(Figure 3h) Immunofluorescence analysis of α -catenin (green) and F-actin (red) in A375 human melanoma cells stably expressing a control shRNA (shCTRL) or an shRNA directed against *NECTIN1* (shNECTIN1) and cultured in the presence or absence of serum (FBS) for 12 hours. Scale bar: 10 μ m. Representative

images of 5 independent experiments. **(Figure 3i)** Quantification of adherens junction (AJ) formation as measured by α -catenin signal intensity on immunofluorescence images as shown in **(h)**. Mean \pm SD of $n=4$ fields per condition. Paired two-tailed t -test. *: $p < 0.05$, **: $p < 0.01$, ***: $p < 0.001$, ns: not significant. **(Figure 3j)** Quantification of cell surface area based on immunofluorescence images as shown in **(h)**. Mean \pm SD of $n=21, 38, 18, 27$ cells for shCTRL+FBS, shCTRL-FBS, shNECTIN1+FBS, and shNECTIN1-FBS, respectively. Paired two-tailed t -test. *: $p < 0.05$, **: $p < 0.01$, ***: $p < 0.001$, ns: not significant.

(Extended Data Figure 4k) Immunofluorescence analysis of N-cadherin (green) and α -actinin (red) in A375 human melanoma cells stably expressing a control shRNA (shCTRL) or an shRNA directed against *NECTIN1* (shNECTIN1) and cultured in the presence or absence of serum (FBS) for 12 hours. Scale bar: 10 μ m. Representative images of 5 independent experiments. **(Extended Data Figure 4l)** Quantification of the number of stress fibers per cell based on α -actinin staining on immunofluorescence images as shown in **(i)**. Mean \pm SD of $n=29, 26, 24, 20$ cells for shCTRL+FBS, shCTRL-FBS, shNECTIN1+FBS, and shNECTIN1-FBS, respectively. Paired two-tailed t -test. **: $p < 0.01$, ***: $p < 0.001$.

16) Regarding the IGF connections. If IGF1 induces junctions, what is the effect of adding IGF (or an inhibitor) in the migration assay? Does it reduce it? Other labs have seen a pro-migratory effect of this factor in melanoma. Also, can authors show the level of over expression of IGFR*, I could not find it.

We showed that the genetic activation of IGF1 signaling in *NECTIN1*-deficient melanoma cells via the expression of a constitutive form of IGF1R reduced their propensity to migrate. We have now assessed the effect of IGF1 treatment itself and found that it partially inhibits the migration of *NECTIN1*-deficient melanoma cells in the modified transwell assay (new Figure 5e). Interestingly, this effect seems specific as IGF1 treatment slightly increases (although not statistically significantly) the migration of control cells.

Indeed, IGF1 signaling has been reported to decrease cell-cell adhesion and promote EMT in several epithelial tumors (for example by Lopez et al, Cancer Cell 2002 or Castano

et al, Cancer Discovery 2013), and in melanoma, inhibition of IGF1 has been shown to decrease proliferation and invasion by inhibiting an EMT-like program (Le Coz et al, Oncotarget 2016). Our data are consistent with the idea that low IGF1 increases epithelial features of melanoma cells. In addition, our results provide an explanation to the apparent paradox regarding the effect of IGF1 inhibition on migration and invasion by suggesting that this effect is governed by the status of cell-cell adhesion, particularly AJs: in cells capable of establishing cell-cell contacts, low IGF1 signaling forces AJ formation and prevents migration whereas it triggers the dissemination of cells that are unable to make cell-cell junctions. This important point is mentioned in the Discussion section of our manuscript.

We are now showing the expression level of IGF1R* in new Extended Data Figure 7g.

(**Figure 5e**) Migration of A375 human melanoma cells stably expressing either a control shRNA (shCTRL) or an shRNA directed against *NECTIN1* (shNECTIN1) in a transwell assay after 12 hours of serum starvation in the presence or absence of 100 ng/mL IGF1. Mean \pm SD of 3 independent experiments. Paired two-tailed t-test. (**Extended Data Figure 7g**) Western Blot analysis of IGF1R levels in A375 human melanoma cells stably transduced with a doxycycline-inducible vector expressing either GFP or a constitutive form of IGF1R (CD8-IGF1R, IGF1R*) and treated with doxycycline for 12 hours. CD8-IGF1R (estimated size: 70 kDa) appears below endogenous IGF1R (estimated size: 95 kDa). Representative example of 3 independent experiments.

17) Data in figure 6, How do the levels of IGF/IGFR relate to any prognostic feature of melanomas?

Our cohort in figure 6 only contains 20 samples and is therefore too small to draw any meaningful correlations with clinical features. From the large TCGA cohort of human cutaneous melanoma, there was a trend towards an association between low *IGF1R* expression and shorter overall survival (log-rank test: p-value = 0.088, Gehan-Breslow-Wilcoxon test: p-value = 0.013, see figure for reviewers only below) and a significant

correlation between low *IGF1R* expression and shorter progression-free survival (new Extended Data Figure 7k). *IGF1* expression did not correlate with patient survival. Additionally, similar to *NECTIN1* copy number, *IGF1R* expression decreases with disease stage but this is not statistically significant (one-way ANOVA: p-value = 0.12, see figure for reviewers only below). Our data suggest that low IGF1 signaling can impact melanoma cells locally so it might be difficult to observe at the level of the whole tumor strong correlations between global IGF1 levels or the ability of melanoma cells to sense the IGF1 signal, and the propensity of these cells to metastasize.

(Left) Overall survival curves of patients with melanoma stratified according to *IGF1R* mRNA expression (top and bottom n = 104 samples). TCGA cohort. Log-rank test: p = 0.088, Gehan-Breslow-Wilcoxon test: p = 0.013. (Extended Data Figure 7k) Progression-free survival curves of patients with melanoma stratified according to *IGF1R* mRNA expression (top and bottom n = 104 samples). TCGA cohort. Log-rank test. (Right) *IGF1R* mRNA expression across disease stages in the TCGA cohort of cutaneous melanoma (n = 406). One-way ANOVA: p = 0.12.

18) In those lines where the nectin high melanomas more proliferative (high ki67), and the IGF high?

To address these questions, we performed immunofluorescence analyses on 2 new tissue-microarrays of melanoma biopsies with either an antibody against *NECTIN1* and an antibody against Ki67 or an antibody against phospho-IGF1R and an antibody against Ki67. Staining intensity was assessed using a scoring system with a scale 0-3 for each marker. No significant differences were observed in the proliferation rates of tumors with different *NECTIN1* expression levels or IGF1 signaling levels. Note that our proliferation data obtained from zebrafish tumors (measuring phospho-H3 levels) is consistent with these results.

Left: Ki67 vs NECTIN1 levels on a tissue-microarray of 176 human melanomas. One-way ANOVA p-value = 0.13. Right: Ki67 vs p-IGF1R levels on a tissue-microarray of 176 human melanomas. One-way ANOVA p-value = 0.29.

Minor points

19) in western blots if you are altering cytoskeleton, you should use other loading control-not Actin

We have repeated all of our experiments potentially affecting the cytoskeleton in order to use GAPDH instead of ACTIN as a loading control in Western Blots. We have not noted any differences between these two housekeeping genes.

20) E-cadherin to N-cadherin switch has been linked to melanoma invasion, but authors say that N-cadherin is downregulated in the migratory process they describe, can authors discuss this apparent discrepancy?

Our data indicates that the increase in the migration and dissemination of melanoma cells in response to a drop in IGF1 signaling upon *NECTIN1* inactivation is independent of the differentiation state of the cells, in particular, of their basal migratory capacity (Figure 3g, Extended Data Figure 4g-i). This suggests that the action of *NECTIN1* is at least partially independent of the E-cadherin to N-cadherin switch associated with a transition from a differentiated, proliferative state to a more undifferentiated, migratory state. We were surprised to see that cell lines that have lost E-cadherin expression, such as A375 cells, formed robust adherens junctions upon serum withdrawal or IGF1

signaling inhibition (Figure 3h and Extended Data Figure 4k), and that these junctions contained N-cadherin. Note that N-cadherin is known to form adherens junctions in the nervous system (Miyamoto et al, 2015). In our models, the formation of these junctions, rather than the expression of N-cadherin itself, appears to inhibit the migration of melanoma cells. Consequently, the loss of these junctions (for example upon *NECTIN1* inactivation) favors migration. In our proteomic analyses, the inactivation of *NECTIN1* and the subsequent loss of adherens junctions was associated with a decrease in the presence of N-cadherin on the cell-surface (Figure 4a), but not with a change in CDH2 expression (data not shown). It is therefore conceivable that the modulation of the presence of cadherins on the surface of melanoma cells by external (e.g. IGF1) or internal (e.g. *NECTIN1*) factors is largely independent of the transcriptional state of these cells. Along those lines, it should be noted that the E-cadherin to N-cadherin switch may be a marker, more than a driver, of the transcriptional changes accompanying the transition to a migratory state in melanoma. We have elaborated on this point in the discussion of our manuscript.

Reviewer #3:

Remarks to the Author:

In this study, the authors investigated the role of NECTIN1 loss, which was found to occur in 55% of human melanoma cases in TCGA, in melanoma metastasis. NECTIN1 loss seemed to occur predominantly in metastatic melanoma lesions and was found to be associated with worse overall survival. Knockout of NECTIN1 in a zebrafish model was found to promote tumor spread and was associated with greater tumor elasticity. In vitro, NECTIN1 knockdown was found to promote a migratory phenotype in serum-free conditions despite slower proliferation of the NECTIN1 knockdown cells. The increase in migratory capacity was dependent on an increase in Integrin/SRC/FAK signaling, diminished focal adhesion formation and was shown to occur in the context of low IGF1. Knockout of IGFR in the NECTIN1-knockout zebrafish blocked tumor spreading capacity of the cells. Finally, the authors show that NECTIN negative melanomas typically lack focal adhesions and that focal adhesions typically occur in NECTIN1 positive melanomas, in areas of low IGF1.

The authors present a very compelling study of great significance to the melanoma

research community showing a novel genetic driver of melanoma metastasis. Overall, the study is thorough, well conducted, and mechanistically strong. The manuscript is written well, is very clear and provides the appropriate context and references. There are a few minor concerns.

We thank the reviewer for their suggestions on how to improve our manuscript. We have followed recommendations and addressed the different points below.

1. The figures could be improved for clarity. The details in Figure 1b are difficult to see and several figures (for example Figure 1 and Figure 3) have panels which appear in random order (panel order should follow the left to right, top to bottom sequence).

We apologize for the low quality of some images that was likely due to a loss of resolution during file transfer at initial submission. We have now increased figure resolution and enlarged panel 1b to make details more visible. We have also reorganized all figures to include new panels and ensure that panel numbers follow their order of appearance in the main text.

2. Could the authors clarify how heterogeneous NECTIN1 was scored in the human melanoma TMA? They mentioned in the manuscript that some tissues (mostly primary) had areas of high and low expression, so what numerical score would this be? Mainly, how does the staining scores depicted in Figure 1d relate to the Extended Figure 1c images (which score would each example have)?

We adopted the standard guidance of the American Society of Clinical Oncology (ASCO)/College of American Pathologists (CAP) guidelines on HER2 testing to interpret our NECTIN1 IHC stains (Ahn et al, J Pathol Transl Med 2020). The stains showed as examples in Extended Data Figure 2c illustrate the different scores from 0 (absent) to 3 (high). We have now modified this figure and expanded our Methods section to clarify this point. Thank you for pointing this out.

(**Extended Data Figure 2c**) Representative histology and scoring system used for the evaluation of NECTIN1 expression by immunohistochemistry, related to Figure 1d. 0 = Negative (<1% tumor cells immunoreactive); 1= weak and incomplete staining in < 10% of targeted cells, 2 = weak/moderate heterogeneous staining in > 10% of targeted cells, or 3 = strong and complete homogenous staining in > 10% of targeted cells. Scale bar: 50 μ m.

3. For the patients whose survival was measured in Figure 1e, are these the primary tissues or metastatic tissues or both? If NECTIN1 deletion is assessed in primary tumors only, would you see similar results in regards to survival (ie is NECTIN1 loss in primary tumors prognostic of worse survival)?

Figure 1e was based on the entire TCGA cohort of cutaneous melanoma samples with copy-number information (358 samples), without distinguishing between primary and metastatic samples. We have run a new analysis excluding samples from metastatic tumors (M1, corresponding to stage IV cases) and have found a similar result (on 318 samples) i.e. that NECTIN1 deletions are associated with shorter overall survival. This result indeed suggests that *NECTIN1* loss in primary tumors is a factor of poor prognosis.

Overall survival curves of patients with primary melanoma (M0) stratified according to NECTIN1 copy number status (> -0.25: no deletion, n=173; < -0.25: deletion, n=145). Log-rank test.

4. The spheroid experiments are not especially convincing. I would argue that based on the images, the shCTRL also don't really form spheroids without FBS (they look very flat compared to typical spheroids. It may be helpful to quantify these in some way.

Thank you for raising this point. We agree that, in the absence of serum, the structures formed by cells expressing a control shRNA in low-attachment conditions are rather flat

and look less like spheroids than in the presence of serum. This may be explained by the fact that they do not grow in the absence of serum. We have repeated these experiments with fluorescent labeling of melanoma cells to be able to image them with a confocal microscope. These images (now displayed in new Extended Data Figure 4a) give a better idea of the shape of these 3D structures. We have modified the main text of our manuscript to avoid the term “spheroid” to describe these structures in the absence of serum.

(Extended Data Figure 4a) Confocal images of the 3D structures formed by 5,000 A375 human melanoma cells stably expressing a control shRNA (shCTRL) or an shRNA directed against *NECTIN1* (shNECTIN1) after 4 days in low-attachment conditions in the absence of serum. Cells were stained with fluorescent dyes (DiD and DiO) before plating. Scale bar: 300 μm . Representative of 3 independent experiments.

Yet, the 3D structures formed by *NECTIN1*-deficient cells are strikingly different from those formed by *NECTIN1*-proficient cells. We have now quantified this observation by measuring the circularity of the colonies formed by melanoma cells in low-attachment conditions upon both shRNA and CRISPR manipulation of *NECTIN1* (new Figure 3b and Extended Data Figure 4d).

(**Figure 3a**) Images of the 3D structures formed by 10,000 A375 human melanoma cells stably expressing a control shRNA (shCTRL) or an shRNA directed against *NECTIN1* (shNECTIN1) after 8 days in low-attachment conditions in the presence or absence of serum (FBS). Scale bar: 200 μ m. Representative of 9 independent experiments. (**Figure 3b**) Circularity of the colonies formed by A375 human melanoma cells in low-attachment conditions as shown in **a**. Circularity was measured using ImageJ and multiplied by 100 for ease of representation. A perfect circle would have a circularity of 100. Mean \pm SD of 9 independent colonies per condition. Paired two-tailed *t*-test.

(**Extended Data Figure 4c**) Analysis of the images of colonies formed by wildtype (WT) or *NECTIN1*-knockout (KO) A375 human melanoma cell clones in low-attachment conditions and in the absence of serum as shown in **b**. (**Extended Data Figure 4d**) Circularity of the colonies shown on the left, as measured with ImageJ (multiplied by 100 for ease of representation). Mean \pm SD of 5 independent colonies per condition. Two-tailed *t*-test.

To further characterize the behavior of *NECTIN1*-deficient cells in a 3D environment, we seeded a small number of A375 human melanoma cells expressing either a control shRNA or an shRNA directed against *NECTIN1* in collagen-rich matrix in the absence of

serum. We observed that after 7 days, *NECTIN1*-deficient cells were significantly more spread out than *NECTIN1*-control cells. We quantified this by measuring the number of spatially separated groups of cells present in each colony using the particle analysis function of ImageJ. These results, displayed in new Figure 3c-d, support the idea that *NECTIN1* loss increases the spreading capacity of melanoma cells in a 3D environment.

(**Figure 3c**) Top: Images of the colonies formed by 1,000 A375 human melanoma cells stably expressing a control shRNA (shCTRL) or an shRNA directed against *NECTIN1* (shNECTIN1) after 7 days in collagen-rich matrix in the absence of serum (FBS). Scale bar: 50 μ m. Representative of 3 independent experiments. Bottom: Analysis of the images shown on top using ImageJ. (**Figure 3d**) Spread of the colonies formed by A375 human melanoma cells in collagen-rich matrix as shown in c, bottom. Spread was measured using ImageJ as the number of individual particles detected in each colony. Mean \pm SD. Two-tailed *t*-test.

5. Please include the +FBS control for *NECTIN* knockdown in Extended data Figure 4D.

We have now added back this control that we omitted to save space. We have also included a quantification of the number of stress fibers per cell based on these experiments.

(Extended Data Figure 4k) Immunofluorescence analysis of N-cadherin (green) and α -actinin (red) in A375 human melanoma cells stably expressing a control shRNA (shCTRL) or an shRNA directed against *NECTIN1* (shNECTIN1) and cultured in the presence or absence of serum (FBS) for 12 hours. Scale bar: 10 μ m. Representative images of 5 independent experiments. **(Extended Data Figure 4l)** Quantification of the number of stress fibers per cell based on α -actinin staining on immunofluorescence images as shown in (i). Mean \pm SD of n=29, 26, 24, 20 cells for shCTRL+FBS, shCTRL-FBS, shNECTIN1+FBS, and shNECTIN1-FBS, respectively. Paired two-tailed *t*-test. **: $p < 0.01$, ***: $p < 0.001$.

6. The western blot for pSRC in Extended Data Fig. 6f could use more clarity, the decrease in pSRC with integrin inhibition is not obvious or striking. Please include quantification of key western blots.

We have now quantified the main western blots from several independent experiments (those presented in Figure 4i and Extended Data Figure 6h are shown below). We agree that the decrease in p-SRC upon integrin avb3/5 inhibition is not major, although statistically significant. This might be due to the possibility that several integrins contribute to the motility of *NECTIN1*-deficient cells (new Figure 4d) and thus also to the activation of the FAK/SRC pathway. However, considering the very strong blockade of cell migration by integrin avb3/5 inhibitors (Figure 4f), the partial reduction in p-SRC suggests that additional pathways downstream of integrins may contribute to the migration of *NECTIN1*-deficient cells. We are now mentioning this point in the manuscript.

(Figure 4i) Western Blot analysis of FAK and SRC phosphorylation levels in A375 human melanoma cells stably expressing a control shRNA (C) or an shRNA directed against *NECTIN1* (sh) in the absence of serum. Representative example of 5 independent experiments. Signal intensity was normalized to GAPDH and presented as a ratio to the vehicle-treated shCTRL condition. Paired two-tailed *t*-test. *: < 0.05; **: < 0.01. **(Extended Data Figure 6h)** Western blot analysis of FAK and SRC phosphorylation levels in A375 human melanoma cells stably expressing a control shRNA (C) or an shRNA directed against *NECTIN1* (sh), serum-starved for 12 hours and treated with an integrin avb3/avb5 inhibitor (ITGi, echistatin), a FAK inhibitor (FAKi, PF562271) or a SRC inhibitor (SRCi, dasatinib) for 6 hours. veh: vehicle-treated. Representative example of 4 independent experiments. Signal intensity was normalized to GAPDH and presented as a ratio to the vehicle-treated shCTRL condition. Paired two-tailed *t*-test. *: < 0.05; **: < 0.01; ***: < 0.001.

7. Does dextran-treated charcoal-stripped serum also fail to activate adherens junctions and instead increase integrin/FAK/SRC signaling in *NECTIN1*-shRNA cells like serum-free media does?

This is an interesting point. We had verified by immunofluorescence that, as serum withdrawal, dextran-treated charcoal-stripped serum was unable to induce the formation of AJs between *NECTIN1*-deficient melanoma cells. We are now showing the corresponding pictures in new Extended Data Figure 7a. We have now also assessed the effect of this treated serum on FAK/SRC signaling by western blotting. We indeed observed that dextran-treated charcoal-stripped serum elicited similar activation of FAK and SRC as serum withdrawal in cells expressing an shRNA against *NECTIN1*, indicating that growth factor or hormone depletion is the cause of this activation. We have added this result to Extended Data Figure 7.

(Extended Data Figure 7a) Immunofluorescence analysis of α -catenin (green) in A375 human melanoma cells stably expressing a control shRNA (shCTRL) or an shRNA directed against *NECTIN1* (shNECTIN1) and cultured in the presence or absence of serum (FBS), or in the presence of dextran-treated/charcoal-stripped serum (treated FBS) for 12 hours. Scale bar: 10 μ m. Representative images of 3 independent experiments. **(Extended Data Figure 7b)** Western Blot analysis of FAK and SRC phosphorylation levels in A375 human melanoma cells stably expressing an shRNA directed against *NECTIN1* (shNECTIN1) in the presence or absence of serum (+FBS or -FBS), or in the presence of dextran-treated/charcoal-stripped serum (tFBS) for 12 hours. Representative example of 3 independent experiments.

8. It may help to have a summary diagram of the presented mechanism moved to the main Figure 7 as opposed to the Extended Data Figure 8.

Thank you for this suggestion. We have moved the model to main Figure 6.

Decision Letter, first revision:

Our ref: NG-A58083R

5th May 2022

Dear Dr. Zon,

Thank you for submitting your revised manuscript "Frequent deletion of adhesion gene NECTIN1 triggers melanoma dissemination upon local IGF1 depletion" (NG-A58083R). It has now been seen by the original referees and their comments are below. The reviewers find that the paper has improved in revision, and therefore we'll be happy in principle to publish it in Nature Genetics, pending minor revisions to comply with our editorial and formatting guidelines.

Sincerely,

Safia Danovi
Editor
Nature Genetics

Reviewer #1 (Remarks to the Author):

The authors have done an excellent job addressing all my suggestions. The inclusion of mutant allele frequencies and analysis of primary tumor cell behaviors in different genetic backgrounds is excellent. The methods section and controls are also very well described. I believe this study will have a significant impact on the cancer field and is appropriate for publication in this journal.

Reviewer #2 (Remarks to the Author):

8Authors have addressed all my comments

Reviewer #3 (Remarks to the Author):

The authors have thoroughly addressed all of my concerns and I believe the manuscript is now ready for publication.

Author Rebuttal, first revision:Response to reviewers' comments

Reviewer #1:

Remarks to the Author:

The authors have done an excellent job addressing all my suggestions. The inclusion of mutant allele frequencies and analysis of primary tumor cell behaviors in different genetic backgrounds is excellent. The methods section and controls are also very well described. I believe this study will have a significant impact on the cancer field and is appropriate for publication in this journal.

Reviewer #2:

Remarks to the Author:

Authors have addressed all my comments

Reviewer #3:

Remarks to the Author:

The authors have thoroughly addressed all of my concerns and I believe the manuscript is now ready for publication.

We thank all reviewers for their positive comments and suggestions that helped improve our work.

Final Decision Letter:

In reply please quote: NG-A58083R1 Ablain

24th Aug 2022

Dear Dr. Ablain,

I am delighted to say that your manuscript "Loss of NECTIN1 triggers melanoma dissemination upon local IGF1 depletion" has been accepted for publication in an upcoming issue of Nature Genetics.

Your paper will be published online after we receive your corrections and will appear in print in the next available issue. You can find out your date of online publication by contacting the Nature Press Office (press@nature.com) after sending your e-proof corrections. Now is the time to inform your Public Relations or Press Office about your paper, as they might be interested in promoting its publication. This will allow them time to prepare an accurate and satisfactory press release. Include your manuscript tracking number (NG-A58083R1) and the name of the journal, which they will need when they contact our Press Office.

10Final Decision Letter:

Please note that Nature Genetics is a Transformative Journal (TJ). Authors may publish their research with us through the traditional subscription access route or make their paper immediately open access through payment of an article-processing charge (APC). Authors will not be required to make a final decision about access to their article until it has been accepted. Find out more about Transformative Journals

Authors may need to take specific actions to achieve compliance with funder and institutional open access mandates. If your research is supported by a funder that requires immediate open access (e.g. according to Plan S principles) then you should select the gold OA route, and we will direct you to the compliant route where possible. For authors selecting the subscription publication route, the journal's standard licensing terms will need to be accepted, including https://www.nature.com/nature-portfolio/editorial-policies/self-archiving-and-license-to-publish. Those licensing terms will supersede any other terms that the author or any third party may assert apply to any version of the manuscript.

Please note that Nature Portfolio offers an immediate open access option only for papers that were first submitted after 1 January, 2021.

10Final Decision Letter:

If you have not already done so, we invite you to upload the step-by-step protocols used in this manuscript to the Protocols Exchange, part of our on-line web resource, natureprotocols.com. If you complete the upload by the time you receive your manuscript proofs, we can insert links in your article that lead directly to the protocol details. Your protocol will be made freely available upon publication of your paper. By participating in natureprotocols.com, you are enabling researchers to more readily reproduce or adapt the methodology you use. [Natureprotocols.com](https://natureprotocols.com) is fully searchable, providing your protocols and paper with increased utility and visibility. Please submit your protocol to <https://protocolexchange.researchsquare.com/>. After entering your [nature.com](https://www.nature.com) username and password you will need to enter your manuscript number (NG-A58083R1). Further information can be found at <https://www.nature.com/nature-portfolio/editorial-policies/reporting-standards#protocols>

Sincerely,

Safia Danovi
Editor
Nature Genetics